# PROVABLY EXPLAINING NEURAL ADDITIVE MODELS

**Shahaf Bassan**[1*]     **Yizhak Yisrael Elboher**[1*]     **Tobias Ladner**[2*]
**Volkan Şahin**[2]     **Jan Křetínský**[3]     **Matthias Althoff**[2]     **Guy Katz**[1]

[1] Hebrew University of Jerusalem, Israel,
[2] Technical University of Munich, Germany,
[3] Masaryk University, Czech Republic
   Contact: `shahaf.bassan@mail.huji.ac.il`,
   `yizhak.elboher@mail.huji.ac.il`, `tobias.ladner@tum.de`

## ABSTRACT

Despite significant progress in post-hoc explanation methods for neural networks, many remain heuristic and lack provable guarantees. A key approach for obtaining explanations with provable guarantees is by identifying a *cardinally-minimal* subset of input features which by itself is *provably sufficient* to determine the prediction. However, for standard neural networks, this task is often computationally infeasible, as it demands a worst-case *exponential* number of verification queries in the number of input features, each of which is NP-hard. In this work, we show that for Neural Additive Models (NAMs), a recent and more interpretable neural network family, we can *efficiently* generate explanations with such guarantees. We present a new model-specific algorithm for NAMs that generates provably cardinally-minimal explanations using only a *logarithmic* number of verification queries in the number of input features, after a parallelized preprocessing step with logarithmic runtime in the required precision is applied to each small univariate NAM component. Our algorithm not only makes the task of obtaining cardinally-minimal explanations feasible, but even outperforms existing algorithms designed to find the relaxed variant of *subset-minimal* explanations — which may be larger and less informative but easier to compute — despite our algorithm solving a much more difficult task. Our experiments demonstrate that, compared to previous algorithms, our approach provides provably smaller explanations than existing works and substantially reduces the computation time. Moreover, we show that our generated provable explanations offer benefits that are unattainable by standard sampling-based techniques typically used to interpret NAMs.

## 1    INTRODUCTION

Various methods have been proposed to explain neural network predictions. Classic additive feature attribution approaches — such as LIME (Ribeiro et al., 2016), SHAP (Lundberg & Lee, 2017), and IG (Sundararajan et al., 2017) — assume near-linear behavior in a local region around the instance. Other methods, like Anchors (Ribeiro et al., 2018) and SIS (Carter et al., 2019), aim to identify a (nearly) sufficient subset of input features — referred to here as an *explanation* — that determines the prediction. While Anchors and SIS rely on probabilistic sampling and lack provable sufficiency guarantees, recent work has shown that neural network verification tools can serve as a backbone for generating provably sufficient explanations (Wu et al., 2023; Bassan & Katz, 2023; La Malfa et al., 2021; Izza et al., 2024), making them particularly valuable in safety-critical domains (Marques-Silva & Ignatiev, 2022). In this context, *smaller* sufficient explanations are typically preferred, as *minimality* is considered an additional key interpretability property (Ignatiev et al., 2019; Carter et al., 2019; Darwiche & Hirth, 2020; Ribeiro et al., 2018; Barceló et al., 2020b).

However, while such explanations are highly desirable, generating them for standard neural networks is notoriously computationally challenging (Barceló et al., 2020b). In particular, obtaining *cardinally-minimal* explanations, requires, in the worst case, an *exponential* number of neural network verification queries (Barceló et al., 2020b; Ignatiev et al., 2019; Bassan & Katz, 2023), each

---

*Equal contribution.

being NP-hard (Katz et al., 2017; Sälzer & Lange, 2021), rendering the task infeasible even for toy examples (Ignatiev et al., 2019). Consequently, existing methods focus on *subset-minimal* explanations (Wu et al., 2023; Bassan & Katz, 2023; Bassan et al., 2025b), which are typically suboptimal in size, potentially large, and thus less informative than their cardinally minimal counterparts. Moreover, even these approaches remain limited to relatively small models, as they still require a linear number of verification queries (Wu et al., 2023; Bassan et al., 2025c).

**Our contributions.** Since computing cardinally-minimal sufficient explanations is provably intractable for general neural networks, a natural question arises: *Can certain neural architectures with more interpretable structures enable efficient computation of such explanations?* Although this task remains challenging even for simplified models such as *binarized* neural networks or those with a *single* hidden layer (Adolfi et al., 2025; Barceló et al., 2020a; Sälzer & Lange, 2021), we show in this work that it becomes tractable for a different class of neural architectures previously unexplored in this context: *Neural Additive Models (NAMs)* (Agarwal et al., 2021). NAMs are a widely adopted architecture that has received significant attention in recent years (Agarwal et al., 2021; Radenovic et al., 2022; Bechler-Speicher et al., 2024; Kim et al., 2024; Zhang et al., 2024). By enforcing an additive structure over input features, NAMs support interpretable, per-feature contributions while maintaining the expressive capabilities of neural networks.

**Our NAM-specific algorithm vs. previous algorithms.** Unlike existing algorithms that require a linear number of verification queries in the number of input features for subset-minimal explanations — or an exponential number for cardinally-minimal ones — our approach exploits the additive structure of NAMs to compute provably cardinally minimal explanation subsets using only a *logarithmic* number of verification queries. This is realized by introducing a highly parallelized preprocessing step — each operating independently on a small univariate component of the NAM — enabling a substantial overall efficiency gain. As a result, our method yields explanations that are both provably sufficient and cardinally-minimal, far more efficiently than standard algorithms typically applied to neural networks. Experiments using state-of-the-art verifiers confirm that our algorithm generates explanations substantially faster and produces notably smaller subsets than prior approaches.

**Our provable NAM explanations vs. standard sampling interpretations.** NAMs are typically interpreted by sampling input points and visualizing the behavior of each univariate function (Agarwal et al., 2021; Radenovic et al., 2022). Thanks to their additive structure, these models allow per-feature contributions to be examined individually. We show that purely sampling-based methods can yield misleading interpretations, whereas our provably sufficient explanations avoid this by design, underscoring their importance in safety-critical domains.

Overall, our work advances explanations with provable guarantees in two ways: (i) it introduces the first method for certifiable explanations in NAMs, boosting their trustworthiness in safety-critical settings, and (ii) by *efficiently* generating provable explanations, NAMs — unlike general neural networks — open a path toward interpretable architectures where such guarantees can be derived at scale. A central challenge ahead is to design models that balance high expressivity and accuracy with efficient provable explanations, and we view our work as a significant first step in that direction.

## 2 PRELIMINARIES

### 2.1 NOTATION

We denote scalars with lower-case letters, vectors with bold lower-case letters, and sets in calligraphic font. The $i$-th entry of a vector $\mathbf{x}$ is denoted by $\mathbf{x}_i$. For $n \in \mathbb{N}$, let $[n] := \{1, \dots, n\}$.

### 2.2 NEURAL NETWORK VERIFICATION

Neural network verification aims to verify certain input-output relationships of neural networks. For a neural network $f \colon \mathbb{R}^n \to \mathbb{R}^c$, a neural network verifier *formally proves* that there does not exist an input $\mathbf{x} \in \mathbb{R}^n$ where both an input specification $\psi_{\text{in}}(\mathbf{x})$, and an *unsafe* output specification $\psi_{\text{out}}(f(\mathbf{x}))$ hold at the same time. Although this problem is NP-hard (Katz et al., 2017; Sälzer & Lange, 2021), these tools have seen rapid scalability improvements in recent years (Kaulen et al., 2025).

## 2.3 NEURAL ADDITIVE MODELS (NAMs).

A neural additive model (NAM) $f$ for a *regression* task, where $f : \mathbb{R}^n \to \mathbb{R}$, is defined as:

$$f(\mathbf{x}) := \beta_0 + \sum_{i=1}^{k} f_i(\mathbf{x}_i), \tag{1}$$

where each $f_i : \mathbb{R} \to \mathbb{R}$ is a univariate neural network, $\beta_0 \in \mathbb{R}$ is the intercept, and $f$ denotes the full NAM. For *binary classification*, we assume that an additional step function is applied: $f(\mathbf{x}) := \text{step}(\beta_0 + \sum_{i=1}^{k} f_i(\mathbf{x}_i))$, where $\text{step}(z) = 1$ if $z \geq 0$ and $\text{step}(z) = 0$ otherwise. In the *multi-class* setting with $c$ classes, the logit for class $j \in [c]$ is given by $f_j(\mathbf{x}) := \beta_{j,0} + \sum_{i=1}^{k} f_{j,i}(\mathbf{x}_i)$, and the model predicts $f(\mathbf{x}) := \arg\max_{j \in [c]} f_j(\mathbf{x})$.

In this work, we develop algorithms for all three settings — (i) regression, (ii) binary classification, and (iii) multi-class classification — but for clarity, we focus the main presentation on the binary classification case, with extensions for the other settings provided in Appendix E.

## 3 PROVABLY SUFFICIENT EXPLANATIONS FOR NEURAL NETWORKS

We begin by reviewing standard algorithms developed for *general* neural networks that identify provably minimal sufficient explanations. We note that we focus on *post-hoc* sufficient explanations for a specific input $\mathbf{x} \in \mathbb{R}^n$, i.e., for the output $f(\mathbf{x})$, and post-hoc indicates that the explanation is generated after the model has been trained.

**Sufficient Explanations.** A common method for interpreting the decisions of classifiers involves identifying subsets of input features $\mathcal{S} \subseteq [n]$ such that fixing these features to their specific values guarantees the prediction remains unchanged. Specifically, these techniques guarantee that the classification result remains consistent across *any* potential assignment within the complementary set $\bar{\mathcal{S}} := [n] \setminus \mathcal{S}$. While in the classic setting features in the complementary set $\bar{\mathcal{S}}$ are allowed to take on any possible feature values (Ignatiev et al., 2019; Darwiche & Hirth, 2020; Bassan & Katz, 2023), a more feasible and generalizable version restricts the possible assignments for $\bar{\mathcal{S}}$ to a bounded $\epsilon_p$-region (Wu et al., 2023; La Malfa et al., 2021; Izza et al., 2024). We use $(\mathbf{x}_{\mathcal{S}}; \tilde{\mathbf{x}}_{\bar{\mathcal{S}}}) \in \mathbb{R}^n$ to denote an assignment where the features of $\mathcal{S}$ are set to the values of the vector $\mathbf{x} \in \mathbb{R}^n$ and the features of $\bar{\mathcal{S}}$ are set to the values of another vector $\tilde{\mathbf{x}} \in \mathbb{R}^n$ within the $\epsilon_p$-region.

**Definition 1** (Sufficient Explanation). *Given a neural network $f$, an input $\boldsymbol{x} \in \mathbb{R}^n$, a perturbation radius $\epsilon_p \in \mathbb{R}_+$, and a subset $\mathcal{S} \subseteq [n]$, we say that $\mathcal{S}$ is a sufficient explanation concerning the query $\langle f, \boldsymbol{x}, \mathcal{S}, \epsilon_p \rangle$ on an $\ell_p$-norm ball $B_p^{\epsilon_p}$ of radius $\epsilon_p \in \mathbb{R}_+$ around $\boldsymbol{x}$ iff it holds that:*

$$\forall \tilde{\boldsymbol{x}} \in B_p^{\epsilon_p}(\boldsymbol{x}): \quad f(\boldsymbol{x}_{\mathcal{S}}; \tilde{\boldsymbol{x}}_{\bar{\mathcal{S}}}) = f(\boldsymbol{x}), \qquad \text{with } B_p^{\epsilon_p}(\boldsymbol{x}) := \{\tilde{\boldsymbol{x}} \in \mathbb{R}^n \mid \|\boldsymbol{x} - \tilde{\boldsymbol{x}}\|_p \leq \epsilon_p\}.$$

*We define $\text{suff}(f, \boldsymbol{x}, \mathcal{S}, \epsilon_p) = 1$ iff $\mathcal{S}$ constitutes a sufficient explanation with respect to the query $\langle f, \boldsymbol{x}, \mathcal{S}, \epsilon_p \rangle$, and $\text{suff}(f, \boldsymbol{x}, \mathcal{S}, \epsilon_p) = 0$ otherwise.*

Def. 1 can be formulated as a neural network verification query (Sec. 2.2). This method has been proposed by prior studies, which employed these techniques to validate the sufficiency of specific subsets (Wu et al., 2023; Bassan & Katz, 2023; La Malfa et al., 2021; Izza et al., 2024).

**Minimal Explanations.** Evidently, selecting the entire input set as the subset $\mathcal{S}$, that is, setting $\mathcal{S} := [n]$, yields a sufficient explanation. Nonetheless, the prevailing consensus in the literature is that smaller subsets tend to be more informative or meaningful (Ribeiro et al., 2018; Carter et al., 2019; Barceló et al., 2020b; Ignatiev et al., 2019). Consequently, there is considerable interest in identifying subsets that are not only sufficient but also satisfy some notion of minimality. We focus on two specific minimality criteria: cardinality minimality and subset minimality.

**Definition 2** (Minimal Sufficient Explanations). *Given a neural network $f$, an input $\boldsymbol{x} \in \mathbb{R}^n$, and a subset $\mathcal{S} \subseteq [n]$ that is a sufficient explanation concerning $\langle f, \boldsymbol{x}, \mathcal{S}, \epsilon_p \rangle$ on $B_p^{\epsilon_p}$ of radius $\epsilon_p$, then:*

1. *We say that $\mathcal{S}$ is a cardinally-minimal sufficient explanation (Barceló et al., 2020a; Bassan et al., 2024) concerning $\langle f, \boldsymbol{x}, \mathcal{S}, \epsilon_p \rangle$ iff there does not exist a sufficient explanation $\mathcal{S}'$ concerning $\langle f, \boldsymbol{x}, \mathcal{S}', \epsilon_p \rangle$ with $|\mathcal{S}'| < |\mathcal{S}|$).*

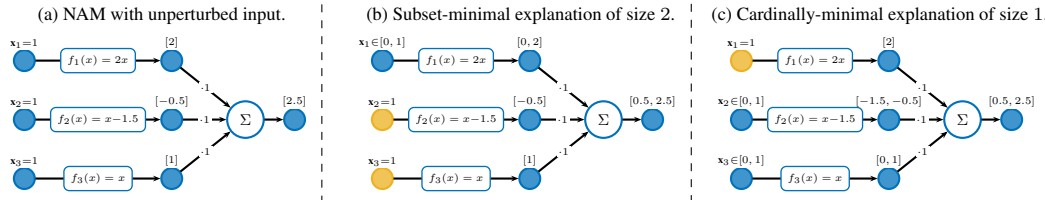

Figure 1: Comparison of a subset-minimal explanation and a cardinally-minimal explanation of a NAM. Both explanations (in yellow) in (b) and (c) are minimal, as perturbing any additional feature can lead the overall output to become negative.

2. We say that $\mathcal{S}$ is a subset-minimal sufficient explanation (Arenas et al., 2022; Ignatiev et al., 2019) concerning $\langle f, \boldsymbol{x}, \mathcal{S}, \epsilon_p \rangle$ iff any $\mathcal{S}' \subset \mathcal{S}$ is not a sufficient explanation concerning $\langle f, \boldsymbol{x}, \mathcal{S}', \epsilon_p \rangle$.

Minimal sufficient explanations can also be determined using neural network verifiers. This process requires executing multiple verification queries to ensure the minimality of the subset. Alg. 1 outlines such a procedure (Ignatiev et al., 2019; Wu et al., 2023; Bassan & Katz, 2023). The algorithm begins with an explanation $\mathcal{S}$ encompassing the entire feature set $[n]$ and iteratively tries to exclude a feature $i$ from $\mathcal{S}$, each time checking whether $\mathcal{S} \setminus \{i\}$ remains sufficient. If $\mathcal{S} \setminus \{i\}$ is still sufficient, feature $i$ is removed; otherwise, it is retained in the explanation. This process is repeated until a subset-minimal sufficient explanation is obtained. Such a subset-minimal explanation of a NAM is visualized in Fig. 1 along with a cardinally-minimal explanation.

---

**Algorithm 1** Greedy Subset Minimal Explanation Search

---
**Input:** Neural network $f \colon \mathbb{R}^n \to \mathbb{R}^c$, input $\mathbf{x} \in \mathbb{R}^n$, perturbation radius $\epsilon_p \in \mathbb{R}_+$
1: $\mathcal{S} \leftarrow [n]$
2: **for each** feature $i \in [n]$ **do**                                     $\triangleright$ suff$(f, \mathbf{x}, \mathcal{S}, \epsilon_p)$ holds
3:      **if** suff$(f, \mathbf{x}, \mathcal{S} \setminus \{i\}, \epsilon_p)$ **then**
4:          $\mathcal{S} \leftarrow \mathcal{S} \setminus \{i\}$
5:      **end if**
6: **end for**
7: **return** $\mathcal{S}$                   $\triangleright$ $\mathcal{S}$ is a *subset-minimal* explanation concerning $\langle f, \mathbf{x}, \mathcal{S}, \epsilon_p \rangle$

---

## 4 PROVABLY CARDINALLY-MINIMAL SUFFICIENT EXPLANATIONS FOR NAMs

While generating cardinally-minimal sufficient explanations is computationally expensive for general neural networks, we present a highly efficient algorithm for NAMs in this section. NAMs are generally considered very interpretable due to the univariate functions, but sufficient guarantees can only be obtained through formal verification to avoid misleading conclusions (Fig. 2). Our algorithm consists of two main stages: (i) As a preprocessing step, we compute an "importance" interval for each feature $i \in [n]$ based on the univariate functions $f_i$ to obtain a total ordering. (ii) This allows us to perform a binary search over the sorted intervals to identify the cardinally-minimal sufficient explanation.

All full proofs are provided in Appendix C.

### 4.1 STAGE 1 — PARALLEL INTERVAL IMPORTANCE SORTING

This subsection outlines the first stage of our algorithm, which involves determining an ordering of the feature importance to be used in the subsequent phase. We adopt a notion of "importance" similar to that of (Barceló et al., 2020a), originally used for binary linear threshold models. Specifically, we say that feature $i \in [n]$ is more "important" than feature $j \neq i$ if perturbing feature $i$ causes a

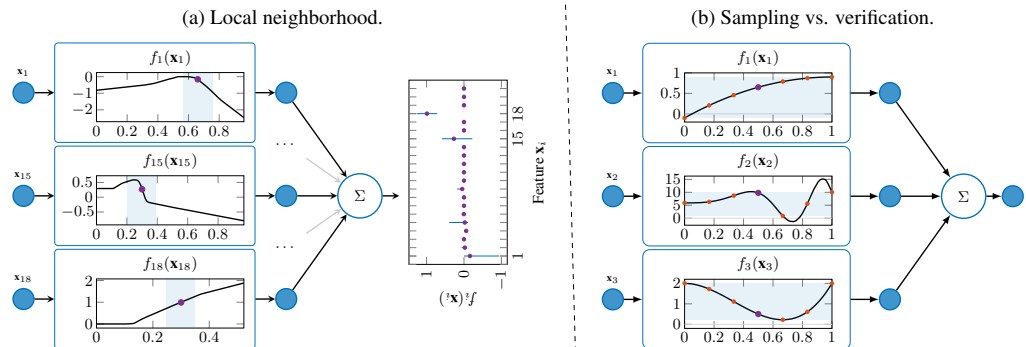

Figure 2: Sufficient explanations in NAMs: (a) Users must examine the neighborhood of an input for proper interpretation; e.g, users might wrongly conclude that feature 1 from the FICO HELOC dataset alone determines a positive output, but small changes in features 15 or 18 can flip the classification. (b) Outputs of continuous neighborhoods can be misleading if not verified, since sampling may miss extrema. For example, users might wrongly believe based on sampling that all features but feature 1 only have positive outputs and thus fixing feature 1 will always lead to a positive output of the entire NAM; however, the output of feature 2 can become so negative to flip the classification.

larger change in the final prediction $f(\mathbf{x})$ than perturbing feature $j$. In particular, we measure the derivation towards the decision boundary to flip the classification. Thanks to the additive structure of the NAM, this analysis can be conducted independently for each univariate component $f_i : \mathbb{R} \to \mathbb{R}$, allowing for a direct comparison of their individual importance.

Without loss of generality, let us assume that our binary classifier predicts $f(\mathbf{x}) = 1$, meaning that $\beta_0 + \sum_{i=1}^{k} f_i(\mathbf{x}_i) \geq 0$ (the case $f(\mathbf{x}) = 0$ follows symmetrically). In this setting, we perturb the input to each univariate component $f_i$ individually and measure how much the overall prediction decreases, i.e., towards the decision boundary. Therefore, for each $i \in [n]$, we want to find

$$\mathbf{x}_i^* = \underset{\tilde{\mathbf{x}}_i \in \mathcal{B}_p^{\epsilon_p}(\mathbf{x}_i)}{\arg\min} \ f_i(\tilde{\mathbf{x}}_i). \tag{2}$$

However, in contrast to the binary linear threshold models studied in (Barceló et al., 2020b), *exactly* determining these minimal values is usually computationally infeasible Katz et al. (2017). Fortunately, we do not need to find the exact minimum but only bounds $[l_i, u_i] \subset \mathbb{R}$ such that $l_i \leq f_i(\mathbf{x}_i^*) \leq u_i$, with sufficient precision to establish a total order over the input features. The procedure is outlined in Alg. 2.

Alg. 2 operates in parallel across each univariate component of the NAM. For each component, the algorithm begins by issuing an incomplete verification query to obtain an initial lower bound $f_i(\tilde{\mathbf{x}}_i)$, denoted by $l_i$ and $u_i$, evaluated over the domain $\tilde{\mathbf{x}}_i \in \mathcal{B}_p^{\epsilon_p}(\mathbf{x}_i)$. Subsequently, each thread independently conducts a binary search using verification queries to iteratively refine $[l_i, u_i]$, narrowing in on the true lower bound $f_i(\mathbf{x}_i^*)$. To quantify the deviation of the unperturbed output $f_i(\mathbf{x}_i)$ from these bounds, we define the relative differences $\Delta l_i$ and $\Delta u_i$. After each iteration within every parallel thread, the algorithm evaluates whether a full, non-overlapping sorting of all pairs $(\Delta l_1, \Delta u_1), \dots, (\Delta l_n, \Delta u_n)$ is possible. If such an ordering cannot yet be achieved, the bounds get iteratively refined. Finally, the ordering according to the determined importance is returned.

While the initially computed bounds $\alpha_i, \beta_i$ enclose the entire output domain of each component $i \in [n]$, the binary search narrows the bounds down around the minimum $f_i(\mathbf{x}_i^*)$ (or maximum for $f(\mathbf{x}) = 0$); thus, no longer covering the entire domain. As $f_i(\mathbf{x}_i^*)$ is just a scalar, this total order can be determined using a complete verifier.

This non-overlapping ordering provides a rigorous measure for the impact of perturbing a single feature through its corresponding component function $f_i(\mathbf{x}_i)$. This sorting forms the foundation for the next phase of the algorithm, which identifies a provably cardinally-minimal sufficient subset. The following proposition formalizes this first step of the argument:

**Proposition 1.** *Given a NAM $f$, an input $\mathbf{x} \in \mathbb{R}^n$ and a perturbation radius $\epsilon_p \in \mathbb{R}_+$, let Alg. 2 return a total list order over the input features according to their importance. Then, the following*

---

**Algorithm 2** Parallel Interval Importance Sorting

---

**Input:** NAM $f$, input $\mathbf{x} \in \mathbb{R}^n$, perturbation radius $\epsilon_p \in \mathbb{R}_+$

1: **for each** feature $i \in [n]$ *in parallel* **do**
2:     Extract initial bounds $\alpha_i, \beta_i$ for $f_i(\tilde{\mathbf{x}}_i)$ such that $\tilde{\mathbf{x}}_i \in \mathcal{B}_p^{\epsilon_p}(\mathbf{x}_i)$
3:     $l_i \leftarrow \alpha_i, u_i \leftarrow \beta_i$
4:     **while** True **do**
5:         $m_i \leftarrow \frac{l_i + u_i}{2}$
6:         **if** verify( $\forall \tilde{\mathbf{x}}_i \in \mathcal{B}_p^{\epsilon_p}(\mathbf{x}_i), \ f_i(\tilde{\mathbf{x}}_i) \geq m_i$ ) **then**
7:            $l_i \leftarrow m_i$
8:         **else**
9:            $u_i \leftarrow m_i$
10:        **end if**
11:        $\Delta l_i \leftarrow f_i(\mathbf{x}_i) - l_i \ ; \ \Delta u_i \leftarrow f_i(\mathbf{x}_i) - u_i$
12:        **if** for all $j \neq i$ it holds that: $\Delta u_i \geq \Delta l_j$ or $\Delta u_j \geq \Delta l_i$ **then**
13:           **break**
14:        **end if**
15:     **end while**
16: **end for**
17: **return** $\arg \text{sort}([(\Delta l_1, \Delta u_1), \ldots, (\Delta l_n, \Delta u_n)])$ in ascending order

---

*holds: For any sufficient explanation $\mathcal{S}$ that includes feature $i$, and for any feature $j \notin \mathcal{S}$ such that $i \prec j$ in the list ordering, the set $\mathcal{S} \setminus \{i\} \cup \{j\}$ is also a sufficient explanation.*

**Complexity.** The complexity of Alg. 2 is governed by the use of $\rho$ parallel processors, where each processor independently carries out a binary search. This binary search iteratively partitions based on the $\Delta u_i$ and $\Delta l_i$ bounds and terminates once the bounds of two distinct univariate components no longer overlap. Given the initial gap between the upper and lower bounds, $\alpha_i - f_i(\mathbf{x}_i)$, for each component $f_i$, and the precision for component $f_i$ defined by its minimal separation from the adjacent features in the sorted ordering — namely, $\xi_i := \min\{|\hat{\Delta}l_{i+1} - \hat{\Delta}u_i|, |\hat{\Delta}l_i - \hat{\Delta}u_{i-1}|\}$, with $\hat{\Delta}l_\square, \hat{\Delta}u_\square$ denoting the bounds in the last iteration. We can now prove that the number of neural network verifier calls is bounded by a (parallelized) *logarithmic* term, as formalized in the following proposition. Limitations and optimizations are further discussed in Appendix D.

**Proposition 2.** *Given $\rho$ parallelized processors, Alg. 2, performs an overall number of $T_p(n) = \mathcal{O}\left(\left(\frac{n}{p}\right)\log\left(\max_{i \in [n]}\left(\frac{\beta_i - \alpha_i}{\xi_i}\right)\right)\right) \xrightarrow{p \to n} \mathcal{O}\left(\log\left(\max_{i \in [n]}\left(\frac{\beta_i - \alpha_i}{\xi_i}\right)\right)\right)$ calls to the neural network verifier, each on a univariate component $f_i(\cdot)$, where $\xi_i := \min\{|\hat{\Delta}l_{i+1} - \hat{\Delta}u_i|, |\hat{\Delta}l_i - \hat{\Delta}u_{i-1}|\}$.*

**The edge case of $\xi_i \to 0$.** While Alg. 2 scales logarithmically with respect to the precision gap between optimal values of the univariate components, this gap could be arbitrarily small in theory. We address this edge case in Appendix D.1 and show that — even when $\xi_i$ approaches zero — the algorithm is still guaranteed to terminate after a *linear* number of steps in the size of the model encoding, given a model with ReLU activations. This remains a substantial improvement over the exponential worst-case behavior known for general neural networks. Crucially, our empirical results demonstrate that, in practice, this parameter is never close to zero, enabling an efficient sorting phase. A detailed ablation of this parameter is provided in Appendix F.5.

## 4.2 STAGE 2 — FEATURE SELECTION BASED ON THE DERIVED FEATURE INTERVALS

In this subsection, we will describe the second part of our algorithm that can obtain a provably cardinally-minimal sufficient explanation, given the derived interval orderings that were obtained from Alg. 2. In contrast to binary linear models (Barceló et al., 2020b), where the exact maximum and minimum contribution of each feature is known and one can simply select the top-$k$ features, NAMs do not offer this convenience. For NAMs, we only have access to bounds on each univariate component, and thus we must explicitly certify that fixing certain features is indeed sufficient. However, to the total order, we can apply a *binary search* to obtain the explanation, resulting in a logarithmic number of verification queries in the number of input features. However, to simplify the presentation of this algorithm, we will start by presenting a naive greedy approach that runs in a

linear number of steps similar to the one proposed by (Barceló et al., 2020b), and then move on to presenting the binary-search approach. The naive approach is depicted in Alg. 3.

---

**Algorithm 3** Greedy Cardinally-Minimal Linear Explanation Search

**Input:** NAM $f$, input $\mathbf{x} \in \mathbb{R}^n$, perturbation radius $\epsilon_p \in \mathbb{R}_+$

1: $\mathcal{S} \leftarrow [n]$
2: **for each** feature $i \in [n]$, ordered by Alg. 2 **do**            $\triangleright$ suff$(f, \mathbf{x}, \mathcal{S}, \epsilon_p)$ holds
3:     **if** suff$(f, \mathbf{x}, \mathcal{S} \setminus \{i\}, \epsilon_p)$ **then**
4:         $\mathcal{S} \leftarrow \mathcal{S} \setminus \{i\}$
5:     **end if**
6: **end for**
7: **return** $\mathcal{S}$            $\triangleright$ $\mathcal{S}$ is a *cardinally-minimal* explanation concerning $\langle f, \mathbf{x}, \mathcal{S}, \epsilon_p \rangle$

---

Alg. 3 closely mirrors the operation of Alg. 1: It begins by initializing the explanation $\mathcal{S}$ to the full feature set $[n]$, and then iteratively removes features, updating $\mathcal{S} \leftarrow \mathcal{S} \setminus \{i\}$, until reaching a minimal explanation. However, unlike Alg. 1, which is only guaranteed to converge to a *subset-minimal* explanation, Alg. 3 converges to the more challenging objective of finding a *cardinally-minimal* sufficient explanation. This stronger guarantee is enabled by the total ordering $(\hat{\Delta}l_i, \hat{\Delta}u_i)_{i=1}^n$ computed by Alg. 2, which ranks the features by their importance. This leads to the following proposition:

**Proposition 3.** *Given a NAM $f$, an input $\mathbf{x} \in \mathbb{R}^n$, and a perturbation radius $\epsilon_p \in \mathbb{R}_+$, Alg. 3 performs $\mathcal{O}(n)$ queries and returns a* cardinally-minimal *sufficient explanation. This stands in contrast to Alg. 1, which is only guaranteed to return a* subset-minimal *sufficient explanation.*

Alg. 3 can be significantly enhanced by replacing the linear ordering with a binary search strategy (Alg. 4). Crucially, this step is not possible with the naive, unsorted approach (Alg. 1), as it does not guarantee convergence to a cardinally-minimal explanation, and may not even yield a *subset-minimal* explanation. This is because, in an arbitrary feature ordering, there may be multiple points at which a non-sufficient subset becomes sufficient, making the binary search unreliable. In contrast, the preprocessing step in Alg. 2 imposes a structured sorting of features, which allows Alg. 4 to reliably converge to a *cardinally-minimal* sufficient explanation using only a *logarithmic* number of queries.

---

**Algorithm 4** Greedy Cardinally-Minimal Logarithmic Explanation Search

**Input:** NAM $f$, input $\mathbf{x} \in \mathbb{R}^n$, perturbation radius $\epsilon_p \in \mathbb{R}_+$

1: $F \leftarrow$ total order of features (Alg. 2)
2: $l \leftarrow 1$ ; $u \leftarrow n$
3: **while** $l \neq u$ **do**
4:     $m \leftarrow \lfloor \frac{l+u}{2} \rfloor$
5:     **if** suff$(f, \mathbf{x}, \{F[1], \ldots, F[m]\}, \epsilon_p)$ **then**
6:         $l \leftarrow m$
7:     **else**
8:         $u \leftarrow m - 1$
9:     **end if**
10: **end while**
11: $\mathcal{S} \leftarrow \{F[1], \ldots F[m]\}$
12: **return** $\mathcal{S}$            $\triangleright$ $\mathcal{S}$ is a *cardinally-minimal* explanation concerning $\langle f, \mathbf{x}, \mathcal{S}, \epsilon_p \rangle$

---

**Proposition 4.** *Given a NAM $f$, an input $\mathbf{x} \in \mathbb{R}^n$, and a perturbation radius $\epsilon_p \in \mathbb{R}_+$, Alg. 4 performs $\mathcal{O}(\log(n))$ queries and returns a* cardinally-minimal *sufficient explanation.*

**Overall complexity results.** By combining Alg. 2 with Alg. 4, we obtain a cardinally-minimal explanation for $\langle f, \mathbf{x}, \epsilon_p \rangle$. This unified algorithm yields a substantial efficiency gain, reducing the worst-case requirement of an *exponential* number of verification queries to only a *logarithmic* number of (parallelized) queries. The first segment of these queries operate by running verification queries on *univariate components* $f_i$ of the model, which are far smaller, and hence more efficient to verify than direct queries to $f$. The resulting complexity bound is formalized in the following theorem:

Table 1: Comparison of average explanation size and computation time.

| Method | Breast Cancer | | CREDIT | | FICO HELOC | |
|---|---|---|---|---|---|---|
| | Size ($\downarrow$) | Time [s] ($\downarrow$) | Size ($\downarrow$) | Time [s] ($\downarrow$) | Size ($\downarrow$) | Time [s] ($\downarrow$) |
| **Ours** | **4.00**±**4.24** | **35.60**±**1.34** | **3.76**±2.62 | **132.67**±**36.76** | **5.59**±1.80 | 317.92±222.07 |
| Lexicographic | 16.58±5.44 | 634.92±77.23 | 12.42±6.45 | 473.63±128.38 | 15.60±7.53 | **146.16**±188.07 |
| Sensitivity | 16.27±5.57 | 636.79±87.44 | 3.82±1.84 | 407.93±126.63 | 9.45±5.90 | 250.09±148.44 |

**Theorem 1.** *Running Alg. 2 and Alg. 4 obtains a* cardinally-minimal *sufficient explanation with* $T_p(n) = \mathcal{O}\left(\left(\frac{n}{p}\right)\log\left(\max_{i\in[n]}\left(\frac{\beta_i - \alpha_i}{\xi_i}\right)\right)\right) \xrightarrow[p\to n]{} \mathcal{O}\left(\log\left(\max_{i\in[n]}\left(\frac{\beta_i - \alpha_i}{\xi_i}\right)\right)\right)$ *queries to* $f_i(\cdot)$ *components, plus* $\mathcal{O}(\log n)$ *queries to* $f(\cdot)$. *In contrast, standard algorithms require* $\mathcal{O}(2^n)$ *verification queries to* $f(\cdot)$ *for a cardinally-minimal explanation, or* $\mathcal{O}(n)$ *verification queries to* $f(\cdot)$ *for only a subset-minimal explanation.*

As discussed earlier, our complexity analysis is parameterized by the separation between the minima of the univariate components $\xi_i$, which is typically non-negligible in practice, though some $\xi_i$ may be small. To address this, we show that running Alg. 2 followed by Alg. 4 takes time *linear* in the encoding size of the model. We further prove matching query bounds: the task can be solved with at most a linear number of queries, and any algorithm requires at least a logarithmic number of queries in the encoding size of $f$.

**Theorem 2.** *(Informally.) Let $f$ be a NAM, $\boldsymbol{x} \in \mathbb{R}^n$ an input, and $\epsilon_p \in \mathbb{R}$. Computing the size of a cardinally-minimal sufficient explanation for $\langle f, \boldsymbol{x}, \epsilon_p \rangle$ requires, in the worst case, at most $\mathcal{O}(m)$ queries to an NP oracle (Claim 1) and at least $\mathcal{O}(\log(m))$ such queries (Lemma 5), where $m$ is the encoding size of $f$. Moreover, running Alg 2 followed by Alg 4 terminates after at most $\mathcal{O}(m)$ queries (Proposition 5).*

## 5 EVALUATION

**Experimental Setup.** We implemented our main algorithmic approach (Alg. 2 followed by Alg. 4) using $\alpha$-$\beta$-CROWN as the backend verifier, the current state-of-the-art in neural network verification (Wang et al., 2021; Zhou et al., 2024; Kotha et al., 2023; Kaulen et al., 2025; Chiu et al., 2025). We conducted extensive experiments on four widely used tabular-data benchmarks in the context of NAMs (Agarwal et al., 2021; Radenovic et al., 2022): (i) Breast Cancer, (ii) CREDIT, (iii) FICO HELOC, all of which are prominent in safety-critical domains. We adopted the same model architectures as prior work in the NAM literature (Agarwal et al., 2021; Radenovic et al., 2022). Evaluation details, additional experiments, and ablation studies are in Appendix F.

### 5.1 OUR ALGORITHM VS. PREVIOUS ALGORITHMS

We begin by comparing our results with prior algorithms proposed in the literature for obtaining provably minimal sufficient explanations. Since our method targets the much stronger notion of cardinally-minimal sufficient explanations for the first time, any naive baseline — that computes such explanations by exhaustively enumerating all $2^n$ input subsets, verifies their sufficiency, and selects the one with the smallest cardinality — would not finish with reasonable timeouts. Thus, we compare our approach to the more scalable task of finding subset-minimal explanations, a weaker notion of minimality, using the standard greedy algorithm employed by previous works (Wu et al., 2023; Bassan et al., 2025b; Izza et al., 2024; Ignatiev et al., 2019; La Malfa et al., 2021) (Alg. 1). Because subset-minimal explanations depend on feature orderings, we consider two setups: (i) a basic lexicographic ordering of features, and (ii) a more sophisticated reverse-sensitivity ordering, following prior approaches (Wu et al., 2023; Bassan et al., 2025b; Izza et al., 2024; Wu et al., 2024b).

The results in Tab. 1 demonstrate that our proposed algorithm achieves substantial improvements in both computation time and explanation size over previous algorithms. Beyond reducing explanation size compared to standard subset-minimal explanation algorithms (which follows by necessity, since we enforce a stronger notion of minimality), our method also achieves substantial runtime gains,

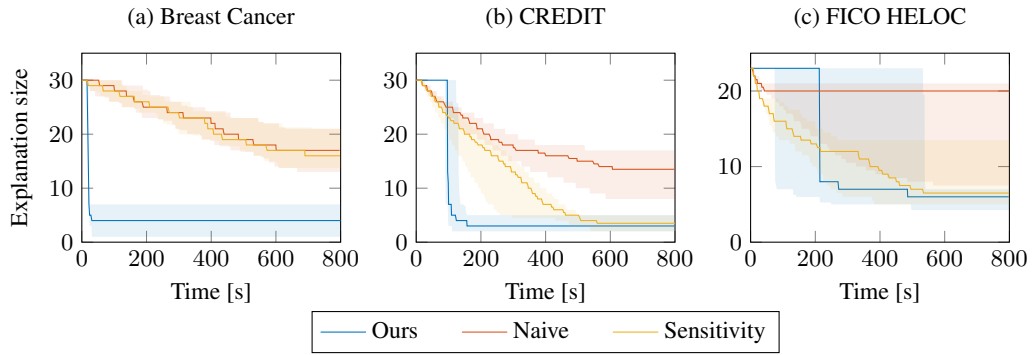

Figure 3: Explanation size over time for all datasets.

Table 2: Comparison against a purely sampling-based approach.

| Method | CREDIT | | | FICO HELOC | | |
|---|---|---|---|---|---|---|
| | Size ($\downarrow$) | Time [s] ($\downarrow$) | Sufficiency [%] ($\uparrow$) | Size ($\downarrow$) | Time [s] ($\downarrow$) | Sufficiency [%] ($\uparrow$) |
| **Ours** | 3.76±2.62 | 132.67±36.76 | **100.00** | 5.59±1.80 | 317.92±222.07 | **100.00** |
| **Sampling** | **2.67±3.10** | **5.10±0.26** | 31.37 | **3.04±3.05** | **6.89±2.57** | 25.49 |

despite solving a much harder task. This advantage stems from our NAM-specific algorithm, which requires only a *logarithmic* number of parallelized queries — rather than a linear number — executed over univariate components $f_i$, which are much faster to verify.

## 5.2 EXPLANATION PROGRESSION IN TIME

To further assess the advantages of our algorithm over prior methods, we analyze the explanation sizes produced by our approach in comparison to subset-minimal methods and track their evolution over time. This analysis is illustrated in Fig. 3. The results show that while subset-minimal approaches converge slowly and often stagnate in local minima. Our method — though it begins later due to the preprocessing step in Alg. 2, which sorts features with only a logarithmic number of parallelized queries — quickly outpaces them once sorting is complete. At this point, it requires significantly fewer queries, relying only on a binary search over the sorted features as in Alg. 4. This second phase is not only substantially faster but also provably attains the cardinally-minimal explanation, i.e., the global optimum, unlike subset-minimal approaches.

## 5.3 COMPARISON TO PURELY SAMPLING-BASED METHODS

NAMs are generally viewed as very interpretable as their univariate functions $f_i$ for each feature allow for simple visualizations (such as in Fig. 2). Most commonly, these visualizations are obtained through sampling over the respective feature domain to get a good approximation of each univariate function. However, we show in this experiment that this discretization through sampling and the resulting interpretations can be misleading. Peaks and other extrema that are missed through sampling can lead to insufficient explanations, which can be fatal in safety-critical domains. An extreme case is abstractly depicted in Fig. 2b, but we have also observed insufficient explanations in practice. To demonstrate this, we evaluate $1,000$ evenly-spaced samples instead of each verification query. After the explanations are generated, we test their sufficiency using $\alpha, \beta$-CROWN (Tab. 2): On the CREDIT and FICO HELOC datasets, more than half of the explanations obtained through sampling could not be verified. In contrast, all our explanations are sufficient by construction.

## 6 RELATED WORK

**Formal XAI.** Our work relates to the field of *formal XAI* (Marques-Silva, 2023), which seeks explanations with provable guarantees. Prior efforts have developed sufficient explanations for models

such as decision trees (Huang et al., 2021), linear models (Marques-Silva et al., 2020; Subercaseaux et al., 2025), and tree ensembles (Izza & Marques-Silva, 2021; Ignatiev et al., 2022; Audemard et al., 2022b; 2023). Other relevant works are on obtaining minimal sufficient explanations for neural networks (La Malfa et al., 2021; Wu et al., 2023; Izza et al., 2024; Bassan et al., 2023; 2025b), which rely on neural network verification queries. While such verifiers have become more scalable in recent years, computing such explanations is still costly, often requiring many (linear or exponential) verification queries. Our method takes a step toward reducing this cost by focusing on neural network families with more interpretable structure, and particularly look into NAMs.

Closer to our setting is the work of Harzli et al. (2023), which studies cardinally minimal sufficient explanations for *monotonic* neural networks, another interpretable model class. There, monotonicity simplifies certifying *sufficiency*, while *minimality* remains challenging. In contrast, for NAMs, certifying *sufficiency* is substantially harder, but *minimality* becomes easier due to the additive structure.

Also related to our work is the study of Barceló et al. (2020b), which shows, among many results, that cardinally minimal sufficient explanations can be computed in polynomial time for *linear (Perceptron) models* over *binary* inputs, using ideas such as sorting and selecting features. However, obtaining the same type of explanation for *neural* additive models over *continuous* inputs, which are far more expressive, introduces new algorithmic challenges: (i) computing and refining bounds for each univariate component using scalable neural network verification methods (Wang et al., 2021), (ii) parallelizing the certification of *univariate* components rather than the full network, and (iii) employing a more efficient selection phase that leverages these outer bounds, sorts the components, and applies a *logarithmic* search to obtain a cardinally minimal sufficient explanation more efficiently.

**Neural Additive Models (NAMs).** NAMs extend *Generalized Additive Models (GAMs)* (Hastie, 2017; Nelder & Wedderburn, 1972), a classic interpretable family of ML models (Caruana et al., 2015; Zhong et al., 2023; Liu et al., 2022; Bordt & von Luxburg, 2023; Enouen & Liu, 2025; Chen et al., 2020), by replacing each univariate component with a neural network, thereby combining interpretability with expressivity. First introduced by (Agarwal et al., 2021), NAMs achieved competitive accuracy on tabular tasks and were applied in healthcare and COVID-19 modeling. Subsequent works suggested potential refinements of their training and architecture (Radenovic et al., 2022; Chang et al., 2021; Bouchiat et al., 2024; Xu et al., 2023) and proposed additional variants (Bechler-Speicher et al., 2024; Jiao et al., 2024) and applications (Thielmann et al., 2024).

## 7 LIMITATIONS

Like all methods that obtain provably minimal and sufficient explanations, our approach depends on invoking neural network verification queries, which do not yet scale to state-of-the-art models. Still, neural network verification has advanced rapidly in recent years (Kaulen et al., 2025; Chiu et al., 2025; Wu et al., 2024a), and the scalability of our approach will improve alongside it. Importantly, our method offers two substantially critical improvements: (i) it reduces the number of queries from exponential (or linear, in relaxed tasks) to *logarithmic*, and (ii) it operates on *univariate* components $f_i$, where verification is far cheaper since the certified models are small and more interpretable by design compared to the entire large model $f$. Together, these make our algorithm far more practical for NAMs, and we show that it indeed efficiently produces explanations on standard benchmarks where prior algorithms fail. Our approach also relies on a complete verifier, which often come with numeric tolerances and timeout restrictions. A more detailed discussion on this can be found in Appendix D.

## 8 CONCLUSION

Provably minimal and sufficient explanations represent a highly desirable goal in explainability, as they offer certifiable guarantees on both faithfulness and conciseness. For standard neural networks, however, this task is computationally prohibitive, requiring an exponential number of verification queries. We present a NAM-specific algorithm that reduces the complexity from *exponential* to *logarithmic* parallelized queries, achieving dramatic gains in both speed and explanation size. Moreover, we show that these explanations reveal insights into NAMs that sampling-based methods cannot capture. Our work thus makes provable explanations feasible in practice and opens the door to extending them across other interpretable neural network families.

ACKNOWLEDGEMENTS

The work of Elboher, Bassan, and Katz was partially funded by the European Union (ERC, VeriDeL, 101112713). Views and opinions expressed are however those of the author(s) only and do not necessarily reflect those of the European Union or the European Research Council Executive Agency. Neither the European Union nor the granting authority can be held responsible for them. The work of Elboher, Bassan, and Katz was additionally supported by a grant from the Israeli Science Foundation (grant number 558/24). The work of Ladner, Şahin, and Althoff was supported by the German Research Foundation (Deutsche Forschungsgemeinschaft, DFG) under grant number AL 1185/33-1. Views and opinions expressed are however those of the author(s) only and do not necessarily reflect those of the European Union, or any other granting authority.

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

# Appendix

The appendix contains all proofs, optimizations, additional settings, and additional experiments that were mentioned throughout the paper:

**Appendix A** contains extended background on neural network verification and formal XAI.
**Appendix B** contains formalizations that will be used throughout the proofs.
**Appendix C** contains the proofs of Prop. 1 to 4.
**Appendix D** contains a theoretical discussion and practical optimizations on the importance sorting.
**Appendix E** contains extensions to multi-class classification and regression tasks.
**Appendix F** contains all experimental details and ablation studies.
**Appendix G** contains an LLM usage disclosure.

## A    EXTENDED BACKGROUND ON NEURAL NETWORK VERIFICATION AND FORMAL XAI

**Neural Network Verification.**  Neural network verification provides provable guarantees about the behavior of a neural network over a continuous input region. Classical SMT- and MILP-based approaches (Katz et al., 2017; Wu et al., 2024a; Katz et al., 2019; Tjeng et al., 2019; Ehlers, 2017) encode ReLU networks and specifications as logical or mixed-integer constraints, yielding exact guarantees but scaling only to small–to–medium models. Abstract interpretation methods (Singh et al., 2019; Gehr et al., 2018; Ferrari et al., 2022; Müller et al., 2022) instead propagate layer-wise over-approximations, producing fast yet incomplete robustness certificates. A major breakthrough came with linear-relaxation–based bound propagation, notably CROWN (Zhang et al., 2018) and its extensions (Wang et al., 2021; Chiu et al., 2025; Zhou et al., 2024; Shi et al., 2025), which compute tight dual linear bounds and function either as scalable incomplete verifiers or as strong relaxations within exact search procedures. Modern branch-and-bound (BaB) frameworks build on these relaxations to achieve *complete* verification at scale. In particular, $\alpha$-$\beta$-CROWN and related variants now dominate VNN-COMP (Brix et al., 2024), handling models with millions of parameters. Recent advances include tighter relaxations (e.g., SDP hybrids (Chiu et al., 2025)), cutting-plane enhancements (Zhou et al., 2024), and support for non-ReLU nonlinearities (Shi et al., 2025). Today, verification methods can routinely certify robustness for moderately large CNNs and ResNets, although significant challenges remain for transformers, highly complex architectures, and richer temporal or relational specifications.

**Formal XAI.** Our work is part of a line of research known as *formal explainable AI* (formal XAI) (Marques-Silva & Ignatiev, 2022), which investigates explanations equipped with provable guarantees (Yu et al., 2023; Darwiche & Ji, 2022; Darwiche, 2023; Shih et al., 2018; Azzolin et al., 2025; Audemard et al., 2021). Because producing such guaranteed explanations is often computationally intractable (Barceló et al., 2020a; Bassan et al., 2026; Marzouk et al., 2025a;b; Marzouk & De La Higuera, 2024; Amir et al., 2024; Blanc et al., 2021; 2022), much of the literature has concentrated on more restricted model classes (Marques-Silva & Ignatiev, 2023), including decision trees (Bounia, 2025; Arenas et al., 2022; Bounia, 2024; Bounia & Koriche, 2023), monotonic models (Marques-Silva et al., 2021; Harzli et al., 2023), and tree ensembles (Audemard et al., 2023; 2022a; Ignatiev et al., 2022; Bassan et al., 2025a; Boumazouza et al., 2021). Related to our works, are frameworks that have obtained such explanations for neural networks (Malfa et al., 2021; Bassan et al., 2023; Hadad et al., 2026; Wu et al., 2023; Labbaf et al., 2025; De Palma et al., 2025; Fel et al., 2023; Soria et al., 2025). Due to the high computational complexity of this problem, several approaches aim to alleviate it through abstractions (Bassan et al., 2025b; Boumazouza et al., 2026), smoothing techniques (Xue et al., 2023; Jin et al., 2025; Anani et al., 2025), or self-explaining frameworks (Alvarez Melis & Jaakkola, 2018; Bassan et al., 2025c;d; You et al., 2025). Another related line of work studies explanations with provable guarantees for linear models (Marques-Silva et al., 2020; Subercaseaux et al., 2025), which form a restricted subclass of additive models, as well as the theoretical analysis of (Bassan et al., 2025e), which investigates the computational complexity of generating such explanations for these models.

# B FORMALIZATIONS

## B.1 COMPUTATIONAL COMPLEXITY CLASSES

We assume readers are familiar with the standard complexity classes polynomial time (PTIME) and nondeterministic polynomial time (NP, coNP); see Arora & Barak (2009) for an introduction. NP and coNP capture the complexity of decision problems — problems whose output is simply "yes" or "no". In contrast, FP is the *functional* analogue of PTIME, describing the class of function problems whose outputs may be arbitrary values and can be computed in polynomial time.

We assume that the readers are familiar with the basic complexity classes of polynomial-time (PTIME), and non-deterministic polynomial-time (NP, coNP) (see an introduction in Arora & Barak (2009)). NP and coNP are problems that address *decision* problems, meaning they determine the complexity of problems with a yes/no answer. The complexity class FP is the *functional* variant of PTIME (meaning it returns a function as an output rather than a yes/no answer), and describes the class of function problems that can be solved in polynomial time.

The complexity class *OptP* of functions computable by taking the maximum (or minimum) of the output values over all accepting paths of an NP machine. Alternatively, a function $F \colon \Sigma^* \to \mathbb{Z}$ is in OptP if there exists a polynomial $p(\cdot)$ and a polynomial-time computable function $G \colon \Sigma^* \times \{0,1\}^{p(n)} \to \mathbb{Z}$ which we call the *value function* such that for every input $X \in \Sigma^*$:

$$F(X) = \max_{Y \in \{0,1\}^{p(|X|)}} G(X, Y), \tag{3}$$

where $Y$ is a witness of polynomial length, the function $G$ can be computed in deterministic polynomial time, and the integer output $G(X, Y)$ is polynomially bounded in $n = |X|$. We say that a problem is in OptP$[z(n)]$ if the solution $F(X)$ is bounded by $z(n)$.

The class $\text{FP}^{\text{NP}}[z(n)]$ consists of all function problems computable in polynomial time using at most $z(n)$ queries to an NP oracle. The seminal work of (Krentel, 1986) established that OptP$[z(n)] \subseteq \text{FP}^{\text{NP}}[z(n)]$, and moreover that every function in $\text{FP}^{\text{NP}}[z(n)]$ can be computed by solving an appropriate OptP$[z(n)]$ optimization problem followed by a polynomial-time postprocessing step, provided that $z(n)$ satisfies the "smoothness" condition defined in (Krentel, 1986).

In functional or optimization problems, a common convention is to work with *metric reductions* (Krentel, 1986). Formally, given some $f, g \colon \Sigma^* \to \mathbb{N}$, a metric reduction from $f$ to $g$ is a pair of polynomial-time computable functions $T_1 \colon \Sigma^* \times \mathbb{N} \to \mathbb{N}$ such that for all $\mathbf{x} \in \Sigma^*$ we have $f(\mathbf{x}) = T_2(\mathbf{x}, g(T_1(\mathbf{x})))$.

## B.2 UNIVARIATE NEURAL NETWORK FORMALIZATION

In this subsection, we will formalize the univariate neural network components $f_i \colon \mathbb{R} \to \mathbb{R}$ under our analysis. Let there be a univariate neural network component $f$ with $\kappa$ layers, and an input $\mathbf{x} \in \mathbb{R}^{n_0}$. The output of $\mathbf{y} := f_i(\mathbf{x}) \in \mathbb{R}$ is obtained as follows:

$$\mathbf{h}^0 := \mathbf{x}, \qquad \mathbf{h}^k := L_k(\mathbf{h}^{k-1}), \qquad \mathbf{y} = \mathbf{h}^\kappa, \quad k \in [\kappa],$$

where we use $L_k \colon \mathbb{R}^{n_{k-1}} \to \mathbb{R}^{n_k}$ to denote the operation of layer $k$ and is defined as

$$L_k(\mathbf{h}^{k-1}) := \sigma(W_k \mathbf{h}^{k-1} + \mathbf{b}^k),$$

with the weight matrix denoted as $W_k \in \mathbb{Q}^{n_k \times n_{k-1}}$, bias vector $\mathbf{b}^k \in \mathbb{Q}^{n_k}$, and activation function $\sigma \colon \mathbb{R}^{n_k} \to \mathbb{R}^{n_k}$, and number of neurons $n_k \in \mathbb{N}$. Following standard conventions, we assume that all weights and biases are rational numbers encoded in binary (as they correspond to trained parameters stored with fixed machine precision), and can be written in the form $\frac{p}{q}$ for some $p, q \in \mathbb{N}$. We additionally assume that $\sigma$ is the standard ReLU activation $\text{ReLU}(\mathbf{x}) := \max(0, \mathbf{x})$ function. A NAM is a regression or classification model whose core computation takes the form $f_1(\mathbf{x}_1) + \cdots + f_n(\mathbf{x}_n) + \beta_0$, where each $f_i$ is a univariate neural network and $\beta_0 \in \mathbb{Q}$. We define the *size* of a univariate component $f_i$ as the total bit-length of all its weights and biases, and the size of the full model $f$ as the sum of the sizes of all univariate components $f_i$ together with the bit-length of the global bias term $\beta_0$.

Although we present the formulation in the standard feedforward setting, this entails no loss of generality: any model with arbitrary connections can be converted into an equivalent feedforward network by inserting intermediate layers and adding zero-weight connections so that the computational graph becomes acyclic without changing its function. Moreover, ReLU networks can trivially express basic algebraic operations such as addition, scalar multiplication, and the rectifier $\max(0, \mathbf{x})$, and can therefore represent any composition of these operations. Consequently, restricting attention to feedforward ReLU architectures is fully general for our purposes.

## C PROOFS

### C.1 PROOF OF PROPOSITION 1

**Proposition 1.** *Given a NAM $f$, an input $\mathbf{x} \in \mathbb{R}^n$ and a perturbation radius $\epsilon_p \in \mathbb{R}_+$, let Alg. 2 return a total list order over the input features according to their importance. Then, the following holds: for any sufficient explanation $\mathcal{S}$ that includes feature $i$, and for any feature $j \notin \mathcal{S}$ such that $i \prec j$ in the list ordering, the set $\mathcal{S} \setminus \{i\} \cup \{j\}$ is also a sufficient explanation.*

*Proof.* We recall that we have assumed in Alg. 2 that $f(\mathbf{x})$ yields a positive prediction, i.e., it is classified as 1. Accordingly, the final list of bounds $[(\hat{\Delta l}_1, \hat{\Delta u}_1), (\hat{\Delta l}_2, \hat{\Delta u}_2), \ldots, (\hat{\Delta l}_n, \hat{\Delta u}_n)]$ is derived by taking the minimum possible value of each $f_i(\tilde{\mathbf{x}}_i)$ and computing lower and upper bounds for $f_i(\mathbf{x}_i) - f_i(\tilde{\mathbf{x}}_i)$. The proof we present applies symmetrically to the case where the prediction is negative: in that case, we instead take the maximum value of $f_i(\tilde{\mathbf{x}}_i)$ and bound $f_i(\tilde{\mathbf{x}}_i) - f_i(\mathbf{x}_i)$. We defer a detailed discussion of that case to later. Since we are assuming $f(\mathbf{x}) \geq 0$, the condition that $\mathcal{S}$ is a sufficient explanation with respect to $\langle f, \mathbf{x}, \epsilon_p \rangle$ means that:

$$\forall \tilde{\mathbf{x}} \in B_p^{\epsilon_p}(\mathbf{x}). \quad f(\mathbf{x}_{\mathcal{S}}; \tilde{\mathbf{x}}_{\bar{\mathcal{S}}}) \geq 0 \iff \\ \min_{\tilde{\mathbf{x}} \in B_p^{\epsilon_p}(\mathbf{x})} f(\mathbf{x}_{\mathcal{S}}; \tilde{\mathbf{x}}_{\bar{\mathcal{S}}}) \geq 0. \tag{4}$$

The value of $f(\mathbf{x}_{\mathcal{S}}; \tilde{\mathbf{x}}_{\bar{\mathcal{S}}})$ is obtained by fixing the features in $\mathcal{S}$ to $\mathbf{x}$ and perturbing the complementary features $\bar{\mathcal{S}}$ to values from $\tilde{\mathbf{x}}$. Owing to this construction, and to the additive form of the NAM $f$, which can be expressed as $f(\mathbf{x}) := \sum_{t \in [n]} f_t(\mathbf{x}_t)$, we can establish the following statements:

$$\sum_{t \in \bar{\mathcal{S}}} \hat{\Delta l}_t \leq (f_t(\mathbf{x}_t) - \min_{\tilde{\mathbf{x}} \in B_p^{\epsilon_p}(\mathbf{x})} \sum_{t \in \bar{\mathcal{S}}} f_t(\tilde{\mathbf{x}}_t)) \leq \sum_{t \in \bar{\mathcal{S}}} \hat{\Delta u}_t \iff$$

$$\sum_{t \in \bar{\mathcal{S}}} \hat{\Delta l}_t \leq \sum_{t \in \bar{\mathcal{S}}}(f_t(\mathbf{x}_t) - \min_{\tilde{\mathbf{x}} \in B_p^{\epsilon_p}(\mathbf{x})} f_t(\tilde{\mathbf{x}}_t)) + \sum_{t \in \mathcal{S}}(f_t(\mathbf{x}_t) - f_t(\mathbf{x}_t)) \leq \sum_{t \in \bar{\mathcal{S}}} \hat{\Delta u}_t \iff$$

$$\sum_{t \in \bar{\mathcal{S}}} \hat{\Delta l}_t \leq \sum_{t \in [n]} f(\mathbf{x}) - \sum_{t \in \mathcal{S}} f_t(\mathbf{x}_t) - \sum_{t \in \bar{\mathcal{S}}} \min_{\tilde{\mathbf{x}} \in B_p^{\epsilon_p}(\mathbf{x})} f(\tilde{\mathbf{x}}_t) \leq \sum_{t \in \bar{\mathcal{S}}} \hat{\Delta u}_t \iff \tag{5}$$

$$\sum_{t \in \bar{\mathcal{S}}} \hat{\Delta l}_t \leq (f(\mathbf{x}) - \min_{\tilde{\mathbf{x}} \in B_p^{\epsilon_p}(\mathbf{x})} f(\mathbf{x}_{\mathcal{S}}; \tilde{\mathbf{x}}_{\bar{\mathcal{S}}})) \leq \sum_{t \in \bar{\mathcal{S}}} \hat{\Delta u}_t.$$

Given our earlier assumption in Equation 4, we know that $\sum_{t \in \bar{\mathcal{S}}} \hat{\Delta u}_t \geq 0$. Now define $\mathcal{S}' := \mathcal{S} \cup \{j\} \setminus \{i\}$. By applying the same line of reasoning as before, we obtain:

$$\sum_{t \in \bar{\mathcal{S}}'} \hat{\Delta l}_t \leq (f(\mathbf{x}) - \min_{\tilde{\mathbf{x}} \in B_p^{\epsilon_p}(\mathbf{x})} f(\mathbf{x}_{\mathcal{S}'}; \tilde{\mathbf{x}}_{\bar{\mathcal{S}}'})) \leq \sum_{t \in \bar{\mathcal{S}}'} \hat{\Delta u}_t \iff$$

$$\sum_{t \in \bar{\mathcal{S}}} \hat{\Delta l}_t + \hat{\Delta l}_j - \hat{\Delta l}_i \leq (f(\mathbf{x}) - \min_{\tilde{\mathbf{x}} \in B_p^{\epsilon_p}(\mathbf{x})} f(\mathbf{x}_{\mathcal{S}'}; \tilde{\mathbf{x}}_{\bar{\mathcal{S}}'})) \leq \sum_{t \in \bar{\mathcal{S}}} \hat{\Delta u}_t + \hat{\Delta u}_j - \hat{\Delta u}_i. \tag{6}$$

Since we assume that $i \prec j$ in the ordering of $[(\hat{\Delta l}_1, \hat{\Delta u}_1), (\hat{\Delta l}_2, \hat{\Delta u}_2), \ldots, (\hat{\Delta l}_n, \hat{\Delta u}_n)]$ and that the bounds are *non-intersecting*, it follows that $\hat{\Delta l}_j - \hat{\Delta l}_i \geq 0$ and $\hat{\Delta u}_j - \hat{\Delta u}_i \geq 0$. Consequently, we obtain that $(f(\mathbf{x}) - \min_{\tilde{\mathbf{x}} \in B_p^{\epsilon_p}(\mathbf{x})} f(\mathbf{x}_{\mathcal{S}'}; \tilde{\mathbf{x}}_{\bar{\mathcal{S}}'}))$ is bounded both above and below by smaller values than $(f(\mathbf{x}) - \min_{\tilde{\mathbf{x}} \in B_p^{\epsilon_p}(\mathbf{x})} f(\mathbf{x}_{\mathcal{S}}; \tilde{\mathbf{x}}_{\bar{\mathcal{S}}}))$. This in turn implies that:

$$(f(\mathbf{x}) - \min_{\tilde{\mathbf{x}} \in B_p^{\epsilon_p}(\mathbf{x})} f(\mathbf{x}_{\mathcal{S}'}; \tilde{\mathbf{x}}_{\bar{\mathcal{S}}'})) - (f(\mathbf{x}) - \min_{\tilde{\mathbf{x}} \in B_p^{\epsilon_p}(\mathbf{x})} f(\mathbf{x}_{\mathcal{S}}; \tilde{\mathbf{x}}_{\bar{\mathcal{S}}})) \leq 0 \iff$$
$$\min_{\tilde{\mathbf{x}} \in B_p^{\epsilon_p}(\mathbf{x})} f(\mathbf{x}_{\mathcal{S}}; \tilde{\mathbf{x}}_{\bar{\mathcal{S}}}) - \min_{\tilde{\mathbf{x}} \in B_p^{\epsilon_p}(\mathbf{x})} f(\mathbf{x}_{\mathcal{S}'}; \tilde{\mathbf{x}}_{\bar{\mathcal{S}}'}) \leq 0. \tag{7}$$

Moreover, since $\min_{\tilde{\mathbf{x}} \in B_p^{\epsilon_p}(\mathbf{x})} f(\mathbf{x}_{\mathcal{S}}; \tilde{\mathbf{x}}_{\bar{\mathcal{S}}})$ is non-negative by Equation 4, it follows that:

$$\min_{\tilde{\mathbf{x}} \in B_p^{\epsilon_p}(\mathbf{x})} f(\mathbf{x}_{\mathcal{S}'}; \tilde{\mathbf{x}}_{\bar{\mathcal{S}}'}) \geq \min_{\tilde{\mathbf{x}} \in B_p^{\epsilon_p}(\mathbf{x})} f(\mathbf{x}_{\mathcal{S}}; \tilde{\mathbf{x}}_{\bar{\mathcal{S}}}) \geq 0 \implies$$
$$\forall \tilde{\mathbf{x}} \in B_p^{\epsilon_p}(\mathbf{x}). \quad f(\mathbf{x}_{\mathcal{S}'}; \tilde{\mathbf{x}}_{\bar{\mathcal{S}}'}) \geq 0. \tag{8}$$

which establishes that $\mathcal{S}'$ constitutes a sufficient explanation for $\langle f, \mathbf{x}, \epsilon_p \rangle$, thereby concluding this part of the proof.

We now turn to the symmetric case, where $f(\mathbf{x}) < 0$. In this setting, Alg. 2 is applied symmetrically by taking the *maximum* admissible value of each $f_i(\tilde{\mathbf{x}}_i)$ and deriving corresponding upper and lower bounds for $f_i(\tilde{\mathbf{x}}_i) - f_i(\mathbf{x}_i)$. $[(\hat{\Delta}l_1, \hat{\Delta}u_1), (\hat{\Delta}l_2, \hat{\Delta}u_2), \ldots, (\hat{\Delta}l_n, \hat{\Delta}u_n)]$ is now sorted in descending importance values, instead of ascending. Given the assumption that $f(\mathbf{x}) < 0$, the requirement that $\mathcal{S}$ constitutes a sufficient explanation with respect to $\langle f, \mathbf{x}, \epsilon_p \rangle$ can be expressed as:

$$\forall \tilde{\mathbf{x}} \in B_p^{\epsilon_p}(\mathbf{x}). \quad f(\mathbf{x}_{\mathcal{S}}; \tilde{\mathbf{x}}_{\bar{\mathcal{S}}}) < 0 \iff$$
$$\max_{\tilde{\mathbf{x}} \in B_p^{\epsilon_p}(\mathbf{x})} f(\mathbf{x}_{\mathcal{S}}; \tilde{\mathbf{x}}_{\bar{\mathcal{S}}}) < 0. \tag{9}$$

Analogous to the earlier case, leveraging the additive structure of the NAM $f$, which can be written as $f(\mathbf{x}) := \sum_{t \in [n]} f_t(\mathbf{x}_t)$, together with the definitions of $f(\mathbf{x}_{\mathcal{S}}; \tilde{\mathbf{x}}_{\bar{\mathcal{S}}})$ and of the bounds $\hat{\Delta}l_i$ and $\hat{\Delta}u_i$, we can derive the following chain of statements:

$$\sum_{t \in \bar{\mathcal{S}}} \hat{\Delta}l_t \leq \sum_{t \in \bar{\mathcal{S}}} (\max_{\tilde{\mathbf{x}} \in B_p^{\epsilon_p}(\mathbf{x})} f_t(\tilde{\mathbf{x}}_t) - f_t(\mathbf{x}_t)) \leq \sum_{t \in \bar{\mathcal{S}}} \hat{\Delta}u_t \iff$$
$$\sum_{t \in \bar{\mathcal{S}}} \hat{\Delta}l_t \leq \sum_{t \in \bar{\mathcal{S}}} (\max_{\tilde{\mathbf{x}} \in B_p^{\epsilon_p}(\mathbf{x})} f_t(\tilde{\mathbf{x}}_t) - f_t(\mathbf{x}_t)) + \sum_{t \in \mathcal{S}} (f_t(\mathbf{x}_t) - f_t(\mathbf{x}_t)) \leq \sum_{t \in \bar{\mathcal{S}}} \hat{\Delta}u_t \iff$$
$$\sum_{t \in \bar{\mathcal{S}}} \hat{\Delta}l_t \leq \sum_{t \in \bar{\mathcal{S}}} \max_{\tilde{\mathbf{x}} \in B_p^{\epsilon_p}(\mathbf{x})} f_t(\tilde{\mathbf{x}}_t) + \sum_{t \in \mathcal{S}} f_t(\mathbf{x}_t) - \sum_{t \in [n]} f_t(\mathbf{x}_t) \leq \sum_{t \in \bar{\mathcal{S}}} \hat{\Delta}u_t \iff$$
$$\sum_{t \in \bar{\mathcal{S}}} \hat{\Delta}l_t \leq (\max_{\tilde{\mathbf{x}} \in B_p^{\epsilon_p}(\mathbf{x})} f(\mathbf{x}_{\mathcal{S}}; \tilde{\mathbf{x}}_{\bar{\mathcal{S}}}) - f(\mathbf{x})) \leq \sum_{t \in \bar{\mathcal{S}}} \hat{\Delta}u_t. \tag{10}$$

Since we know that $\sum_{t \in \bar{\mathcal{S}}} \hat{\Delta}u_t \geq 0$, and by defining $\mathcal{S}' := \mathcal{S} \cup \{j\} \setminus \{i\}$ as before, we can now derive that:

$$\sum_{t \in \bar{\mathcal{S}}'} \hat{\Delta}l_t \leq (\max_{\tilde{\mathbf{x}} \in B_p^{\epsilon_p}(\mathbf{x})} f(\mathbf{x}_{\mathcal{S}'}; \tilde{\mathbf{x}}_{\bar{\mathcal{S}}'}) - f(\mathbf{x})) \leq \sum_{t \in \bar{\mathcal{S}}'} \hat{\Delta}u_t \iff$$
$$\sum_{t \in \bar{\mathcal{S}}} \hat{\Delta}l_t + \hat{\Delta}l_i - \hat{\Delta}l_j \leq (\max_{\tilde{\mathbf{x}} \in B_p^{\epsilon_p}(\mathbf{x})} f(\mathbf{x}_{\mathcal{S}'}; \tilde{\mathbf{x}}_{\bar{\mathcal{S}}'}) - f(\mathbf{x})) \leq \sum_{t \in \bar{\mathcal{S}}} \hat{\Delta}u_t + \hat{\Delta}u_i - \hat{\Delta}u_j. \tag{11}$$

As before, since we assume $i \prec j$ in the ordering of $[(\hat{\Delta}l_1, \hat{\Delta}u_1), (\hat{\Delta}l_2, \hat{\Delta}u_2), \ldots, (\hat{\Delta}l_n, \hat{\Delta}u_n)]$, and given that the bounds are non-intersecting, together with our assumption that this list is sorted by *decreasing* values, it follows that $\hat{\Delta}l_i - \hat{\Delta}l_j \geq 0$ and $\hat{\Delta}u_i - \hat{\Delta}u_j \geq 0$. Consequently, we obtain a different outcome: namely, $\max_{\tilde{\mathbf{x}} \in B_p^{\epsilon_p}(\mathbf{x})} (f(\mathbf{x}_{\mathcal{S}'}; \tilde{\mathbf{x}}_{\bar{\mathcal{S}}'}) - f(\mathbf{x}))$ is bounded above and below by strictly *larger* values than $\max_{\tilde{\mathbf{x}} \in B_p^{\epsilon_p}(\mathbf{x})} (f(\mathbf{x}_{\mathcal{S}}; \tilde{\mathbf{x}}_{\bar{\mathcal{S}}}) - f(\mathbf{x}))$. This in turn implies that:

$$\big(\max_{\tilde{\mathbf{x}}\in B_p^{\epsilon_p}(\mathbf{x})} f(\mathbf{x}_{\mathcal{S}'};\tilde{\mathbf{x}}_{\bar{\mathcal{S}}'}) - f(\mathbf{x})\big) - \big(\max_{\tilde{\mathbf{x}}\in B_p^{\epsilon_p}(\mathbf{x})} f(\mathbf{x}_{\mathcal{S}};\tilde{\mathbf{x}}_{\bar{\mathcal{S}}}) - f(\mathbf{x})\big) \le 0 \iff$$

$$\max_{\tilde{\mathbf{x}}\in B_p^{\epsilon_p}(\mathbf{x})} f(\mathbf{x}_{\mathcal{S}'};\tilde{\mathbf{x}}_{\bar{\mathcal{S}}'}) - \max_{\tilde{\mathbf{x}}\in B_p^{\epsilon_p}(\mathbf{x})} f(\mathbf{x}_{\mathcal{S}};\tilde{\mathbf{x}}_{\bar{\mathcal{S}}}) \le 0. \tag{12}$$

Moreover, since Equation 9 ensures that $\max_{\tilde{\mathbf{x}}\in B_p^{\epsilon_p}(\mathbf{x})} f(\mathbf{x}_{\mathcal{S}};\tilde{\mathbf{x}}_{\bar{\mathcal{S}}})$ is negative, we obtain:

$$\max_{\tilde{\mathbf{x}}\in B_p^{\epsilon_p}(\mathbf{x})} f(\mathbf{x}_{\mathcal{S}'};\tilde{\mathbf{x}}_{\bar{\mathcal{S}}'}) \le \max_{\tilde{\mathbf{x}}\in B_p^{\epsilon_p}(\mathbf{x})} f(\mathbf{x}_{\mathcal{S}};\tilde{\mathbf{x}}_{\bar{\mathcal{S}}}) < 0 \implies$$

$$\forall \tilde{\mathbf{x}}\in B_p^{\epsilon_p}(\mathbf{x}).\quad f(\mathbf{x}_{\mathcal{S}'};\tilde{\mathbf{x}}_{\bar{\mathcal{S}}'}) < 0. \tag{13}$$

This establishes that $\mathcal{S}'$ is a sufficient explanation for $\langle f, \mathbf{x}, \epsilon_p\rangle$. With this, the negative case for $f(\mathbf{x})$ is resolved, and together with the positive case, the proof is complete.

$\square$

## C.2 PROOF OF PROPOSITION 2

**Proposition 2.** *Given $\rho$ parallelized processors, Alg. 2, performs an overall number of $T_p(n) = \mathcal{O}\big(\big(\frac{n}{p}\big)\log\big(\max_{i\in[n]}(\frac{\beta_i-\alpha_i}{\xi_i})\big)\big) \xrightarrow{p\to n} \mathcal{O}\big(\log\big(\max_{i\in[n]}(\frac{\beta_i-\alpha_i}{\xi_i})\big)\big)$ calls to the verifier, each on a $f_i(\cdot)$ component, where $\xi_i := \min\{|\hat{\Delta}l_{i+1} - \hat{\Delta}u_i|, |\hat{\Delta}l_i - \hat{\Delta}u_{i-1}|\}$.*

*Proof.* The algorithm terminates once no two univariate functions $f_i$ and $f_j$ have overlapping bounds. Because the procedure relies on binary search, each phase divides the current interval into two. Initially, the gap between the upper and lower bounds for a feature $i$ is exactly $\alpha_i - f_i(\mathbf{x}_i)$. The precision achieved for feature $i$ is limited by the smaller of the two distances: either the distance to the bound of the feature directly above it in the ordering $(i+1)$ or the one directly below it $(i-1)$. Accordingly, we denote the overall precision for feature $i$ by $\xi_i$ as:

$$\xi_i := \min\{|\hat{\Delta}l_{i+1} - \hat{\Delta}u_i|, |\hat{\Delta}l_i - \hat{\Delta}u_{i-1}|\}. \tag{14}$$

Overall, given the binary-search procedure, where the interval is split at each iteration, we define the number of splits $k_i$ performed for a single feature $i$ as:

$$\frac{\beta_i - \alpha_i}{2^{k_i}} \le \xi_i \iff$$

$$k_i \le \mathcal{O}\big(\log(\frac{\beta_i - \alpha_i}{\xi_i})\big). \tag{15}$$

Consequently, the feature on which the maximum number of splits is carried out, denoted by $k_{\max}$, is:

$$k_{max} \le \mathcal{O}\Big(\max_{i\in[n]}\big(\log(\frac{\beta_i - \alpha_i}{\xi_i})\big)\Big). \tag{16}$$

Each feature $i \in [n]$ is therefore bounded by at most $k_{max}$ verification queries. Consequently, the total workload is upper bounded by $n \cdot k_{max}$, and when distributed across $\rho$ threads, this yields the following parallelized complexity result $T_p(n)$:

$$T_\rho(n) \le \mathcal{O}\big((\frac{n}{\rho}) \cdot k_{max}\big) \le \mathcal{O}\Big((\frac{n}{\rho}) \log\big(\max_{i\in[n]}(\frac{\beta_i - \alpha_i}{\xi_i})\big)\Big). \tag{17}$$

This completes the proof.

$\square$

$$T_\rho(n) \leq \mathcal{O}\big((\frac{n}{\rho}) \cdot k_{max}\big) \leq \mathcal{O}\Big(\big(\frac{n}{\rho}\big) \max \big( \log \big( \max_{i \in [n]} (\frac{\beta_i - \alpha_i}{\xi_i})\big), \xi'\big)\Big). \tag{18}$$

## C.3 Proof of Proposition 3

**Proposition 3.** *Given a NAM $f$, an input $\boldsymbol{x} \in \mathbb{R}^n$, and a perturbation radius $\epsilon_p \in \mathbb{R}_+$, Alg. 3 performs $\mathcal{O}(n)$ queries and returns a* cardinally-minimal *sufficient explanation. This stands in contrast to Alg. 1, which is only guaranteed to return a* subset minimal *sufficient explanation.*

*Proof.* We begin by noting that the algorithm proceeds iteratively, making $|n|$ calls to the query $\text{suff}(f, \mathbf{x}, \mathcal{S} \setminus \{i\}, \epsilon_p)$. Each such query can be encoded using a neural network verifier, which implies that the algorithm requires $\mathcal{O}(n)$ invocations in total. We now turn to proving that Alg. 3 indeed produces a cardinally-minimal sufficient explanation with respect to $\langle f, \mathbf{x}, \epsilon_p \rangle$. First, let us prove that Alg. 3 provides a valid sufficient explanation. This result is straightforward since the last condition that is checked is that: $\text{suff}(f, \mathbf{x}, \mathcal{S} \setminus \{i\}, \epsilon_p)$, and after this condition is met $\mathcal{S}$ is updated to be $\mathcal{S} \setminus \{i\}$ and is returned. Hence, by definition, the sufficiency of the returned subset is satisfied.

We will now demonstrate that the generated set $\mathcal{S}$ is a cardinally-minimal sufficient explanation with respect to $\langle f, \mathbf{x}, \epsilon_p \rangle$. Let $1 \leq \ell \leq n$ represent the last feature that was attempted to be removed from $\mathcal{S}$ in Alg. 3. Then, for $\mathcal{S}' := \mathcal{S} \setminus \{\ell\}$, it follows that: $\text{suff}(f, \mathbf{x}, \mathcal{S}', \epsilon_p)$ does not hold true, implying that $\mathcal{S}'$ is *not* a sufficient explanation for $\langle f, \mathbf{x}, \epsilon_p \rangle$. We begin by proving a first lemma that will help us proving our proposition:

**Lemma 1.** *Given a NAM $f$, let Alg. 2 return the sorted, non-intersecting list of pairs: $[(\hat{\Delta}l_1, \hat{\Delta}u_1), (\hat{\Delta}l_2, \hat{\Delta}u_2), \ldots, (\hat{\Delta}l_n, \hat{\Delta}u_n)]$. Then, the following holds: if $\mathcal{S}$ that denotes the top $|\mathcal{S}|$ features ordered by $[(\hat{\Delta}l_1, \hat{\Delta}u_1), (\hat{\Delta}l_2, \hat{\Delta}u_2), \ldots, (\hat{\Delta}l_n, \hat{\Delta}u_n)]$ is not a sufficient explanation concerning $\langle f, \boldsymbol{x}, \epsilon_p \rangle$, then any subset $\mathcal{S}' \subseteq [n]$ of size $|\mathcal{S}|$ is also not a sufficient explanation concerning $\langle f, \boldsymbol{x}, \epsilon_p \rangle$.*

*Proof.* We begin by noting that $\mathcal{S} \subseteq [n]$ is *not* a sufficient explanation with respect to $\langle f, \mathbf{x}, \epsilon_p \rangle$. Assume, for contradiction, that there exists some $\mathcal{S}' \neq \mathcal{S}$ of the same cardinality as $\mathcal{S}$ that *is* a sufficient explanation with respect to $\langle f, \mathbf{x}, \epsilon_p \rangle$. Since both $\mathcal{S}$ and $\mathcal{S}'$ have equal size, we can map each feature in $\mathcal{S}'$ with one in $\mathcal{S}$ according to their position in the ordering $[(\hat{\Delta}l_1, \hat{\Delta}u_1), (\hat{\Delta}l_2, \hat{\Delta}u_2), \ldots, (\hat{\Delta}l_n, \hat{\Delta}u_n)]$.

By definition, $\mathcal{S}$ consists of the top $|\mathcal{S}|$ features in this ordering. Consequently, under the mapping, each feature in $\mathcal{S}'$ is mapped to a feature of strictly higher or equal rank in $\mathcal{S}$. Now consider a sequence of replacements: at each step, replace a feature of $\mathcal{S}'$ with its corresponding equivalent or higher-ranked feature from $\mathcal{S}$. Prop. 1 ensures that each such replacement preserves sufficiency, since a "lower-ranked" feature is being swapped for a "higher-ranked" one. Iterating this process eventually transforms $\mathcal{S}'$ into $\mathcal{S}$, while preserving sufficiency throughout. Thus, $\mathcal{S}$ must itself be a sufficient explanation with respect to $\langle f, \mathbf{x}, \epsilon_p \rangle$ — contradicting the initial assumption that it is not. This completes the proof.

$\square$

From Lemma 1, since the features in $\mathcal{S}' \setminus \{\ell\}$ are the features with the highest $|\mathcal{S}'| = |\mathcal{S}| - 1$ orderings, it holds that any subset $\mathcal{S}'' \subseteq [n]$ of size $|\mathcal{S}'|$ is not a sufficient explanation. To conclude the remaining parts of our proof, we now will make use of another lemma:

**Lemma 2.** *Let there be some $f$, $\boldsymbol{x}$, and $\epsilon_p$. Then, if $\mathcal{S} \subseteq [n]$ is not a sufficient explanation concerning $\langle f, \boldsymbol{x}, \epsilon_p \rangle$, then any $\mathcal{S}' \subseteq \mathcal{S}$ is not a sufficient explanation w.r.t $\langle f, \boldsymbol{x}, \epsilon_p \rangle$.*

*Proof.* If $\mathcal{S} \subseteq [n]$ is not a sufficient explanation with respect to $\langle f, \mathbf{x}, \epsilon_p \rangle$, then:

$$\exists \mathbf{z} \in B_p^{\epsilon_p}(\mathbf{x}). \quad f(\mathbf{x}_{\mathcal{S}}; \mathbf{z}_{\bar{\mathcal{S}}}) \neq f(\mathbf{x}). \tag{19}$$

Assume, towards contradiction, that there exists some $\mathcal{S}' \subseteq \mathcal{S}$ which is a sufficient explanation. In other words:

$$\forall \tilde{\mathbf{x}} \in B_p^{\epsilon_p}(\mathbf{x}). \quad f(\mathbf{x}_{\mathcal{S}'}; \tilde{\mathbf{x}}_{\bar{\mathcal{S}'}}) = f(\mathbf{x}). \tag{20}$$

However, consider a vector $\mathbf{z}'$ obtained by fixing the features in $\mathcal{S}'$ to $\mathbf{x}$, the features in $\mathcal{S} \setminus \mathcal{S}'$ also to $\mathbf{x}$, and setting all remaining coordinates according to $\mathbf{z}$. By the earlier implication from Equation 20, we must then have that: $f(\mathbf{x}_{\mathcal{S}'}; \mathbf{z}'_{\bar{\mathcal{S}}'}) \neq f(\mathbf{x})$ and this contradicts the assumption that $\mathcal{S}'$ is sufficient (Equation 19).

$\square$

We now proceed with the remaining part of proving our proposition. Since we know that there is no explanation of size $|\mathcal{S}'| = |\mathcal{S}| - 1$ concerning $\langle f, \mathbf{x}, \epsilon_p \rangle$ from the previous part of the proof, we now can use the result in Lemma 2 to conclude that none of the subsets of these subsets of size $|\mathcal{S}'|$ is not a sufficient explanation too, which implies that there does not exist any explanation of size *lower or equal* to $|\mathcal{S}'| - 1$ which is a sufficient explanation of $\langle f, \mathbf{x}, \epsilon_p \rangle$, which proves that $\mathcal{S}$ is a cardinally-minimal sufficient explanation, hence concluding the proof.

$\square$

### C.4 PROOF OF PROPOSITION 4

**Proposition 4.** *Given a NAM $f$, an input $\boldsymbol{x} \in \mathbb{R}^n$, and a perturbation radius $\epsilon_p \in \mathbb{R}_+$, Alg. 3 performs $\mathcal{O}(\log(n))$ queries and returns a cardinally-minimal sufficient explanation.*

*Proof.* We will show that, given the ordering of features $[(\hat{\Delta l}_1, \hat{\Delta u}_1), (\hat{\Delta l}_2, \hat{\Delta u}_2), \ldots, (\hat{\Delta l}_n, \hat{\Delta u}_n)]$ there exists exactly one index $i \in [n]$ such that $\mathcal{S} = [i+1]$ is a sufficient explanation while $\mathcal{S}' = [i]$ is not. Moreover, this statement holds for any ordering. Our proof follows as a consequence of a lemma closely related to Lemma 2, which we restate and establish below:

**Lemma 3.** *Let there be some $f$, $\boldsymbol{x}$, and $\epsilon_p$. Then, if $\mathcal{S} \in [n]$ is a sufficient explanation concerning $\langle f, \boldsymbol{x}, \epsilon_p \rangle$, then any $\mathcal{S}'$ for which $\mathcal{S} \subseteq \mathcal{S}'$ is also a sufficient explanation w.r.t $\langle f, \boldsymbol{x}, \epsilon_p \rangle$.*

*Proof.* This follows directly from Lemma 2. Since $\mathcal{S}$ is known to be a sufficient explanation, assume for contradiction that there exists some $\mathcal{S}' \subseteq [n]$ with $\mathcal{S} \subseteq \mathcal{S}'$ such that $\mathcal{S}'$ is not a sufficient explanation. By Lemma 2, this would imply that no subset $\mathcal{S}'' \subseteq \mathcal{S}'$ could be a sufficient explanation — contradicting the fact that $\mathcal{S} \subseteq \mathcal{S}'$ is sufficient.

$\square$

To conclude the proof of the proposition, consider iterating over the features sequentially according to their ordering. We begin with the empty set $\emptyset$ and test whether it is a sufficient explanation with respect to $\langle f, \mathbf{x}, \epsilon_p \rangle$. If it is not, we proceed by adding features one at a time: first $\{1\}$, then $\{1, 2\}$, then $\{1, 2, 3\}$, and so forth. Eventually, we encounter some feature $i \in [n]$ such that $[i]$ is sufficient with respect to $\langle f, \mathbf{x}, \epsilon_p \rangle$. By Lemma 3, any $\mathcal{S}$ satisfying $[i] \subseteq \mathcal{S}$ is also sufficient. Hence, the unique transition from insufficiency to sufficiency occurs between $[i-1]$ and $[i]$, and there can be no later index $j > i$ for which $[j]$ reverts to being non-sufficient before becoming sufficient again.

Since we have already established this claim, it follows that the binary search in Alg. 4, which halts upon identifying the first feature $i$ where $[i]$ is sufficient but $[i-1]$ is not, will return the same subset as the iterative "naive" Alg. 3, which incrementally traverses features in a greedy manner and outputs $[i]$. Moreover, Prop. 3 shows that Alg. 3 always converges to a cardinally-minimal explanation. Consequently, Alg. 4 must also converge to this same cardinally-minimal explanation, but with only $\mathcal{O}(\log(n))$ sufficiency checks rather than $\mathcal{O}(n)$. This completes the proof.

$\square$

### C.5 COMPUTATIONAL COMPLEXITY RESULTS ON FINDING CARDINALLY-MINIMAL SUFFICIENT EXPLANATIONS IN NAMS

In this subsection, we study the nature of the computational complexity of obtaining cardinally minimal sufficient explanations for NAMs. For clarity of presentation, we begin by formally defining and naming the computational problem under consideration.

**Minimum Sufficient Explanation for NAMs (MSE-NAM).**
**Input.** A NAM $f : \mathbb{R}^n \to \{0, 1\}$, an input $\mathbf{x} \in \mathbb{R}^n$, and an $\epsilon \in \mathbb{Q}$ perturbation around $\mathbf{x}$.

**Output.** The smallest $k \in \mathbb{N}$ for which $\mathcal{S} \subseteq [n]$ is a sufficient explanation concerning $\langle f, \mathbf{x}, \epsilon \rangle$, such that $|\mathcal{S}| = k$.

The proof is divided into a membership component and a hardness component. We begin by proving membership.

**Claim 1.** *The MSE-NAM problem is in FP$^{NP}[\mathcal{O}(N)]$, where $N$ represents the model size encoding.*

*Proof.* We establish membership in FP$^{NP}[\mathcal{O}(N)]$ by demonstrating that the problem can be solved via an OptP$[\mathcal{O}(N)]$ computation followed by a polynomial-time post-processing step, which together place the problem in FP$^{NP}[\mathcal{O}(N)]$ (Krentel, 1986). We begin with the first part of the proof:

**Lemma 4.** *Consider a ReLU neural network $f \colon \mathbb{R}^n \to \mathbb{R}^n$, an input $\mathbf{x} \in \mathbb{R}^n$, and a radius $\epsilon \in \mathbb{Q}$. Define the $\epsilon$-ball around $\mathbf{x}$ as $B_\epsilon(\mathbf{x}) := \{\mathbf{z} \in \mathbb{R}^n \colon \mathbf{x}_i - \epsilon \leq \mathbf{z}_i \leq \mathbf{x}_i + \epsilon\}$ for all $i \in [n]$. Then, the problem of computing $\max_{\mathbf{z} \in B_\epsilon(\mathbf{x})}(f(\mathbf{z})_1 + \cdots + f(\mathbf{z})_n)$ lies in OptP$[\mathcal{O}(N)]$.*

*Proof.* We prove this lemma by expressing our problem as a special case of the *neural network reachability* problem introduced by Sälzer & Lange (2021). In their formulation, a broad class of input specifications $\psi_{\text{in}}$ and output specifications $\psi_{\text{out}}$ is supported. For our purposes, we restrict to a particularly simple instance of their framework: each input feature $\mathbf{x}_i$ is constrained by rational lower and upper bounds $l_i, u_i \in \mathbb{Q}$ within $\psi_{\text{in}}$, and each output feature $\mathbf{y}_i$ is constrained by the same type of bounds within $\psi_{\text{out}}$. That is, we only consider specifications of the form $l_i \leq \mathbf{x}_i \leq u_i$ and $l_i \leq \mathbf{y}_i \leq u_i$, which are directly supported in the formulation of Sälzer & Lange (2021). We now proceed to formally define the neural network reachability problem.

**Neural Network Reachability (NNReach)** (Sälzer & Lange, 2021).
**Input.** A neural network model $f \colon \mathbb{R}^n \to \mathbb{R}^d$ with ReLU activations, an input specification $\psi_{\text{in}}(\mathbf{x}_1, \ldots, \mathbf{x}_n)$ and an output specification $\psi_{\text{out}}(\mathbf{y}_1, \ldots, \mathbf{y}_d)$.
**Output.** *Yes* if there is an $\mathbf{x} \in \mathbb{R}^n$ such that $\psi_{\text{in}}(\mathbf{x})$ and $\psi_{\text{out}}(f(\mathbf{x}))$ are true.

Sälzer & Lange (2021) showed that the NNReach problem lies in NP. Since input assignments cannot be guessed directly as witnesses in a continuous domain, their approach instead guesses the *activation pattern* (active/inactive) of every neuron. Once this activation-status is fixed, the network becomes a purely linear system, and the reachability question reduces to a linear program (LP) over all input variables $\mathbf{x}_1, \ldots, \mathbf{x}_n$, output variables $\mathbf{y}_1, \ldots, \mathbf{y}_d$, and additional auxiliary variables corresponding to neuron operations. This LP can then be solved in polynomial time, establishing NP membership.

We now will describe the NNReach problem in our specific context. We now have a model $f \colon \mathbb{R}^n \to \mathbb{R}^n$ (meaning the output vector $\mathbf{y}$ is of size $n$). We take the input specification $\psi_{\text{in}}$ to be set to be $\mathbf{x}_i - \epsilon \leq \mathbf{x}_i \leq \mathbf{x}_i + \epsilon$ for each $\mathbf{x}_i$ and taking no output specification over $\mathbf{y}$. We know via the result of (Sälzer & Lange, 2021) that this problem is in NP. We now can define the mapping function $G(X, Y)$ (within our definition of the complexity class OptP in Appendix B) such that $Y \in \{0, 1\}^{p(|X|)}$ is the witness assignment used by (Sälzer & Lange, 2021) to show membership in NP for the NNReach problem (the activation status of each ReLU), $\{0, 1\}^{p(|X|)}$ denotes the set of all 0-1 assignments to active/in-active neurons within the linear program, and the function $G$ defines the same linear program as in (Sälzer & Lange, 2021) while solving the task of *maximizing the value of $\mathbf{y}_1 + \mathbf{y}_2 + \ldots \mathbf{y}_n$ (the sum of all output values)*. The result for a single witness will give us this maximal value in the $\epsilon$-ball region around $\mathbf{x}$, for *one specific* linear region in the domain determined by the witness guess of activation statues $Y$. However, by taking

$$F(X) = \max_{Y \in \{0,1\}^{p(|X|)}} G(X, Y), \tag{21}$$

$F(X)$ will give us the maximum value out of *all* of the aforementioned maximal values in the $\epsilon$-ball region around $\mathbf{x}$ that are determined by guessing activation statues of neurons. This, together, implies that the $F(X) = \max_{\mathbf{z} \in B_\epsilon(\mathbf{x})}(f(\mathbf{z})_1 + f(\mathbf{z})_2 + \ldots + f(\mathbf{z})_n)$, which shows that this problem is in OptP. Now, since this optimal is a solution to some linear program defined over the paramters of the model — we know that it is finite and bounded by the precision of the program's parameters. Suppose each weight and bias in every univariate model $f_i$ is encoded as a rational number $\frac{p_j}{q_j}$ with $q_j, p_j \in \mathbb{N}$, given in binary. Let $p'$ and $q'$ denote the maximum numerator and denominator of weights/biases appearing anywhere in the overall model $f$, and define $d := \max(|p'|, |q'|)$. A naïve upper bound on the magnitude of the optimal LP solution is $\mathcal{O}(d^m)$, where $m$ is the number of

weights and biases in $f$, which in turn can be encoded using $\mathcal{O}(m \cdot \log(d))$ bits, and is particularly linear in $N$. This shows that this problem is particularly in $\text{OptP}[\mathcal{O}(N)]$.

$\square$

After establishing that the optimization step lies in $\text{OptP}[\mathcal{O}(N)]$, we now complete the membership proof for $\text{FP}^{\text{NP}}[\mathcal{O}(N)]$ for our original problem by describing the required polynomial post-processing. The first observation is that the result proven for any ReLU-activated neural network $f: \mathbb{R}^n \to \mathbb{R}^n$ (Lemma 4) also applies to networks of the form $\beta_o + f_1(\mathbf{x}_1) + \ldots + f_n(\mathbf{x}_n)$, i.e., the inherent structure of NAM models where each $f_i$ is itself a univariate ReLU network. This follows because such a sum is again a neural network under the same formalization. In this structure, maximizing or minimizing the full expression $\beta_o + f_1(\mathbf{x}_1) + \ldots + f_n(\mathbf{x}_n)$ reduces to maximizing or minimizing each univariate component $f_i(\mathbf{x}_i)$ *independently*. After carrying out this procedure to maximize each univariate component, this yields a total ordering over the features — equivalently, all bounds in Alg. 2 "collapse" to discrete singular points — allowing us to apply either Alg. 3 or Alg. 4 to compute a cardinally-minimal sufficient explanation (as proven in Prop. 3) and return its size $k$. Since the entire procedure consists of an $\text{OptP}[\mathcal{O}(N)]$ computation followed by a polynomial-time post-processing step, the overall problem belongs to $\text{FP}^{\text{NP}}[\mathcal{O}(N)]$. This concludes the membership proof.

$\square$

**Lemma 5.** *The MSE-NAM problem is OptP$[\mathcal{O}(\log(N))]$-Hard under metric reductions.*

*Proof.* To prove hardness, our main reduction will be from the Maximum satisfiability (MaxSAT) problem, which is known to be $\text{OptP}[\mathcal{O}(\log(N))]$-complete (Krentel, 1986). Before introducing this problem formally, we first recall the relevant background notion of CNF-Satisfiability (SAT):

Consider a set of Boolean variables $\mathbf{X}_1, \ldots, \mathbf{X}_n \in \{0, 1\}$. Let $m \in \mathbb{N}$ denote the number of clauses. For each $1 \leq i \leq m$, we define the clause $C_i$ as follows:

$$C_i := \mathbf{X}_{i,1} \vee \mathbf{X}_{i,2} \vee \ldots \mathbf{X}_{i,k_i}, \tag{22}$$

where each literal $i, j$ is an element of $\mathbf{X}_{i,j} \in \{\mathbf{X}_1, \ldots, \mathbf{X}_n, \neg\mathbf{X}_1, \ldots, \neg\mathbf{X}_n\}$ and $k_i \in \mathbb{N}$ denotes the number of literals appearing in clause $C_i$. The full CNF formula $\phi$ is then given by:

$$\phi := C_1 \wedge C_2 \wedge \ldots \wedge C_m. \tag{23}$$

For instance, take the following CNF formula $\phi$:

$$\phi := (\mathbf{X}_{1,1} \vee \ldots \vee \mathbf{X}_{1,k_1}) \wedge (\mathbf{X}_{2,1} \vee \ldots \vee \mathbf{X}_{2,k_2}) \wedge \ldots \wedge (\mathbf{X}_{m,k_1} \vee \ldots \vee \mathbf{X}_{m,k_m}) = \\ (\mathbf{X}_7 \vee \neg\mathbf{X}_2 \vee \mathbf{X}_3 \vee \mathbf{X}_5) \wedge (\mathbf{X}_2) \wedge (\mathbf{X}_3 \vee \neg\mathbf{X}_4). \tag{24}$$

We say that a clause $C_i$, or the entire formula $\phi$, is *satisfiable* if there exists a Boolean assignment $\alpha \in \{0, 1\}^n$ to the variables $\{\mathbf{X}_1, \ldots, \mathbf{X}_n\}$ that makes it evaluate to true. For instance, in the example above, assigning $\mathbf{X}_5 = 1$, $\mathbf{X}_2 = 1$, and $\mathbf{X}_4 = 0$ satisfies all three clauses $C_1$, $C_2$, and $C_3$, and therefore satisfies $\phi$ as well. This leads us to the classical definition of the *SAT* problem:

**CNF Satisfiability (SAT).**
**Input.** A Boolean formula $\phi$ in CNF.
**Output.** *Yes*, if $\phi$ is satisfiable, and *No*, otherwise.

In our setting, we also rely on the optimization versions of these problems, specifically the *MaxSAT* problem (Li & Manya, 2009), which asks for an assignment that maximizes the number of satisfied clauses, or equivalently:

**Maximum SAT (MaxSAT).**
**Input.** A Boolean formula $\phi$ in CNF.
**Output.** The maximum number of simultaneously satisfied clauses, i.e.,

$$\max_{\alpha \in \{0,1\}^n} \sum_{i=1}^{m} \mathbf{1}_{\{C_i \text{ is satisfiable by } \alpha\}}. \tag{25}$$

Before presenting the proof, we first formalize several auxiliary problems that will be needed. We begin by recalling the *Neural Network Reachability* (NNReach) problem of Sälzer & Lange (2021),

which we previously used for establishing membership. That work shows that NNReach is NP-Hard. Moreover, it proves that hardness persists even for networks with a *single* output and under very simple input specifications, namely, when each input feature $\mathbf{x}_i$ is constrained only by the bounds $0 \leq \mathbf{x}_i \leq 1$ (Theorem 3 in their paper). Importantly, all reductions in (Sälzer & Lange, 2021) proceed by first showing hardness over a binary input domain $\{0,1\}^n$, and subsequently extending the construction to the continuous domain $[0,1]^n$. As a result, NP-hardness holds in both settings. Building on this, we frame the restricted version of the neural network reachability problem considered in their work as *Restricted Neural Network Reachability (R-NNReach)*, which was established to be NP-Hard in Sälzer & Lange (2021).

**Restricted Neural Network Reachability (R-NNReach).**
**Input.** A neural network $f\colon \{0,1\}^n \to \mathbb{R}$, an input specification $\psi_{\text{in}}(\mathbf{x}_1, \ldots, \mathbf{x}_n)$ set to: $\psi_{\text{in}} := \bigwedge_{i=1}^{n} \mathbf{x}_i \geq 0 \wedge \mathbf{x}_i \leq 1$ and an output specification $\psi_{\text{out}}(\mathbf{y})$, set to: $\psi_{\text{out}} := \mathbf{y} = r$ for some $r \in \mathbb{Q}$.
**Output.** *Yes* if there is an $\mathbf{x} \in \mathbb{R}^n$ such that $\psi_{\text{in}}(\mathbf{x})$ and $\psi_{\text{out}}(f(\mathbf{x}))$ are true.

We will also introduce an additional variant of this problem that will support our maximization-related proofs:

**Maximum Restricted Neural Network Reachability (MR-NNReach).**
**Input.** A neural network $f\colon \{0,1\}^n \to \mathbb{R}$, an input specification $\psi_{\text{in}}(\mathbf{x}_1, \ldots, \mathbf{x}_n)$ set to: $\psi_{\text{in}} := \bigwedge_{i=1}^{n} \mathbf{x}_i \geq 0 \wedge \mathbf{x}_i \leq 1$.
**Output.** The maximal value of $f(\mathbf{x})$ for which $\psi_{\text{in}}(\mathbf{x})$ is true.

We will break down the rest of the proof into four central parts:

1. **Lemma 6** demonstrates how the NNReach problem, specifically its restricted variant, R-NNReach, which was originally obtained via a reduction from CNF-SAT (Sälzer & Lange, 2021; Katz et al., 2017), can be extended to support a reduction from the *Maximum* SAT problem. In particular, the lemma establishes that MaxSAT admits a polynomial-time reduction to MR-NNReach.

2. **Lemma 7** demonstrates that by incorporating an additional technical gadget, similar to the construction in (Sälzer & Lange, 2021) but requiring a non-trivial technical adaptation, we can extend the hardness result to the *continuous* domain. In particular, this shows that the reduction continues to hold when considering a function $f\colon [0,1]^n \to \mathbb{R}$.

3. **Lemma 8** shows that a ReLU network can implement a binary decomposition procedure, and that this construction can be layered on top of our model. This, in turn, demonstrates that the reduction from MaxSAT to MR-NNReach continues to hold even for $f_i$ that is a *univariate* network of the form $f_i\colon \{0, 1, \ldots, 2^n-1\} \to \mathbb{R}$. **Lemma 9** then further extends this reduction to the fully continuous setting, i.e., to functions of the form $f_i\colon \mathbb{R} \to \mathbb{R}$.

4. **Lemma 10** provides a reduction from MaxSAT to MSE-NAM by constructing a NAM containing $m^2$ univariate copies of the formula $\phi$ that are constructed via Lemma 9 (where $m$ is the number of clauses in the original CNF). This construction allows us to recover the exact MaxSAT optimum, thereby establishing that the problem is OptP[$\mathcal{O}(\log N)$]-hard.

**Lemma 6.** *The MaxSAT problem can be reduced in polynomial time to MR-NNReach.*

*Proof.* We begin by revisiting the NP-hardness proof from (Sälzer & Lange, 2021), which provides a technical correction to the original argument in (Katz et al., 2017). This correction addresses subtleties that arise when extending the reduction from the Boolean domain to the continuous setting — a point we will return to later. For the moment, ignoring this distinction, the two reductions are effectively identical. We present this construction first. The core idea is to reduce a CNF formula $\phi$ with $m$ clauses to a ReLU-activated neural network $f\colon \{0,1\}^n \to \mathbb{R}$, which can be formalized as follows:

$$y := m - \sum_{i=1}^{m} \max(0, 1 - \sum_{j=0}^{k_i} f_i(\mathbf{x}_{ij})) \tag{26}$$

where $f_j(\mathbf{x}_{j,k}) = \mathbf{X}_{jk}$ if $\mathbf{X}_{j,k}$ appears positively in $C_j$, and $f_j(\mathbf{x}_{j,k}) = 1 - \mathbf{X}_{j,k}$ if it appears negatively. The corresponding neural network can then be constructed in a direct manner, since both linear transformations and $\max(0, \cdot)$ operations are readily expressible using ReLU units. This

reduction was explicitly established in both (Katz et al., 2017) and (Sälzer & Lange, 2021). As shown therein, it follows immediately that for any Boolean assignment $\alpha \in \{0,1\}^n$ to $\phi$, we have:

$$\max(0, 1 - \sum_{j=0}^{k_i} f_i(\mathbf{x}_{ij})) = 0 \iff C_i \text{ is satisfied by } \alpha. \tag{27}$$

From this point, it is straightforward to show that imposing an output constraint of $\psi_{\text{out}} := y = m$ holds if and only if every clause in $\phi$ is satisfied. From here, it is straightforward to prove that when setting an output constraint $\psi_{\text{out}}$ of $y = m$, it holds that the formula is satisfied if and only if all clauses in $\phi$ are satisfied. Moreover, observe that in this construction the output value $y$ already encodes the number of satisfied clauses. Consequently, the same reduction immediately yields a polynomial-time reduction from *MaxSAT* to the *MR-NNReach* problem.

$\square$

**Lemma 7.** *The MaxSAT problem can be reduced in polynomial time to MR-NNReach when defined over a neural network of the form $f : [0, 1]^n \to \mathbb{R}$.*

*Proof.* We now shift the reduction from the binary input domain $\{0,1\}^n$ to the continuous region $[0,1]^n$. To do so, we employ a technique analogous to the Bool* gadgets introduced in (Sälzer & Lange, 2021). We begin by briefly recalling the role of Bool* gadgets in the standard CNF-SAT reduction, specifically, how they allow one to migrate a hardness proof from a discrete Boolean domain to a continuous one. We then describe how this construction can be adapted to our setting.

In the standard reduction from SAT to the R-NN-Reach problem, the authors of (Sälzer & Lange, 2021) propose the following construction.

$$y := \sum_{i=1}^{n}(\max(0, \frac{1}{2} - \mathbf{x}_i) + \max(0, \mathbf{x}_i - \frac{1}{2})) - \sum_{i=1}^{m} \max(0, 1 - \sum_{j=0}^{k_i} f_i(\mathbf{x}_{ij})) + (m - \frac{n}{2}) \tag{28}$$

The term $+m - \frac{n}{2}$ is omitted in their presentation, but including it here does not affect the correctness of the reduction and will later help streamline our exposition. As demonstrated in (Sälzer & Lange, 2021), the expression $\mathbf{z}_i = \max(0, \frac{1}{2} - \mathbf{x}_i) + \max(0, \mathbf{x}_i - \frac{1}{2})$ evaluates to:

$$\mathbf{z}_i = \begin{cases} \frac{1}{2} - \mathbf{x}_i & \text{if } \mathbf{x}_i < \frac{1}{2}, \\ \mathbf{x}_i - \frac{1}{2} & \text{otherwise} \end{cases} \tag{29}$$

Thus, we have that $\max(0, \frac{1}{2} - \mathbf{x}_i) + \max(0, \mathbf{x}_i - \frac{1}{2}) = 0$ if and only if $\mathbf{x}_i \in \{0, 1\}$. This gadget therefore forces each binary feature to take a value of either $0$ or $1$. In practice, (Sälzer & Lange, 2021) implement this in their reduction by imposing an output constraint $\psi_{\text{out}}$ requiring $\mathbf{y} = m$. This is because the expression:

$$-\sum_{i=1}^{m} \max(0, 1 - \sum_{j=0}^{k_i} f_i(\mathbf{x}_{ij})),$$

which we refer to as the *"maximal-clauses optimization term"*, is always non-positive. Consequently, in order to enforce $y = m$, it must hold that the following term:

$$\sum_{i=1}^{n}(\max(0, \frac{1}{2} - \mathbf{x}_i) + \max(0, \mathbf{x}_i - \frac{1}{2})),$$

which we refer to as the *"binary-value regularizer term"*, always evaluates to exactly $\frac{n}{2}$. As explained earlier, this occurs *if and only if* every $\mathbf{x}_i$ takes a strictly binary value in $\{0, 1\}$.

We now aim to apply an analogous transition, from binary inputs to continuous domains, in order to reduce *MaxSAT* to our problem. In the MaxSAT case, note that there are *no output constraints*; the objective is simply to maximize the number of satisfied clauses. Intuitively, we want to maximize the maximal-clause optimizer term while simultaneously enforcing that the binary-value regularizer

term remains exactly $\frac{n}{2}$, which, as before, holds if and only if all variables $\mathbf{x}_i$ take values strictly in $\{0, 1\}$.

However, when using the construction of (Sälzer & Lange, 2021) described in Equation 30, jointly maximizing all components may, in some cases, prioritize increasing the maximal-clause optimizer term at the expense of the binary regularizer. This can cause the input variables to drift away from strictly binary values. To prevent this, we need to amplify the "weight" of the binary regularizer so that maximizing it is always strictly preferable. One way to achieve this is by maximizing a term of the following form:

$$y := C \cdot \sum_{i=1}^{n} (\max(0, \frac{1}{2} - \mathbf{x}_i) + \max(0, \mathbf{x}_i - \frac{1}{2})) - \sum_{i=1}^{m} \max(0, 1 - \sum_{j=0}^{k_i} f_i(\mathbf{x}_{ij})) + (m - \frac{C \cdot n}{2}) \quad (30)$$

We can now set $C := |\phi| + 1$, noting that the term below can still be implemented by a neural network simply by adjusting the appropriate weight. With this choice of $C$, we obtain the following property for $\mathbf{y}$:

**Claim 2.** *The maximal value of $\mathbf{y}$ when $\mathbf{x}$ is taken from $[0, 1]^n$ is equal to the maximal value of $\mathbf{y}$ when $\mathbf{x}$ is taken from $\{0, 1\}^n$.*

*Proof.* Assume, for the sake of contradiction, that the maximum attainable value of $y$ is achieved at some input $\mathbf{x}$ for which at least one coordinate satisfies $\mathbf{x}_i \in (0, 1)$. Suppose in particular that $\mathbf{x}_i \geq \frac{1}{2}$. Then, by Equation 29, we have:

$$\max(0, \frac{1}{2} - \mathbf{x}_i) + \max(0, \mathbf{x}_i - \frac{1}{2}) = \mathbf{x}_i - \frac{1}{2} \quad (31)$$

We now show that raising the value of $\mathbf{x}_i$ to exactly 1 strictly increases the value of $\mathbf{y}$. Suppose the current value is $\mathbf{x}_i = 1 - \epsilon$. Increasing it to 1 increases the binary value regularizer term by $\epsilon \cdot C = \epsilon \cdot (|\phi| + 1)$. Importantly, regardless of how this change affects the maximal clause optimizer term, even if it leads to a decrease in $y$, that decrease cannot exceed $(|\phi| + 1) \cdot \epsilon$, since this term contains strictly fewer than $|\phi| + 1$ summands. Consequently, the net effect is that $y$ increases, implying that the current value of $y$ was not maximal.

Now suppose that $\mathbf{x}_i < \frac{1}{2}$. In this case, Equation 29 implies that:

$$\max(0, \frac{1}{2} - \mathbf{x}_i) + \max(0, \mathbf{x}_i - \frac{1}{2}) = \frac{1}{2} - \mathbf{x}_i \quad (32)$$

We now show that decreasing the value of $\mathbf{x}_i$ to exactly 0 increases the value of $y$. Let the current value be $\mathbf{x}_i := \epsilon$. Reducing it to 0 increases the binary value regularizer term by $\epsilon \cdot C = \epsilon \cdot (|\phi| + 1)$. As argued earlier, although this change may affect the maximal clause optimizer term — and may even decrease $y$ — the decrease is bounded above by $(|\phi| + 1) \cdot \epsilon$, since this term contains strictly fewer than $|\phi| + 1$ components. Consequently, the net value of $y$ must increase, contradicting maximality. $\square$

**Lemma 8.** *Let there be some input region: $D_n := [0, 2^n - 1] \cap \mathbb{Z}$. It is possible to construct in polynomial time a ReLU neural network $f \colon D_n \to \{0, 1\}^n$ with polynomial size such that $f(x) = bin(x)$, where $bin(x)$ is the binary decomposition vector of $x$.*

*Proof.* We begin by mapping a single discrete value $x \in \{0, 1, \ldots, 2^n - 1\}$ to its corresponding binary representation using a polynomial-size ReLU network. This step performs the classic binary decomposition procedure, which is outlined in Algorithm 5.

We construct the neural network to directly implement the mathematical operations of the algorithm. Starting from the input integer $x$ provided to $f$, the network proceeds through $n$ iterations, where each iteration consists of three sequential layers. At iteration $i$, the values $r^i$, $q^i$, and $b^i$ encode the corresponding intermediate values maintained by the algorithm. The value $b^i$ at each iteration is passed through a residual connection to the $i$-th component $\mathbf{o}_i$ of the output layer vector $\mathbf{o}$. Formally,

---

**Algorithm 5** Binary Decomposition

---

**Input:** integer $x$, number of bits $n$ with $0 \leq x < 2^n$

1: $r \leftarrow x$
2: **for each** $i = 0$ to $n - 1$ **do**
3:     $q \leftarrow r/2^{n-1-i}$
4:     **if** $q \geq 1$ **then**
5:         $\mathbf{b}_i \leftarrow 1$
6:         $r \leftarrow r - 2^{n-1-i}$
7:     **else**
8:         $\mathbf{b}_i \leftarrow 0$
9:     **end if**
10: **end for each**
11: **return** $(\mathbf{b}_0, \ldots, \mathbf{b}_{n-1})$

---

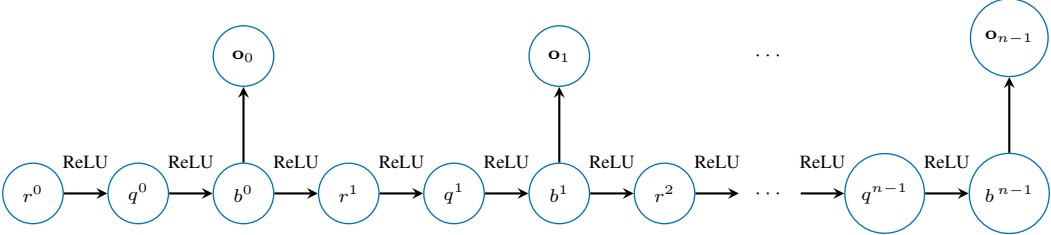

Figure 4: The computation of the binary decomposition defined in Equation 33, with input $\mathbf{x}$ and the output vector $\mathbf{o}_0, \ldots, \mathbf{o}_{n-1}$ representing the binary decomposition.

the construction is defined as follows:

$$
\begin{aligned}
q^i &= \max(0, \frac{r^i}{2^{n-1-i}} - 1), \\
b^i &= \max(0, 2^n \cdot q^i) - \max(0, (2^n \cdot q^i) - 1), \\
r^{i+1} &= r^i - 2^{n-1-i} \cdot \max(0, b^i), \\
\mathbf{o}_i &= b^i,
\end{aligned}
\tag{33}
$$

where $r^0 = x$ is the input provided to $f$. We illustrate this construction in Fig. 4. Although this construction is not optimized, it still has polynomial size and can be implemented using ReLU activations. In particular, all required operations — exact ReLU constraints, addition, multiplication, and subtraction — can be realized through standard neuron and weight configurations.

By induction, the neural network precisely simulates the algorithm. At iteration $i = 0$, the model computes $\frac{\mathbf{r}^0}{2^{n-1}} = \frac{x}{2^{n-1}}$. If $x \geq 2^{n-1}$, then $q^i$ receives a positive value in the range $\left[\frac{1}{2^n}, 1\right]$ (the lower bound follows from the assumption that $x$ is a *discrete* value in $\{0, 1, \ldots, 2^{n-1}\}$). If $x < 2^{n-1}$, then $q^i$ is set to 0. When $q^i = 0$, the corresponding bit $b^i$ is also set to 0 and mapped directly to the output $\mathbf{o}_0$. When $q^i$ is positive — necessarily within $\left[\frac{1}{2^n}, 1\right]$ due to discreteness — $b^i$ is set to 1. The value of $r^1$ is then updated exactly as prescribed by the algorithm, and the same process continues recursively for each iteration. All steps mirror the standard binary decomposition algorithm, and their correctness follows immediately from that correspondence. The only modification concerns the computation of $b^i$, which remains valid as long as the $q^i$ values stay above $\frac{1}{2^n}$. This condition always holds because the input is assumed to be discrete in the range $\{0, \ldots, 2^{n-1}\}$, ensuring that all values $b^i$ and $r^i$ remain discrete throughout the computation. Consequently, the final outputs $\mathbf{o}_0, \ldots, \mathbf{o}_{n-1}$ correctly represent the binary decomposition of $x$.

$\square$

**Claim 3.** *It is possible to construct in polynomial time a ReLU neural network $f \colon [0,1] \to \mathbb{R}^n$ with polynomial size such that $\{0,1\}^n \subseteq f([0,1]) \subseteq [0,1]^n$, where we denote $f([0,1]) := \{f(x) \mid x \in [0,1]\}$.*

*Proof.* We construct the neural network $f$ as follows. First, we apply a linear transformation to the input layer $x$ that multiplies it by $2^n - 1$. This scales the input so that, in the first hidden layer, we obtain a single neuron $\mathbf{y}$ whose range is precisely $[0, 2^n - 1]$. Next, we feed this neuron $y$ into the construction from Lemma 8. That lemma establishes that every *discrete* value in the domain $\{0, 1, \ldots, 2^n - 1\}$ is mapped to its corresponding $n$-bit binary representation. Since these binary representations cover the entire space $\{0, 1\}^n$, the image of our network $f$, when restricted to inputs where $y \in \{0, 1, \ldots, 2^n - 1\}$, already contains all binary vectors. Moreover, because the actual domain of $y$ is the full interval $[0, 2^n - 1]$ — which strictly contains all discrete values $\{0, 1, \ldots, 2^n - 1\}$ — it follows that $\{0, 1\}^n$ remains within the image of the overall model $f$.

For the second part of the proof, namely, establishing that $f([0, 1]) \subseteq [0, 1]^n$, we observe that in the construction of Lemma 8, each output neuron $\mathbf{o}_i$ is fed the value of $b^i = \max(0, 2^n \cdot q^i) - \max(0, (2^n \cdot q^i) - 1)$. By construction, this expression always evaluates to a value in the interval $[0, 1]$ for any real-valued $q^i$. Consequently, every output coordinate lies in $[0, 1]$, completing the proof.

$\square$

**Lemma 9.** *The MaxSAT problem can be reduced in polynomial time to MR-NNReach when defined over a neural network of the form $f : \mathbb{R} \to \mathbb{R}$.*

*Proof.* In this proof, we show that the reduction from MaxSAT to MR-NNReach established for neural networks of the form $f : [0, 1]^n \to \mathbb{R}$ (Lemma 7) remains valid even when the target model has the form $f : \mathbb{R} \to \mathbb{R}$. Since the MR-NNReach problem includes input constraints $\psi_{\text{in}}$ requiring each input variable $\mathbf{x}_i$ to satisfy $0 \leq \mathbf{x}_i \leq 1$, it suffices to consider models with domain $[0, 1]$. Using Claim 3, we construct a neural network $f : [0, 1] \to \mathbb{R}^n$ whose image satisfies $\{0, 1\}^n \subseteq f([0, 1]) \subseteq [0, 1]^n$. Next, we apply the construction from Lemma 7 to obtain a model $f' : [0, 1]^n \to \mathbb{R}$ whose maximum value over $[0, 1]^n$ corresponds exactly to the optimal value of the MaxSAT instance. Importantly, that reduction explicitly enforces that any maximizer of $f'(\mathbf{x})$ must satisfy $\mathbf{x} \in \{0, 1\}^n$, even though the domain is $[0, 1]^n$. We now define $g := f' \circ f$, and for clarity denote the components by $g_f := f$ and $g_{f'} := f'$, so that $g = g_{f'} \circ g_f$. The constraints introduced in Lemma 7 still apply to the inner component $g_{f'}$, ensuring that its maximal value is attained only on Boolean inputs. Moreover, Claim 3 guarantees that the range of $g_f$ includes all Boolean vectors $\{0, 1\}^n$ and excludes any point outside $[0, 1]^n$. Therefore, the maximal value attainable by $g_{f'}$ when composed with $g_f$ is exactly the same as the maximal value attainable by $f'$ over $[0, 1]^n$. Consequently, the overall reduction continues to hold for neural networks of the form $f : \mathbb{R} \to \mathbb{R}$, completing the proof.

$\square$

**Lemma 10.** *There exists a metric reduction from MaxSAT to MSE-NAM.*

*Proof.* Given the CNF formula $\phi$, we first apply Lemma 9 to reduce it to a univariate neural network $f_i : \mathbb{R} \to \mathbb{R}$. We pick an arbitrary value $\mathbf{x}_i \in [0, 1]$. Let us assume for simplicity that $f_i(\mathbf{x}_i) = r$. We can "normalize" this value by appending an additional final layer with weight 1 and bias $-r$, yielding a modified model $f'$ for which $f'_i(\mathbf{x}_i) = 0$ (though, for ease of notation, we continue to write $f$). Note that under this transformation, the *optimal* value of $f$ is also shifted by $r$. Furthermore, we note that by appending an additional layer with weight $-1$, we can "mirror" the univariate component, yielding $-f_i$. We then construct a NAM $f$ consisting of $m^2$ univariate components $-f_i$. We now form a vector $\mathbf{x} \in \{0, 1\}^{m^2}$ by repeating the $\mathbf{x}_i$ value $m^2$ times. We set $\epsilon = [0, 1]$ and define the intercept of the NAM as $\beta_0 := m^2$. This completes the construction of the corresponding MSE-NAM instance. We now observe that it holds that:

$$-\sum_{i=1}^{m^2} f_i(\mathbf{x}_i) + \beta_0 = 0 + m^2 > 0 \tag{34}$$

In other words, this means that $f(\mathbf{x})$ receives the positive label, i.e., $f(\mathbf{x}) = 1$. Let us assume that some explanation $\mathcal{S}$ is a sufficient explanation of size $k$. For $\mathcal{S}$ to indeed qualify as sufficient, it must

satisfy the following:

$$-\sum_{i\in\mathcal{S}} f_i(\mathbf{x}_i) - \sum_{i\in\bar{\mathcal{S}}} \max_{\mathbf{z}_i\in[0,1]} f_i(\mathbf{z}_i) + \beta_0 \geq 0 \iff -\sum_{i\in\bar{\mathcal{S}}} \max_{\mathbf{z}_i\in[0,1]} f_i(\mathbf{z}_i) + \beta_0 \geq 0 \iff$$

$$-|\bar{\mathcal{S}}| \cdot \max_{\mathbf{z}_i\in[0,1]} f_i(\mathbf{z}_i) + m^2 \geq 0 \iff \max_{\mathbf{z}_i\in[0,1]} f_i(\mathbf{z}_i) \leq \frac{m^2}{|\bar{\mathcal{S}}|} = \frac{m^2}{m^2 - k} \tag{35}$$

This implies that a sufficient explanation of size $k$ exists if and only if:

$$(m^2 - k) \cdot \max_{\mathbf{z}_i\in[0,1]} f_i(\mathbf{z}_i) \leq m^2 \tag{36}$$

This also implies that a sufficient explanation of size $k'$ does *not* exist if and only if:

$$(m^2 - k') \cdot \max_{\mathbf{z}_i\in[0,1]} f_i(\mathbf{z}_i) > m^2 \iff m^2 - k' > \frac{m^2}{\max_{\mathbf{z}_i\in[0,1]} f_i(\mathbf{z}_i)} \tag{37}$$

We additionally note that $\max_{\mathbf{x}_i\in[0,1]} f_i(\mathbf{x}_i) \leq m - r \leq m$, since at most $m$ clauses can be satisfied. Therefore, it follows that:

$$\frac{1}{\max_{\mathbf{z}_i\in[0,1]} f_i(\mathbf{z}_i)} \geq \frac{1}{m} \tag{38}$$

Moreover, by applying Equation 37 to any subset that is not a sufficient explanation of size $k'$, we obtain that:

$$m^2 - k' > \frac{m^2}{\max_{\mathbf{z}_i\in[0,1]} f_i(\mathbf{z}_i)} \iff m^2 - k' > m \iff m^2 - k' \geq m + 1 \tag{39}$$

Putting everything together, we have established that there is *no* sufficient explanation of size $k'$ if and only if $m^2 - k' \geq m + 1$. Now, assume that an explanation $\mathcal{S}$ of size $k$ is *cardinally-minimal*. This implies that $\mathcal{S}$ satisfies Equation 36, and moreover, every explanation of size $k - 1$ fails to be sufficient. Hence, Equation 37 holds when we set $k' := k - 1$, yielding $m^2 - (k - 1) \geq m + 1$ which is equivalent to $m^2 - k \geq m$. We therefore obtain the following conclusion:

$$\frac{m^2}{m^2 - (k - 1)} < \max_{\mathbf{z}_i\in[0,1]} f_i(\mathbf{z}_i) \leq \frac{m^2}{m^2 - k} \tag{40}$$

Moreover, the following relation follows directly from simple algebraic manipulation:

$$\frac{m^2}{m^2 - k} - \frac{m^2}{m^2 - (k - 1)} = m^2 \cdot \frac{m^2 - k + 1 - (m^2 - k)}{(m^2 - k) \cdot (m^2 - k + 1)} = \frac{m^2}{(m^2 - k) \cdot (m^2 - k + 1)} \tag{41}$$

Hence, we obtain that:

$$\frac{m^2}{m^2 - (k - 1)} < \max_{\mathbf{z}_i\in[0,1]} f_i(\mathbf{z}_i) \leq \frac{m^2}{m^2 - (k - 1)} + \frac{m^2}{(m^2 - k) \cdot (m^2 - k + 1)} \tag{42}$$

Since we know that $m^2 - k \geq m$, it immediately follows that $m^2 - k + 1 > m$. Altogether, we obtain:

$$\frac{m^2}{m^2 - (k - 1)} < \max_{\mathbf{z}_i\in[0,1]} f_i(\mathbf{z}_i) < \frac{m^2}{m^2 - (k - 1)} + 1 \tag{43}$$

Overall, under our construction we obtain that $\max_{\mathbf{z}_i\in[0,1]} f_i(\mathbf{z}_i)$ is always confined to an interval whose width is strictly less than 1. From Lemma 9, we already know that $\max_{\mathbf{z}_i\in[0,1]} f_i(\mathbf{z}_i)$ equals the MaxSAT value of $\phi$ minus $r$ (the amount subtracted during the normalization step). Since the MaxSAT optimum is an integer, it follows directly that $\max_{\mathbf{z}_i\in[0,1]} f_i(\mathbf{z}_i) + r$ is an integer as well. Moreover, the interval $[\frac{m^2}{m^2 - (k-1)} + r, \frac{m^2}{m^2 - (k-1)} + r + 1]$ must contain exactly one integer because it contains $\max_{\mathbf{z}_i\in[0,1]} f_i(\mathbf{z}_i)$ and its length is strictly less than 1. Consequently, given the encoding of $\phi$ into the NAM instance and into the MSE-NAM problem, we can construct a simple polynomial-time function that, from the value of $k$, computes the unique integer in this interval — recovering the optimal number of satisfiable clauses of $\phi$.

$\square$

## D ON THEORY AND PRACTICE OF IMPORTANCE SORTING

### D.1 THE THEORETICAL WORST CASE BEHAVIOR OF THE PARAMETER $\xi_i$

We recall the previous Proposition 2, where we have shown that our algorithm performs a total number of steps equal to:

$$\frac{\beta_i - \alpha_i}{2^{k_i}} \leq \xi_i \iff k_i \leq \mathcal{O}\big(\log(\frac{\beta_i - \alpha_i}{\xi_i})\big). \tag{44}$$

A natural question is what happens when $\xi_i$ approaches $0$, or when the interval $\beta_i - \alpha_i$ becomes very large. In the following, we show that in such cases, our algorithm is still guaranteed to terminate after a linear number of queries in the size of the representation of the model given that each weight $w_{i,j}$ and bias $b_{i,j}$ in every univariate model is encoded as a rational number $\frac{p_{i,j}}{q_{i,j}}$ with $q_{i,j}, p_{i,j} \in \mathbb{N}$, given in binary.

**Proposition 5.** *Let $f$ be a NAM with ReLU activations. Then, the process of running Alg. 2, followed by Alg. 3 or Alg. 4 terminates after at most a linear number of queries to a neural network verifier, with respect to the size of the model $f$'s encoding.*

*Proof.* We show this proposition in three stages: (i) Firstly, we demonstrate that for each $i \in [n]$, the bit-length of the minimal value of the $i$'th univariate function, namely $min_{z \in B_\epsilon(\mathbf{x}_i)}\big(f_i(\mathbf{z}_i)\big)$ where $B_\epsilon(\mathbf{x}_i)$ is the $\epsilon$-ball around $\mathbf{x}_i$, is finite and is bounded by the encoding size of $f_i$. (ii) Secondly, we show that the bit lengths of the verification bound parameters $\alpha_i, \beta_i$ are also bounded by the encoding size of $f_i$. (iii) Finally, we use this claim to show that Alg. 2 terminates after at most a linear number of queries to a neural network verifier with respect to the model size encoding, from which it is straightforward to establish that running this algorithm, followed by either Alg. 3 or Alg. 4, also must terminate after at most a linear number of queries. We start off with the first part of the claim:

**Lemma 11.** *Consider a univariate ReLU neural network component $f_i \colon \mathbb{R} \to \mathbb{R}$, an input $\boldsymbol{x}_i \in \mathbb{Q}$, and a radius $\epsilon \in \mathbb{Q}$. Define the $\epsilon$-ball around $\boldsymbol{x}_i$ as $B_\epsilon(\boldsymbol{x}_i) := \{\boldsymbol{z}_i \in \mathbb{R} \colon \boldsymbol{x}_i - \epsilon \leq \boldsymbol{z}_i \leq \boldsymbol{x}_i + \epsilon\}$ for all $i \in [n]$. Then, $\min_{z_i \in B_\epsilon(\boldsymbol{x})}(f_i(\boldsymbol{z}_i))$ is finite, with a bit length that is bounded by the encoding size of $f_i$.*

*Proof.* We prove this claim for some general univariate model $f_i$. However for ease of presentation, we will define a copy "g" of $f_i$, i.e., $g \colon \mathbb{R} \to \mathbb{R}$ such that $g := f_i$ to avoid repetitive usage of the subscript $i$ . We will particularly prove that $\min_{z^0 \in B_\epsilon(x)}\big(g(z^0)\big)$ is finite with bit length that is bounded by the encoding size of $g$. We use the 0 subscript in $z^0$ to denote the fact that it's the "0" input layer in the ReLU model (i.e., the input layer).

As our NAM has ReLU activations, the minimal value, i.e., the value for which $\min_{z^0 \in B_\epsilon(x)}\big(g(z^0)\big)$ is obtained, is the solution to some linear program whose constraints are directly determined by the model's weights and biases. This observation is a standard cornerstone of neural-network verification. For completness, we provide a more explicit formalization here. As shown by (Sälzer & Lange, 2021), once the activation pattern (active/inactive state) of every ReLU constraint is "guessed", the verification query reduces to a linear program with the model parameters defining its constraints. The minimal value $\min_{z^0 \in B_\epsilon(x)}\big(g(z^0)\big)$ is therefore the minimal value of an LP problem induced by a "guess" of the activation patterns.

More particularly, given that we "guess" all activations $\alpha_{j,k} \in \{0, 1\}$ for each ReLU constraint that is the $j$'th variable of layer $k$, then one can define the following LP: *minimizing $g(z^0)$, subject to the input specifications:* $x - \varepsilon \leq z^0 \leq x + \varepsilon$, the linear specifications that are specified by $\mathbf{h}^k = W^k \mathbf{z}^{k-1} + \mathbf{b}^k$, for all $k \in \kappa$ where $\kappa \in \mathbb{N}$ denotes the number of layers in $g$, and the ReLU specifications that are given by setting $\mathbf{z}_j^k = \mathbf{h}_{kj}$ and $\mathbf{h}_{kj} \geq 0$ if $\alpha_{j,k} = 1$, or $\mathbf{z}_j^k = 0$ and $\mathbf{h}_{kj} \leq 0$ if $\alpha_{j,k} = 0$, where $\mathbf{z}^k$ denotes the output variables in layer $k$. The output $g(z^0)$ that is minimized within the linear program is simply the value of the variable $\mathbf{z}^\kappa$ (the value of the last output layer).

It is well known that any solution of a linear program admits finite precision (Vera, 1998; Dadush et al., 2020), where a naïve upper bound on the percision of the solution is $\mathcal{O}(n \cdot L)$ where $n$ are the number of variables and $L$ is the maximum bit-length of any input coefficient. Hence, similarly,

in the LP that we have described, if $m_i$ denotes the overall number of weights and biases in $f_i$ and let $d$ denote the maximal bit length of any parameter in the model (i.e., weights/biases), meaning $2^d$ itself denotes the magnitude of this encoding, then the maximal value in turn can be encoded using $\mathcal{O}(m_i \cdot d)$ bits, which is linear in the encoding size of $f_i$.

It follows immediately that the optimal value of this linear program — namely, the minimum attained within a fixed linear region — is finite and bounded by the numeric precision of the program's parameters. Furthermore, since $\min_{z^0 \in B_\epsilon(x)}(g(z^0))$ is realized as the minimum over one of these linear regions determined by at least one activation pattern $\alpha_{j,k}$, the same bound on bit-length applies to $\min_{z^0 \in B_\epsilon(x)}(g(z^0))$ as well. $\quad\square$

By Lemma 11, it follows that $\xi_i$ (as defined in equation 2), which represents the smallest gap between two minimal univariate values over a continuous domain, also admits finite precision. In particular, its bit-length is bounded by $\max_{i \in \{1,\ldots,n\}} \big(m_i \cdot \log(d)\big)$. We can now proceed to the remaining part of the proof:

**Lemma 12.** *Given that the weights, the biases, and the input interval of an univariate model $f_i$ have maximal bit length of at most $d_i$, the output interval's bounds $\alpha_i$ and $\beta_i$ are finite and their bit length is bounded by $m_i \cdot d_i$, where $m_i$ is the number of parameters in $f_i$.*

*Proof.* The parameters $\beta_i$ and $\alpha_i$ can be obtained using a wide range of bound-propagation techniques developed in neural network verification. Even under a "naïve" approach such as Interval Bound Propagation (IBP) (Zhang et al., 2020; Gowal et al., 2018) — which propagates bounds layer by layer by computing an outer approximation at each step and exploiting the monotonicity of ReLU activations — the resulting bounds still have a (very loose) worst-case linear bit-length in the model size.

Formally, for each $k \in [\kappa]$, where $\kappa$ is the number of layers of the univariate model, the values of the propagated interval in the $k$'th layer, denoted $\alpha^k, \beta^k$, can be computed from the interval of the $(k-1)$'th layer, $\alpha^{k-1}, \beta^{k-1}$, using the weights and biases of the $k$'th layer via interval arithmetic:

$$\alpha^k = \min_{\alpha^{k-1} \leq z \leq \beta^{k-1}} \big(\mathrm{ReLU}(W^k z + b^k)\big), \qquad \beta^k = \max_{\alpha^{k-1} \leq z \leq \beta^{k-1}} \big(\mathrm{ReLU}(W^k z + b^k)\big),$$

where $\alpha^0 = x_i - \epsilon$ and $\beta^0 = x_i + \epsilon$. Since $\mathrm{ReLU}(x) = \max(0, x)$, the activation does not increase the bit-length. A multiplication can increase the bit-length by at most $d_i$, and an addition can increase it by at most 1. Thus, calculating $W^k z + b^k$ in layer $k$ — which consists of element-wise multiplications of $\alpha^{k-1}, \beta^{k-1}$ with numbers of maximal bit-length $d_i$, followed by summation and adding the bias — increases the bit-length by at most $d_i$ (from multiplications) and by at most $m_i$ in total (from additions across all layers). As a result, the bit-length of $\alpha^k, \beta^k$ can be at most $d_i + \ell_k$ larger than that of $\alpha^{k-1}, \beta^{k-1}$, where $\ell_k$ is the number of parameters in the $k$'th layer.

Let $L$ denote the number of layers in $f_i$. We can then bound the bit-length of the output interval values:

$$d_i \cdot L + \sum_{k \in [\kappa]} \ell_k \ \leq \ d_i \cdot m_i + \sum_{k \in [\kappa]} \ell_k \ \leq \ d_i \cdot m_i + m_i = m_i(d_i + 1).$$

Overall, the propagated input bounds have bit-length at most $\mathcal{O}(m_i \cdot d_i)$. $\quad\square$

Now, using Lemma 11 and Lemma 12 we can conclude the proof for Prop. 5. Particularly, in every iteration of the loop in Alg. 2, Line 5 halves the bounds of the verification problem and thereby decrease by 1 the bit length of the approximated minimal values. Line 12 then checks, for every pair of univariate networks $f_i, f_j$ ($i, j \in [n]$), whether either the lower bound of $f_i$ is greater than or equal to the upper bound of $f_j$, or vice versa. By Lemma 11, the minimal values of $f_i$ and $f_j$ have bit length bounds of $m_i \cdot d, m_j \cdot d$ bits, respectively. Moreover, from Lemma 12, the bit lengths of $\alpha_i, \beta_i$ are bounded from above by $m_i \cdot (d + 1)$.

Consequently, the terms $\beta_i - \alpha_i$ and $\xi_i$ can both be represented using at most $\mathcal{O}(m \cdot d)$ bits. thus, a binary search over this quantity will require no more than $\mathcal{O}(m \cdot d)$ steps. Specifically, after a linear number of steps in $m \cdot d$, the condition in Line 12 must be satisfied: for every pair of $i, j \in [n]$, either the lower bound of $f_i$ will match the upper bound of $f_j$, or the lower bound of $f_j$ will match the upper bound of $f_i$ — including the case where the true minimal values are equal.

Put differently, we execute $n$ parallel threads, and after at most a linear number of iterations in the encoding size of the largest univariate function (itself linear in the total function size), Prop. 5 ensures termination. Furthermore, since Alg. 3 and Alg. 4 respectively run in linear or logarithmic time in $n$ (which is likewise linear in the model encoding size), it follows that the overall procedure terminates after at most a linear number of queries in the model encoding size.

$\square$

We note that since this bound is linear in the model encoding size, it represents a substantial improvement over the exponential worst-case complexity observed in general neural networks (Barceló et al., 2020b).

**Relation to verifier and machine precision.** In practice, it is highly unlikely to reach this worst-case bound on precision (see Appendix F.5 for empirical evidence). Hypotheticallly, even the verifier or machine precision will dominate this parameter. Moreover, our experiments show that the values of $\xi_i$ in trained NAMs are in fact rarely exactly on the decision boundary. Consequently, in our empirical evaluations, the effective precision parameter governing the runtime is $\xi_i$. As demonstrated in our experiments — where we efficiently compute cardinally minimal explanations for commonly used NAMs — and in the ablation study of this parameter in Appendix F.5, this yields a substantial improvement in overall runtime.

## D.2 PRACTICAL OPTIMIZATIONS FOR IMPORTANCE SORTING

While analyzing the behavior of Alg. 2 in practice, we noticed that the algorithm makes unnecessarily many verifier calls in certain edge cases. We briefly mention these here, along with our (sound) optimizations based on heuristics. As in Sec. 4.1, we consider the case where $f(\mathbf{x}) = 1$, which requires us to find bounds $[l_i, u_i]$ for the exact minimum value $f_i(\mathbf{x}_i^*)$ for each feature $i \in [n]$. The case $f(\mathbf{x}) = 1$ follows symmetrically. The two edge cases we consider deal with cases where the exact minimum value is (close to) either of the bounds:

**Lower bound close to exact minimum.** Please note that as we iteratively refine the bounds using neural network verification calls, the verifier sometimes gives a counterexample $\mathbf{x}'$ along with the verification result. We can use $\min(f_i(\mathbf{x}_i'), m)$ as our new upper bound for the minimum $u_i$ (Alg. 2, line 9). We observed that, in practice, this speeds up the sorting process, particularly as the verifier often returns a counterexample $x'$ that is close to or even the exact minimum. This works well in practice for the case $[0, u_i]$ on NAMs with ReLU activations, as $\mathbf{x}_i'$ just has to hit the same piece-wise linear region containing $\mathbf{x}_i^*$, which gets mapped to $0$ through $f_i$.

**Upper bound close to exact minimum.** It also sometimes occurs that the upper bound $u_i$ is already (close to) the exact upper bound. Thus, instead of iteratively running verification queries until $l_i$ converges to $u_i$, it helps to directly test if $u_i - \delta$ can still be reached for some small $\delta \in \mathbb{R}_+$. A good heuristic to switch to this test is if the verifier concludes that $f(\mathbf{x}_i^*) \in [m_i, u_i]$ but is unable to return a counterexample demonstrating this (as the verifier has to hit $u_i$ more or less exactly given it's close proximity to $f(\mathbf{x}_i^*)$).

These two optimizations often reduce the number of verifier calls described in Prop. 2 to very few verifier calls. In practice, we have seen that often only 3 verifier calls per feature are required to determine the total ordering.

# E EXTENSIONS TO ADDITIONAL SETTINGS

## E.1 EXTENSION TO MULTI-CLASS CLASSIFICATION

For multi-class classification, let us assume the winner class $t = f(\mathbf{x}) = \arg\max_{j \in [c]} f_j(\mathbf{x})$ (Sec. 2.3). Then, we can distinguish between two cases: (i) *winner-vs.-all* explanations, and (ii) all pair-wise *winner-vs.-one* explanations, where each case can be reduced to a binary classification task, where the new binary network in (i) is given by $\hat{f}(\mathbf{x}) = f_t(\mathbf{x}) - \max_{j \neq t} f_j(\mathbf{x})$, and for (ii) by $\hat{f}_j(\mathbf{x}) = f_t(\mathbf{x}) - f_j(\mathbf{x})$, $j \in [n]$, with the class 1 corresponding to the original winner class $t$. Then, we can apply Alg. 2 and Alg. 4 to generate the respective explanations.

## E.2 Extension to regression

Let us assume we want to find the subset $\mathcal{S}$ that is sufficient to determine that the prediction will always be *larger* than some deviation $\delta \in \mathbb{R}_+$ to the original output (the same result for finding the same guarantee for a prediction which is *smaller* will be symmetricly opposite). The first part of the algorithm will be identical to the case of binary classification with a *positive* outcome (the other use-case will align with a *negative* outcome).

---

**Algorithm 6** Regression: Parallel Interval Importance Sorting

---

**Input:** NAM $f$, input $\mathbf{x} \in \mathbb{R}^n$, perturbation radius $\epsilon_p \in \mathbb{R}_+$

1: **for each** feature $i \in [n]$ *in parallel* **do**
2:     Extract initial bounds $\alpha_i, \beta_i$ for $f_i(\tilde{\mathbf{x}}_i)$ such that $\tilde{\mathbf{x}}_i \in \mathcal{B}_p^{\epsilon_p}(\mathbf{x}_i)$
3:     $l_i \leftarrow \alpha_i, u_i \leftarrow \beta_i$
4:     **while** True **do**
5:         $m_i \leftarrow \frac{l_i + u_i}{2}$
6:         **if** verify( $\forall \tilde{\mathbf{x}}_i \in \mathcal{B}_p^{\epsilon_p}(\mathbf{x}_i), \ f_i(\tilde{\mathbf{x}}_i) \geq m_i$ ) **then**
7:             $l_i \leftarrow m_i$
8:         **else**
9:             $u_i \leftarrow m_i$
10:         **end if**
11:         $\Delta l_i \leftarrow f_i(\mathbf{x}_i) - l_i \ ; \ \Delta u_i \leftarrow f_i(\mathbf{x}_i) - u_i$
12:         **if** for all $i \neq j$ it holds that: $\Delta u_i \geq \Delta l_j$ or $\Delta u_j \geq \Delta l_i$ **then**
13:             **Break**
14:         **end if**
15:     **end while**
16: **end for**
17: **return** $\arg \text{sort}([(\Delta l_1, \Delta u_1), \ldots, (\Delta l_n, \Delta u_n)])$ in ascending order

---

Now we move on to the second part of the algorithm for the regression case:

---

**Algorithm 7** Regression: Greedy Cardinally-Minimal Linear Explanation Search

---

**Input:** NAM $f$, input $\mathbf{x} \in \mathbb{R}^n$, perturbation radius $\epsilon_p \in \mathbb{R}_+$, output deviation $\delta \in \mathbb{R}_+$

1: $\mathcal{S} \leftarrow [n]$
2: **for each** feature $i \in [n]$, ordered by Alg. 6 **do**           $\triangleright$ suff$(f, \mathbf{x}, \mathcal{S}, \delta, \epsilon_p)$ holds
3:     **if** suff$(f, \mathbf{x}, \mathcal{S} \setminus \{i\}, \delta, \epsilon_p)$ **then**
4:         $\mathcal{S} \leftarrow \mathcal{S} \setminus \{i\}$
5:     **end if**
6: **end for**
7: **return** $\mathcal{S}$          $\triangleright$ $\mathcal{S}$ is a *cardinally-minimal* explanation concerning $\langle f, \mathbf{x}, \delta, \epsilon_p \rangle$

---

which is the same algorithm we used for binary classification (Alg. 3), but where now the evaluation of *suff* evaluates to checking whether fixing the feature subset $\mathcal{S}$ ensures that the output remains larger than the original input by less than $\delta$. Given this result, this can be extended to our logarithmic version (Alg. 4) as well:

# F Experimental Details and Ablation Studies

## F.1 Dataset and Experimental Details

**Datasets.** We evaluate our approach on four widely used benchmark datasets (Agarwal et al., 2021; Radenovic et al., 2022). covering both classification and regression tasks. The Breast-Cancer dataset contains 569 samples with 30 numeric features per sample. to classify tumors as malignant or benign. The CREDIT dataset includes $1,000$ samples with 20 attributes each, for assessing loan repayment probability. The FICO HELOC dataset comprises $10,459$ samples with 23 financial and demographic features, for predicting creditworthiness. These datasets collectively allow us to evaluate the performance and robustness of our method across different problem types, input

---

**Algorithm 8** Regression: Greedy Cardinally-Minimal Logarithmic Explanation Search

---

**Input:** NAM $f$, input $\mathbf{x} \in \mathbb{R}^n$, perturbation radius $\epsilon_p \in \mathbb{R}_+$, output deviation $\delta \in \mathbb{R}_+$

1:   $F \leftarrow$ total order of features (Alg. 6)
2:   $l \leftarrow 1$ ; $u \leftarrow n$
3:   **while** $l \neq u$ **do**
4:      $m \leftarrow \lfloor \frac{l+u}{2} \rfloor$
5:      **if** $\text{suff}(f, \mathbf{x}, \{F[1], \dots, F[m]\}, \delta, \epsilon_p)$ **then**
6:        $l \leftarrow m$
7:      **else**
8:        $u \leftarrow m - 1$
9:      **end if**
10: **end while**
11: $\mathcal{S} \leftarrow \{F[1], \dots F[m]\}$
12: **return** $\mathcal{S}$          ▷ $\mathcal{S}$ is a *cardinally-minimal* explanation concerning $\langle f, \mathbf{x}, \delta, \epsilon_p \rangle$

---

Table 3: Varying perturbation radius $\epsilon$ on all datasets.

| $\epsilon$ | Breast Cancer | | CREDIT | | FICO HELOC | |
|---|---|---|---|---|---|---|
| | Size ($\downarrow$) | Time [s] ($\downarrow$) | Size ($\downarrow$) | Time [s] ($\downarrow$) | Size ($\downarrow$) | Time [s] ($\downarrow$) |
| 0.01 | 4.00±4.24 | 35.60±1.34 | — | — | — | — |
| 0.1 | 4.45±3.98 | 115.08±73.48 | 1.79±1.07 | 76.76±88.21 | 2.77±1.62 | 131.31±141.90 |
| 0.2 | 6.29±3.50 | 143.81±121.27 | 2.80±2.07 | 97.16±34.45 | 3.88±1.58 | 136.82±77.97 |
| 0.5 | 9.76±2.75 | 146.01±64.76 | 3.76±2.62 | 132.67±36.76 | 5.59±1.80 | 317.92±222.07 |

dimensions, and domain characteristics. **Models.** We trained 3 binary classification NAMs on the first three datasets, and a regression NAM on the last model. We follow the standard architectures in (Agarwal et al., 2021; Radenovic et al., 2022), and train for each feature a network with hidden layers of size $(64, 64, 32)$. The accuracy of the models for Breast Cancer, CREDIT, and FICO HELOC are $97.37\%$, $94.92\%$, and $69.02\%$, respectively.

**Evaluation.** All presented results are averaged over 50 samples with a time out of 600s, perturbations are w.r.t to the $\ell_\infty$-norm on the normalized input and a perturbation radius $\epsilon = 0.5$ is used if not stated otherwise. We filtered trivial samples where, e.g., all features are returned as an explanation. For the Breast Cancer dataset, we use $\epsilon = 0.01$ for an interesting comparison to the local strategies. Our experiments are running on a Ubuntu 24.04 machine with 64GB RAM and 13th Gen Intel(R) Core(TM) i7-1365U. The CREDIT experiments are run on a Ubuntu 24.04 machine with a Intel(R) Xeon(R) Platinum 8380 CPU @ 2.30GHz and two NVIDIA A100-PCIE with 40GB. If not otherwise specified, all experiments were limited to 32 CPU threads. In figures, we show the median along with a shaded region depicting the $25/75\%$ quantiles.

## F.2   ABLATING THE PERTURBATION RADIUS

In formal XAI, the generated explanations heavily depend on the chosen perturbation radius $\epsilon$. In Tab. 3, we show how the size and generation time of the explanation change with varying $\epsilon$. Generally, the explanation size and the generation time increase with $\epsilon$, which is expected as previously obtained minimal explanations indeed become insufficient and the verification queries become harder to solve as $\epsilon$ is increased. Please note that for $\epsilon = 0.01$, insufficiently many samples for proper averages were found where the explanation is non-trivial on CREDIT and FICO.

## F.3   NUMBER OF PROCESSED FEATURES OVER TIME

In this experiment, we demonstrate how our approach — after the initial sorting phase — processes the features much quicker to obtain a cardinally-minimal explanation, as it only requires a logarithmic number of verification queries to do so. This even outperforms approaches that obtain subset-minimal approaches, including the time needed to sort the features (Fig. 5).

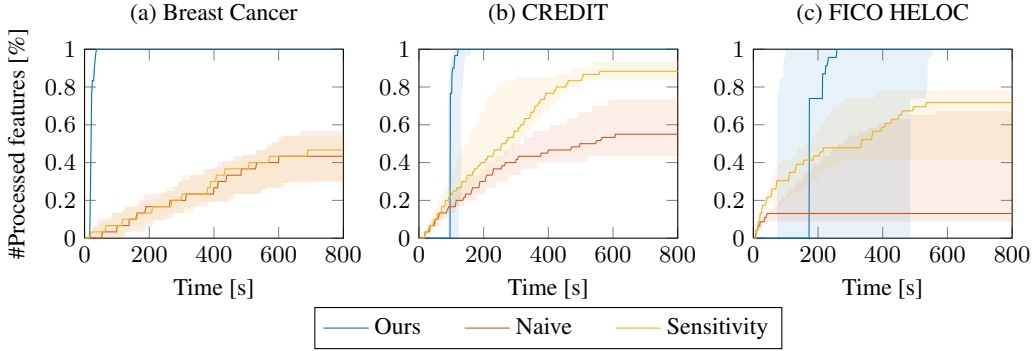

Figure 5: Number of processed features over time.

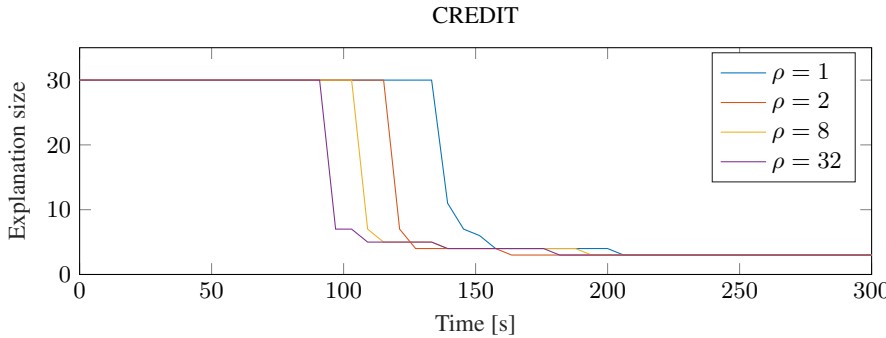

Figure 6: A comparison of explanation generation with different number of processors $\rho$.

### F.4 UNDERSTANDING THE PARALLELIZATION: ABLATING THE NUMBER OF PROCESSORS

A key factor influencing the runtime of our approach — particularly the sorting of univariate component importances in Alg. 2 is the *number of processors* allocated for parallelizing the logarithmic binary search. To assess this effect, we conducted an ablation study with varying processor counts.

Fig. 6 illustrates the impact of parallelization on both explanation size and computation time. In particular, the time to sort the features according to their importance (Alg. 2) can be reduced as the number of processors $\rho$ increases, as the bound refinement can be parallelized. In contrast, the subsequent explanation generation (Alg. 4) is barely impacted by the number of processors $\rho$. Importantly, even with only a single processor (i.e., no parallelization), our algorithm still computes *cardinal*-minimal explanations — a harder task than subset-minimal ones — thus improving explanation size and computation time by design (compare to Fig. 3).

### F.5 ABLATION STUDY ON NEAR IDENTICAL FEATURES

Our approach relies on an efficient importance sorting of the features (Sec. 4.1). This subsection provides further insights into the required precision to effectively sort them (Prop. 2).

We first show the frequency of each required precision in our datasets in Fig. 7. The impact of these precisions on the sorting time is visualized in Fig. 8. These results show that the required precision is not arbitrarily small (smallest around $10^{-8}$), and the sorting time is also influenced by other factors, such as the verifier, in practice. This implies that, in practice, sorting based on the parameter $\xi_i$ is highly effective, which aligns with, and helps explain, the strong empirical performance we observe when computing cardinally minimal explanations for NAMs.

To isolate the effect, we created an ablation study using a synthetic neural network with near identical feature networks $f_i$, $i \in [2]$, where only the bias in the last layer of $f_2$ is shifted slightly. Thus, a large bias shift corresponds to no overlap between the computed bounds, and a small bias shift

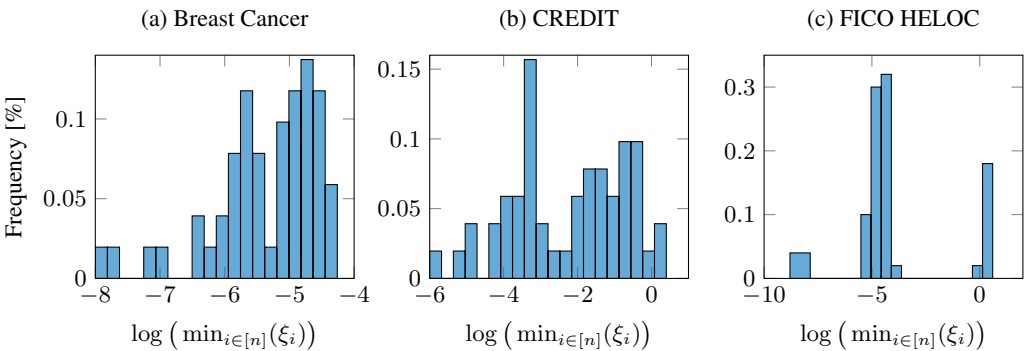

Figure 7: Frequencies of the required precision to sort the features according to their importance.

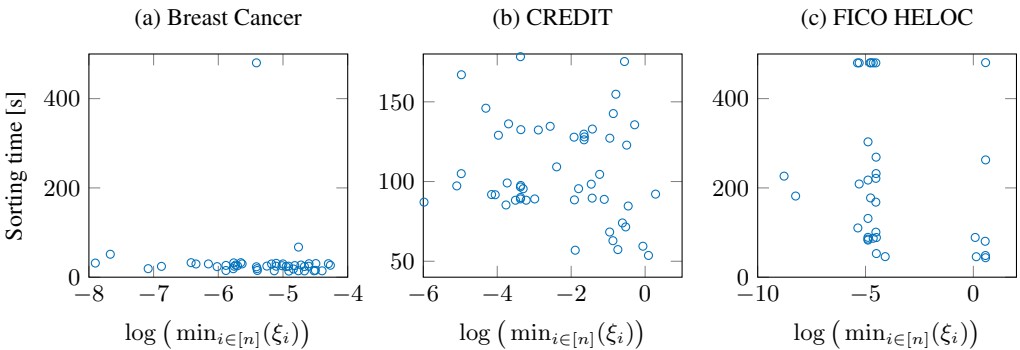

Figure 8: Impact of the required precision on the sorting time.

results in nearly identical bounds. We use the same network architecture as the ones trained on the real data sets, and do not apply any optimizations discussed in Appendix D.2 in this study. We record the sorting time for different bias values starting from $1$ (no overlap even for initial bounds) down to $10^{-7}$ (almost entirely overlapping) in Tab. 4. We observe that the sorting time only increases linearly with the order of magnitude of the bias shift (corresponding to $\xi_i$) due to the binary search, confirming our result in Prop. 2.

# G   DISCLOSURE: USE OF LARGE LANGUAGE MODELS (LLMS)

A large language model (LLM) was engaged solely as a writing aid to polish language and enhance expression. It played no role in generating research ideas, designing the study, conducting the analysis, or interpreting results. These aspects were performed entirely by the authors.

Table 4: Influence of near identical features on the sorting time on synthetic example.

| Bias Shift | Sorting Time [s] |
| --- | --- |
| 1 | 0.08 |
| 0.1 | 7.58 |
| 0.01 | 12.71 |
| 0.001 | 16.24 |
| 0.0001 | 20.95 |
| 0.00001 | 28.37 |
| 0.000001 | 31.54 |
| 0.0000001 | 36.57 |

