# OpenReview forum: "Provably Explaining Neural Additive Models"
_ICLR.cc/2026/Conference — ICLR 2026 Poster_

### Official Review · Reviewer_uiyA · 2025-10-17

**Soundness:** 4
**Presentation:** 4
**Contribution:** 4
**Rating:** 6
**Confidence:** 3

**Summary:**

This paper proposes a new algorithm to extract cardinal-minimal sufficient explanations for Neural Additive Models (NAMs).
It does so by exploiting key design choices of NAMs, showing how this family of models supports explanations with guarantees.

This is achieved as follows. First, the paper introduces a method to rank features based on how much they influence the final prediction. Then, after this ranking is obtained, an algorithm is discussed to exploit this order to efficiently explore which features to remove from the current sufficient explanation until a cardinal-minimal explanation is obtained.

**Strengths:**

- Making minimal sufficient explanations scaling is a timely and valuable research direction.

- The proposed idea is simple but effective

- The paper flows well

- The paper is formal and precise in its claims

**Weaknesses:**

### Proof of Proposition 3 is unclear

- **W1:** In line 842, the authors write *Let 1 ≤ l ≤ n represent the last feature added to $S$ in line ?? of Alg. 3*. However, I do not understand what this statement refers to, as in no part of Alg. 3 features are added to $S$.

- **W2:** In line 843, the authors write *Then, for $S^{'}= S \setminus \\{ l \\}$, it follows that: $suff(f, x, S, \epsilon)$ does not hold true, implying that $S^{'}$ is not a sufficient explanation*, but it is unclear why this is true. I feel there could be some writing issues in this paragraph that prevent me from understanding the proof.



### Definition of Sufficient explanation

- **W3:** Definition 1 expresses the condition $\\forall \\tilde{x} \\in B_p^{\\epsilon_p}(x)$, where $B_p^{\\epsilon_p}(x) = \\{\\tilde{x} \\in R^{n} | ...\\}$. However, this condition seems to be impossible to satisfy, as real numbers are dense and therefore would require to run the sufficiency check for an infinite number of $\\tilde{x}$.


- **W4:** The set of allowed perturbations is defined over an $\\epsilon$-ball around $x$. Nonetheless, defining a fixed $\\epsilon$ for every feature could hinder some feature-specific behaviors induced by, for example, different magnitudes of features. For example, let us consider a binary feature representing the gender of a person (0=male, 1=female), and a function $f_i(x_i)$ shaped as: $f_i(x_i)=100$ if $x_i < 0.5$, and  $f_i(x_i)=-100$ if $x_i >= 0.5$. Then a value of $\\epsilon=0.1$ will not be able to check for the sufficiency of the explanation when the gender of the person is switched from male to female, as this feature will be assigned zero importance even if its true impact is actually very high.


### Figure 1 unclear

- **W5:** Figure 1 is difficult to interpret, and its description is not self-contained. I suggest improving its description. For example, the part on *users might wrongly believe that only feature 1 yields negative outputs, while feature 2 can also flip the classification* is non-trivial to a non-expert reader.


**Minors**

- Propositions 2 and 3 are in reverse order in the Appendix.

- The link between lines 46 and 47 is a bit unclear. In fact, while the first sentence in the paragraph is referred to sufficient explanations, the following sentence refers only to cardinal-minimal explanations, and not cardinal-minimal sufficient explanations, making the context of the sentence less clear.

- In line 52, the authors write: *Consequently, existing methods focus on (locally) subsetminimal explanations which are typically suboptimal in size, potentially large, and thus less informative than their globally minimal counterparts*, which is, however, not clear why this is true without further context. I would recommend adding an explanation of why this is the case.


- I personally find the wording local vs global sufficient explanation a bit misleading, as this could be confused with standard notions of local (instance-level) and global (model-level) explanations. I believe cardinal-minimal and subset-minimal are already discriminative enough and may not need further quantifications.

**Questions:**

**Q1:** In line 2 of Alg. 2, how is the operation $\\in$ implemented? Is this a uniform random sampling?

**Q2:** How is the proposed algorithm applicable to NAMs for data types different from tabular data, like images and graphs ?

---

> ### Author Response · Authors · 2025-11-25
>
> We thank the reviewer for their helpful and constructive feedback and for acknowledging the significance of our work.
>
>
> **Sufficiency check on infinitely many samples**
>
>
> We thank the reviewer for raising this important point, which we will highlight more clearly in the final version. Our algorithm provides *provable guarantees over the entire continuous input domain*. In neural network verification, reasoning is indeed carried out over *infinitely many* points in a *continuous* space, but crucially, this is *not* done via sampling, which would make exact guarantees impossible. Instead, neural network verification relies on the following principle: if the activation pattern (active/inactive state) of every ReLU neuron were known in advance, the resulting model would reduce to a purely linear function. In that case, verifying a property over the *entire continuous region* becomes equivalent to solving a single linear program (LP), which is both sound and efficient. In theory, iterating over all possible activation patterns (which is *exponential* in the number of neurons) would allow exact verification.
>
> The core computational challenge in neural network verification is therefore avoiding this exponential enumeration. Modern verifiers overcome this through scalable *branch-and-bound*-style methods that iteratively refine bounds on each neuron and propagate interval constraints forward to derive sound bounds on the output. These methods have achieved remarkable improvements in scalability in recent years [e.g., 1-6]. Our algorithm employs these verifiers as its backend - specifically the state-of-the-art $\alpha$-$\beta$-CROWN verifier [1], and therefore inherits their ability to certify properties over entire continuous domains. We design our NAM-based algorithm so that it strategically ``sorts’’’ features based on their interval bounds, allowing us to fully leverage the efficiency gains offered by such verifiers. Moreover, our algorithm dramatically reduces the number of required neural network verification queries - cutting down from the worst-case exponential number of queries needed for general neural networks to a level that is highly practical and attainable for NAMs.
>
> We thank the reviewer for this comment. In the final version, we will make it clearer that our guarantees hold over entire continuous domains and explicitly explain how this connects to neural network certification.
>
> **Typo in proof of Proposition 3**
>
> We thank the reviewer for catching this! Indeed, we corrected the phrasing from “the last feature that was added to $S$” to “the last feature that was attempted to be removed from $S$”, which is the accurate description. This change also clarifies why $S' := S \setminus \{\ell\}$ is not a sufficient explanation: the algorithm attempted to remove $\ell$ in the final iteration and halted, and by the loop’s stopping condition this implies that $S'$ cannot be sufficient. We appreciate the reviewer’s attention to this detail, and believe the revised phrasing helps make the proof clearer.
>
>
> **Choosing the proper value of $\epsilon$**
>
> We agree with the reviewer that the choice of $\epsilon$ must be made carefully in order to obtain a meaningful explanation. Importantly, our approach is valid for *any* choice of $\epsilon$, and can naturally be extended to handle any disjoint union of continuous regions in the input space - an aspect we will emphasize more clearly in the final version. While choosing $\epsilon = 0.1$ may be reasonable for certain continuous features (e.g., “blood pressure” in tabular data), different choices are appropriate for thresholded inputs that behave as strict 0/1 indicators based on whether a value falls below or above 0.5, as in the reviewer’s example. In such cases, one may choose a larger $\epsilon$ radius or even take the entire relevant range (e.g., $\epsilon = [0,1]$ for normalized features).
> Although selecting this parameter is indeed important, it concerns the inherent task of obtaining provably minimal and sufficient explanations, rather than something specific to our NAM-based algorithm. We have also conducted an ablation study in Appendix D.2 examining how the choice of $\epsilon$ affects runtime. We thank the reviewer for highlighting this point and will include a more detailed discussion in the final version.

---

> ### Author Response · Authors · 2025-11-25
>
> **Would the results apply to other data types?**
>
> Yes, our algorithm is entirely independent of the underlying data modality. In principle, it can be applied not only to tabular data but also to computer vision, graph-structured data, or any domain in which a NAM serves as the core model architecture. The reason we focus our experiments on tabular benchmarks is simply that these are the standard evaluation setting for NAMs.
>
> While NAMs are impressively expressive, their inherently interpretable structure limits their ability to capture certain more complex feature interactions. Tabular tasks are therefore an excellent fit: many of them involve high-stakes, safety-critical decisions (e.g., medical diagnostics) where interpretability is essential, and at the same time they typically do not require the rich high-order interactions that arise in domains such as computer vision. For this reason, NAMs have been predominantly evaluated on tabular datasets - not due to any limitation of our algorithm, which is fully agnostic to the data type.
>
> As a promising direction for future work, we believe it would be highly interesting to extend our framework to enriched NAM variants that are better suited for non-tabular modalities, and to use our algorithm as the core certifiable-explanation engine behind them. We think that our results provide a strong foundation for exploring these directions.
>
>
> **In Alg. 2, line 2 - how is the $\in$ implemented to extract initial output bounds for certification?**
>
> We will clarify the two occurrences of the “$\in$” notation in Algorithm 2 (Line 2 and Line 6). Importantly, these do *not* correspond to uniform sampling, but rather to the certification of the model over an entire continuous domain. As discussed in our previous response, a neural network verification engine can certify robustness over continuous input regions - conceptually by reducing the verification task to a collection of linear programs that can be solved exactly.
>
> In our implementation, this is handled by the $\alpha$-$\beta$-CROWN verifier [1], which is the current state-of-the-art in neural network verification [2-3]. Such verifiers operate by first propagating initial input bounds through the network to obtain an outer approximation of the output. They then iteratively refine these bounds in a computationally intensive procedure until they become sufficiently tight to certify the property of interest. Accordingly, the use of “$\in$” in Line 6 corresponds to invoking the verifier to certify the property over the entire continuous region. The notation in Line 2 refers to the initial bounds supplied by the verifier before the refinement process begins.
>
> We thank the reviewer for drawing our attention to this subtle point. We will make sure to clarify this in the final version, as we believe it will improve the readability and understanding of our method.
>
> **Additional minor suggestions, typos, and presentation**
>
> We thank the reviewer for identifying these small yet important points. We have updated the manuscript accordingly. In particular, we agree with the concern about the potential ambiguity between “local” and “global” explanations when referring to explanations with global guarantees (over the entire dataset). Following the reviewer’s suggestion, we now consistently specify either “subset-minimal” or “cardinally-minimal” in all relevant instances to avoid confusion. We have additionally added a running-example with a new figure (Figure 1, Section 2) to clearly illustrate the distinction between subset-minimal explanations and cardinally-minimal explanations. We have also expanded the running example in Fig. 2 to further clarify this point, as requested by the reviewer. Thank you for these important suggestions!
>
>
> [1] Beta-crown: Efficient Bound Propagation with Per-Neuron Split Constraints for Neural Network Robustness Verification (Wang et al., NeurIPS 2021)
>
> [2] First Three Years of the International Verification of Neural Networks Competition (VNN-COMP) (Brix et al., STTT 2023)
>
>
> [3] The Fifth International Verification of Neural Networks Competition (vnn-comp 2024): Summary and results (Brix et al., STTT 2024)
>
>
> [4] Scalable Neural Network Verification with Branch-and-bound Inferred Cutting Planes (Zhou et al., NeurIPS 2024)
>
>
> [5] SDP-CROWN: Efficient Bound Propagation for Neural Network Verification with Tightness of Semidefinite Programming (Chiu et al., ICML 2025)
>
>
> [6] Clip-and-Verify: Linear Constraint-Driven Domain Clipping for Accelerating Neural Network Verification (Zhou et al., NeurIPS 2025)

---

> > ### Comment · Reviewer_uiyA · 2025-11-26
> > **Official comment by Reviewer**
> >
> > Thank you for the clarifications, which addressed my concerns. I will maintain my positive score.

---

### Official Review · Reviewer_mvRn · 2025-10-20

**Soundness:** 4
**Presentation:** 3
**Contribution:** 3
**Rating:** 8
**Confidence:** 4

**Summary:**

This paper presents a novel algorithm for computing provably cardinality-minimal explanations for Neural Additive Models (NAMs). The authors focus on post-hoc, per-instance explanations: given a trained NAM f and an input x, they seek to compute a subset of features S \subseteq [n]that is sufficient to guarantee the same prediction under bounded perturbations of the remaining features (an
\epsilon-ball). Among all sufficient subsets, the goal is to find one of minimum cardinality (the global optimum).

The paper provides a novel contribution to the state-of-the-art in the broad area of explainability with provable guarantees (in this case, minimality). The paper focuses on NAMs, which to the best of my knowledge it is still a Still a niche but growing area in the interpretability subfield. They are Not widely used in industry production pipelines yet. but research interest persists. In fact,  (NAMs) occupy an interesting middle ground in machine learning — they’re not mainstream, but they are important in specific contexts where interpretability and nonlinear modelling both matter. Their main limitation is that in a pure NAM, features don’t interact directly because the model assumes additivity. This means that the effect of each feature x_i on the output y is independent of any other feature x_j, which can be a strong limitation in some practical settings.

The proposed algorithm proceeds in two stages. In the first stage each univariate subnetwork f_i(x_i) is verified independently to estimate its influence on the model’s decision. This is done via parallelised binary search over feature importance intervals. In Stage 2, after sorting features by importance, a binary search identifies the globally cardinal-minimal sufficient subset of features that provably determines the model’s prediction. This reduces complexity from exponentially many calls to the network to logarithmically many.

Experiments on standard tabular benchmarks demonstrate feasibility and show smaller, faster provable explanations than prior methods; sampling-based visualisations were also shown to be unreliable in some cases, whereas the proposed method always produces verifiably sufficient explanations.

Overall, I am supportive of this paper. It makes a meaningful and well-justified contribution to formal explainability by showing that NAMs enable efficient computation of globally minimal sufficient explanations -- something previously infeasible for general neural networks. With minor revisions, I feel that this paper is a valuable contribution to the state of the art.

**Strengths:**

Strengths.

(1) Addresses an important formal-XAI problem (provable minimal explanations) and achieves a stronger guarantee (global cardinality minimality) than most prior work for neural networks.

(2) Elegant exploitation of the additive NAM structure to reduce verification complexity.

(3) Clear algorithmic presentation with theoretical propositions and proofs in the appendix.

(4) Empirical results convincingly illustrate both efficiency and the need for provable guarantees.

**Weaknesses:**

Limitations / concerns.

(1) The guarantees rely on an exact, sound verifier and on refining intervals until importance bounds separate. In practice, verifier soundness, numerical issues, or timeouts can undermine the provable guarantees; the authors acknowledge this but more discussion of practical mitigations (timeouts, numerical tolerances) would be useful.

(2) The attractive complexity (logarithmic verifier calls) is obtained assuming parallel refinement across features. The number of wall-clock verifier calls depends on available parallelism; the paper states the complexity in terms of ρ processors. If you cannot parallelise, wall-clock time will be higher (though still fewer full-model queries than naive approaches).

(3) The method requires refining per-feature bounds until features’ importance intervals separate (so a total order exists). If two features have extremely close effects, the required precision may be tiny, increasing verification cost. The complexity bound explicitly depends on that precision factor. The propositions report complexity in terms of that separation quantity.

(4) The related work is largely comprehensive, but the paper omits a relevant reference on computing cardinality-minimal explanations for a different class of networks (monotonic fully-connected networks). That work should be cited and briefly contrasted. While that setting differs from NAMs (monotonic FCNs vs. additive univariate subnetworks), the goals and guarantees are similar; I recommend citing and briefly contrasting that work to highlight how different architectural restrictions enable tractable exact explanations.

@inproceedings{DBLP:conf/ijcai/HarzliG023,
  author       = {Ouns El Harzli and Bernardo Cuenca Grau and Ian Horrocks},
  title        = {Cardinality-Minimal Explanations for Monotonic Neural Networks},
  booktitle    = {Proceedings of the Thirty-Second International Joint Conference on
                  Artificial Intelligence (IJCAI) 2023},
  pages        = {3677--3685},
  publisher    = {ijcai.org},
  year         = {2023},
  doi          = {10.24963/IJCAI.2023/409}
}

**Questions:**

Suggestions.

(1) Expand on practical verifier settings, in particular timeout and floating/rounding issues.

(2) Add an ablation that shows performance as a function of the separation parameter or a synthetic example with near-identical features.

(3) Cite and comment on the work by El Harzli et al. at IJCAI-2023.

---

> ### Author Response · Authors · 2025-11-25
>
> We thank the reviewer for their helpful and constructive feedback and for acknowledging the significance of our work.
>
>
>
>
> **The impact of parallelization on the tractability of the algorithm**
>
> We thank the reviewer for raising this important point. We agree that this parameter is central, and indeed our complexity analysis is provided with respect to $\rho$, which denotes the number of available processors. As expected, increasing the number of processors improves the approximation quality. We additionally conducted an ablation study (Appendix D.4) demonstrating how the number of processors significantly accelerates the runtime.
>
> Interestingly, we also observe that the runtime often improves even with a small number of processors, and in many cases even with a *single* processor, despite the fact that the worst-case number of queries is larger in this setting compared to the standard greedy algorithm. This behavior arises because the verification queries are applied to *univariate* models $f_i$, which are substantially easier to certify than the full model $f$. That said, increasing the number of processors yields the most desirable outcome, reducing the number of required verification queries to logarithmic in the precision separating the minima of two univariate components.
>
>
>
> **Problems that might arise with the verifier such as numerical instability and timeouts**
>
> We thank the reviewer for highlighting this point and have expanded our discussion accordingly in the revised version, specifically in Sec. 7 and Appendix C. Naturally, as with any approach that relies on neural network verification, issues such as verifier precision, numerical instability, and potential timeouts are inherent. However, these challenges stem from the verification tools themselves rather than from our method, and they are actively being addressed by the verification community. As these improvements continue to evolve, our method will directly benefit from them as well.
>
> **Runtime in the specific edge-case where $\xi_i$ is very small**
>
> We thank the reviewer for raising this very interesting point. We first emphasize that even in the hypothetical scenario where the gap between the minimum achievable values of two univariate components approaches zero, our algorithm is still guaranteed to terminate after a worst-case linear number of verification queries (in the size of the model encoding). This follows from the fact that the bit-precision required to encode an optimal solution to a ReLU neural network verification query is at most linear in the model size (we have formalized this statement in the revised manuscript under Appendix C.1 and have referred to it in the main paper at the end of Section 4.1). Consequently, even this highly contrived scenario is still far better than the worst case for general neural networks, which may require exponentially many queries.
>
> Importantly, *in practice* we consistently observe that the gaps between the minima of different univariate components are sufficiently large to make the sorting step highly efficient - requiring only a *logarithmic* number of queries with respect to this precision factor. Following the reviewer’s suggestion, we conducted an empirical study of this parameter and added the results to Appendix E.5. Specifically, we include: (1) a histogram showing that the relevant features typically fall within a range that enables reliable separation and sorting; (2) an analysis of how varying $\xi_i$ affects runtime, demonstrating that our algorithm efficiently handles this step regardless of the observed value; and (3) a synthetic pathological example where $\xi_i \to 0$, illustrating that - even in this extreme case - the required precision remains logarithmic in scale, preventing any exponential blow-up in practice. Overall, these results confirm our empirical finding that the algorithm reliably computes cardinally minimal explanations efficiently, thanks both to the practical magnitude of this parameter and to the logarithmic dependence of the runtime on it.

---

> > ### Comment · Reviewer_mvRn · 2025-11-26
> >
> > I thank the authors for their comments and clarifications. I have no further questions.

---

> ### Author Response · Authors · 2025-11-25
>
> **Relation to the work of El Harzli et al.**
>
> We thank the reviewer for raising this point. We agree that the work of El Harzli et al. on sufficient explanations for monotonic neural networks deserves discussion, and we have added a detailed discussion in the revised manuscript (see Related Work). Their approach indeed provides an alternative route to interpretability and is, in many ways, complementary to ours. At a high level, the complexity hardness of finding minimum sufficient explanations arises from two core challenges: (1) establishing *sufficiency*, which requires robustness verification (an NP-hard task), and (2) obtaining *minimality*, which may require exploring exponentially many subsets in the worst case.
>
> Monotonic networks simplify the *sufficiency* component by enabling polynomial-time verification, but retain the difficulty of minimality. NAMs, on the other hand, simplify the *minimality* component while preserving the hardness of sufficiency. As such, the two approaches address opposite sides of the problem, and we now explicitly highlight this complementarity in the final version.
>
> We are particularly excited about our results on NAMs, given their growing prominence in the interpretability literature [e.g., 1-7]. NAMs achieve strong performance - especially on tabular tasks - and ongoing extensions and related interpretable neural architectures we believe can open the door to promising directions for obtaining certifiable explanations efficiently and more broadly.
>
> [1] Neural Additive Models: Interpretable Machine Learning with Neural Nets (Agarwal et al., NeurIPS 2021)
>
> [2] Neural Basis Models for Interpretability (Radenovic et al., NeurIPS 2022)
>
> [3] The Intelligible and Effective Graph Neural Additive Network (Bechler-Speicher et al., NeurIPS 2024)
>
> [4] Generalizing Neural Additive Models via Statistical Multimodal Analysis (Kim et al., TMLR 2024)
>
> [5] Node-GAM: Neural Generalized Additive Model for Interpretable Deep Learning (Chang et al., ICLR 2021)
>
> [6] NAISR: A 3D Neural Additive Model for Interpretable Shape Representation (Jiao et al., ICLR 2024)
>
> [7] Neural Additive Models for Location Scale and Shape: A Framework for Interpretable Neural Regression Beyond the Mean (Thielmann et al., AISTATS 2024)

---

### Official Review · Reviewer_f9gy · 2025-10-22

**Soundness:** 1
**Presentation:** 1
**Contribution:** 2
**Rating:** 2
**Confidence:** 4

**Summary:**

This paper focuses on explainable artificial intelligence and aims to provide concise explanations for the predictions made by Neural Additive Models (NAMs). The primary issue addressed in this study is as follows: given a classifier $ f $ represented by a NAM, an input data instance $ x $ that requires an explanation, and a ball $ B $ centered at $ x $, the goal is to identify a feature subset $ S $ of the minimum size. This subset must ensure that for every instance $ z $ within the ball $ B $, if the values of $ z $ and $ x $, restricted to the features in $ S $, are indistinguishable, then the classifications made by $ f $ for both $ z $ and $ x $ are the same. Such an explanation $ S $ is referred to as a (ball-restricted) minimum-size abductive explanation or a minimum-size sufficient reason.

To address this problem, the authors propose a two-stage method. In the first stage, the univariate functions $ f_i $ are sorted based on their importance intervals. In the second stage, a minimal-size explanation $ S $ is derived using a greedy approach. The paper includes formal proofs for the correctness and complexity of this method, and it presents comparative experiments conducted on four different datasets that support the theoretical findings.

**Strengths:**

**S1.** The problem of computing minimum-size sufficient reasons is $\Sigma_p^2$-hard for neural networks (Barcelo et al., 2020). Therefore, it is reasonable to explore simpler model classes that can lead to efficient algorithms with a reasonable number of calls to an NP oracle. In this context, Neural Additive Models are a rational choice.

**S2.** Experimental results across four datasets demonstrate that concise explanations can be generated in a reasonable amount of time.

**Weaknesses:**

**W1.** The computational complexity of finding minimum-size sufficient reasons for the class of NAMs has not been proven. Identifying the corresponding complexity class (P, NP, $\Sigma_p^2$) is crucial for justifying the overall interest in this approach.

**W2.** Barcelo et al. (2020) have already demonstrated that the problem is solvable in polynomial time for linear threshold functions. Essentially, the theoretical results presented in this study are just an extension of their previous work, with the primary innovation being the computation of an ordering of non-intersecting intervals. I am not entirely convinced that this alone is sufficient for publication in ICRL.

**W3.** The correctness and runtime complexity of the main algorithm used to compute a total ordering of non-intersecting intervals (Alg. 2) appear to be flawed. In particular, the runtime complexity tends toward infinity as the value of $\xi$ approaches zero.

**W4.** The clarity of the paper could be improved. Currently, there are several issues related to notation. Additionally, the organization of the paper could benefit from the inclusion of some proofs (or at least sketches of proofs) for the most significant results.

**Questions:**

The following comments and questions are related to the aforementioned weaknesses.

**C1.** What is the computational complexity of finding minimum-size sufficient reasons for the class of NAMs? Is this problem in NP? In other words, does the decision version of the problem have a polynomial-time certificate? Additionally, is the problem NP-hard, or is it even more difficult?

**C2.** As mentioned earlier, Barcelo et al. (2020) have demonstrated that the problem known as "Minimum Sufficient Reason" (MSR) is in P for the class of linear threshold functions (LTs). Essentially, MSR is the decision version of the problem being explored in this study, where the class of LTs is replaced by the class of NAMs, and the instance space is defined as a ball centered at $x$. For the class of LTs, Barcelo et al. advocate a strategy that involves sorting each feature $x_i$ based on its importance, which is measured by the weight $w_i$ and the class $f(x)$. The next step is to determine whether the top $k$ features in this ordered list provide a sufficient explanation for $x$ and $f$.

If I’m not mistaken, this study adopts a similar strategy, with the primary challenge being the computation of the minimum value of each univariate function $f_i$ over a specified ball centered at $x$. Therefore, it is important to discuss the overall approach in comparison to Barcelo et al.’s strategy, as well as to elaborate on the concept of using "non-intersecting intervals" to approximate feature importances.

**C3.** As mentioned earlier, the primary innovation of this study is to compute an ordering of “non-intersecting” intervals, from which the final explanation is derived. From this perspective, it is essential to demonstrate that the main algorithm (Alg. 2) is correct. Assume without loss of generality that $f(x) = 1$ with $x = (1,1)$ and consider two univariate functions, $f_1$ and $f_2$, where the minimum of $f_1$ (over the ball centered at $x$) is slightly below the minimum of $f_2$.  Does the algorithm guarantee that, at the end of the final iteration, the output intervals $(l_1, u_1)$ and $(l_2, u_2)$ satisfy the property $u_1 < l_2$ (indicating that $x_1$ precedes $x_2$ in the final ordering)? Additionally, how many iterations of the main loop are required to derive such intervals?

Regarding the last question, the complexity analysis appears to be flawed. Again, consider the two univariate functions $f_1$ and $f_2$. Here, assume that $f_1(z) = f_2(z) = w$, where $w$ is a constant scalar. According to Line 12 of Algorithm 2, the algorithm terminates when $\Delta u_1 \geq \Delta l_2$. However, if $\Delta u_1 = \Delta l_2$, we find that $\xi_1 = 0$. This situation suggests that the runtime complexity might be infinite. Therefore, I am unconvinced that $\xi$ is the appropriate parameter for the complexity analysis of the algorithm, particularly when the global minima of different univariate functions are identical.

**C4.** The paper currently has several clarity issues that make it difficult to follow. For instance, the $i$th entry of a vector $\mathbf{x}$ is sometimes denoted as $\mathbf x_{(i)}$ (as defined in Line 93) and at other times as $x_{(i)}$ (in Section 4.1). Additionally, the explanation is occasionally referred to as $\mathcal{S}$ (in the main paper) and other times as $S$ (in Lemmas 2 and 3). Furthermore, the center of Figure 1 appears to be misplaced, and there are missing references (indicated by symbol ??) in the Appendix.

---

> ### Author Response · Authors · 2025-11-25
>
> We thank the reviewer for their helpful and constructive feedback. We appreciate the time and care invested in the review, as it highlighted several important points that we now understand should be clarified in the final version. We hope that our responses satisfactorily address all of the concerns.
>
> **The worst-case complexity of obtaining cardinally-minimal sufficient explanations in NAMs**
>
> We thank the reviewer for this important comment and agree that the discussion can be refined. First, we emphasize that *in practice* our algorithm consistently yields provably cardinally-minimal sufficient explanations for NAMs using only a small number of verifier queries - representing a substantial improvement over prior methods for general neural networks, where this task is infeasible. Our complexity analysis relies on two logarithmic factors: (i) the precision gap between the minimum attainable values of different univariate components, and (ii) the number of features (in the selection phase).
>
> Empirically, we observe that standardly trained NAMs exhibit a clear separation between minima of different univariate components, which enables efficient “sorting’’ of these components and allows our method to run effectively in practice. While the theoretical complexity case in which this precision parameter approaches zero is important to acknowledge, we will later explain why even under this extreme scenario our algorithm is still guaranteed to terminate after at most a *linear* number of queries in the encoding size of the model (still, a substantial improvement over standard neural networks). Nevertheless, we view this situation as a largely theoretical corner case: in all our experiments the separation is sufficiently large to allow efficient sorting, and we also provide empirical evidence for this behavior in the appendix.
>
> Following the reviewers’ suggestion, we added two new worst-case complexity results that clarify the theoretical nature of our problem (Appendix B.5), and we will update the final version to include these results accordingly. These results do not depend on the practical separation parameter $\xi_i$ that we observe in practice. In particular, we show that computing the minimal $k$ forming a sufficient explanation for a NAM lies in $\text{FP}^{\text{NP}}[\mathcal{O}(n)]$, meaning it can be solved with at most a *linear* number of NP-oracle queries. This matches our algorithm: even in the hypothetical worst case - *independent* of $\xi_i$ - it still uses only a linear number of verifier calls. Notably, even this worst case is a substantial improvement over the exponential number of oracle calls required for general neural networks, where the corresponding decision problem is $\Sigma^P_2$-hard [1].
>
> Second, we prove that the problem is $\text{OptP}[\mathcal{O}(\log n)]$-hard under metric reductions, meaning that *at least* a logarithmic number of verifier queries is unavoidable. This follows from a non-trivial metric reduction from MaxSAT. While this bound is not tight, and the exact worst-case complexity remains an interesting direction for future work, we stress that in practice the algorithm never approaches this behavior: the empirical separation between components keeps the runtime far below the theoretical worst case. We will include both proofs and their corresponding complexity results in the final version, and we thank the reviewer for this important suggestion.

---

> ### Author Response · Authors · 2025-11-25
>
> **Connection to results on binary linear threshold models established by Barcelo et al.**
>
> We thank the reviewer for raising this point and agree that it merits a more detailed discussion. Accordingly, we have expanded the discussion of this important work in our revised manuscript in the “related work” part of the paper and also in relevant sections in the main paper - around line 209, around line 241, around line 312, and around line 319. The important theoretical work of Barcelo et al. shows, among many results, that finding a minimum sufficient explanation can be solved in polynomial time for *linear (i.e., Perceptron) models* defined over *binary* inputs. While the additive structure creates some parallels, such as the need to “sort’’ features, our setting is substantially more challenging: NAMs are additive models composed of *neural networks* over *continuous* domains, which are far more expressive. The key difficulty here is computing the maximum or minimum of each univariate component over a continuous domain, a significantly harder task. In fact, we show that even this subproblem is $\text{OptP}[\mathcal{O}(\log n)]$-hard (Lemma 9).
>
> More specifically, we believe the following novel elements are the key factors driving the scalability of our algorithm for NAMs (which are all different from Perceptrons):
>
> 1. It is well known in neural network verification that computing *exact* output bounds is extremely difficult in practice. Recent progress in scalable verification comes from avoiding exact reasoning and instead using sound and *bounded* over-approximations, (e.g., Branch-and-Bound (BaB) as implemented in state-of-the-art verifiers like $\alpha$-$\beta$-CROWN [2]). Our algorithm follows the same principle: it incrementally tightens bounds to determine the ordering of features *before* exact values are computed, which would be computationally impractical, thereby enabling scalability.
>
>
> 2. Secondly, perhaps the largest scalability bottleneck in neural network verification is the exponential blowup from many non-linear constraints. Our algorithm avoids this by shifting most of the computation to a parallelized verification of the *univariate* components $f_i$, which are far smaller and easier to certify.
>
>
> 3. Third, because we only compute *bounds* on the minimum and maximum of each univariate component, rather than their exact values, the sorting step alone cannot guarantee the identification of a cardinally minimal sufficient explanation (a task that would be trivial in a purely linear model). Consequently, in the subsequent selection phase, we leverage these bounds to perform a binary search, which reduces the number of neural network verification queries required at this stage to a *logarithmic* amount.

---

> ### Author Response · Authors · 2025-11-25
>
> **Worst-case behavior in the edge-case when $\xi_i$ is very small**
>
> We agree that this is an interesting point deserving further discussion, although our experiments indicate that it represents a pathological edge case that we can avoid in practice.
>
> We begin by explaining why, even under a hypothetical worst-case, our algorithm is still guaranteed to terminate after at most a *linear* number of verifier queries in the size of the model’s encoding. Importantly, this pathological behavior does *not* arise in practice: empirically, the separation between the optima of different univariate components is never arbitrarily small, and we are therefore able to sort the components efficiently (we elaborate on this in the experimental section).
>
> Regarding the theoretical claim, we have discussed this in Section 4.1 and Appendix C.1 in the revised manuscript. A ReLU-network verification query reduces to solving a linear program whose constraints depend on the network’s weights and biases. Hence, the minimum/maximum value of each univariate component has *finite* precision, inherited from the LP’s solution precision. Concretely, if all weights and biases have rational precision $p_i / q_i$​, then any optimal LP solution has bit-length $\mathcal{O}(m \cdot \log(|p_i| + |q_i|)$, where $m$ is the number of weights/biases - i.e., *linear* in the model’s encoding size. Therefore, a binary search over each component’s minimum can require only a linear number of verifier calls before two intervals “collapse” to equality. This gives our algorithm a worst-case linear number of verifier queries, still far better than the exponential worst case for general neural networks, and consistent with the membership result in $\text{FP}^{\text{NP}}[\mathcal{O}(n)]$.
>
> Importantly, *in practice* we consistently observe that the gaps between the minima of different univariate components are sufficiently large to make the sorting step highly efficient - requiring only a *logarithmic* number of queries with respect to this precision factor. Following a suggestion by reviewer mvRn, we conducted an empirical study of this parameter and added the results to Appendix E.5. Specifically, we include: (1) a histogram showing that the relevant features typically fall within a range that enables reliable separation and sorting; (2) an analysis of how varying $\xi_i$ affects runtime, demonstrating that our algorithm efficiently handles this step regardless of the observed value; and (3) a synthetic pathological example where $\xi_i \to 0$, illustrating that - even in this extreme case - the required precision remains logarithmic in scale, preventing any exponential blow-up in practice. Overall, these results confirm our empirical finding that the algorithm reliably computes cardinally minimal explanations efficiently, thanks both to the practical magnitude of this parameter and to the logarithmic dependence of the runtime on it.
>
>
> **Additional minor comments on notations and typos**
>
> We thank the reviewer very much for pointing out these notational issues. We agree that these can easily lead to confusion and we corrected them in the revised version. We also double-checked all proofs in Appendix A and will add proof sketches to our final version. Thank you for these suggestions.
>
> [1] Model interpretability through the lens of computational complexity (Barcelo et al., NeurIPS 2020)
>
> [2] Beta-crown: Efficient Bound Propagation with Per-Neuron Split Constraints for Neural Network Robustness Verification (Wang et al., NeurIPS 2021)

---

> > ### Comment · Reviewer_f9gy · 2025-11-26
> > **Re: Official Comments**
> >
> > I thank the authors for their detailed responses.
> >
> > Well, the revised version of the paper is now in much better shape. In particular, the computational complexity results presented in Appendix B.5 clearly underscore the significance of this study and justify the proposed method.
> >
> > I like the idea presented in Appendix C for addressing the issue of vanishing values. But to improve the paper further, I would suggest including a formal proof (in Section C.1) demonstrating that the number of queries remains linear.
> >
> > I have updated my review accordingly and have raised my rating.

---

### Official Review · Reviewer_Bqyn · 2025-10-30

**Soundness:** 4
**Presentation:** 4
**Contribution:** 3
**Rating:** 8
**Confidence:** 2

**Summary:**

A computationally-efficient, novel method to compute explanations with provable guarantees for Neural Additive Models (NAMs). The explanations are guaranteed to be the smallest in size, globally. The method claims to be efficient in generating such certified explanations.

**Strengths:**

- Certifying explanations is a key open research direction: the research problem highlighted by the authors is well-defined and relevant to the ML community. To the best of my knowledge this is the first approach specifically designed for NAMs
- Logarithmic number of verification queries required is a step forward compared to baselines. Formal complexity analysis included.
- Convincing experimental campaign, aligned with best-practice in the community. Performance of the approach shows interesting increase over baselines, both in size reduction and time required to generate an explanation (which is a welcome contribution).
- Paper is clear and the authors train of thought is well articulated.

**Weaknesses:**

- Limited impact, as the proposed method is applicable to NAMs only by design.
- Although evaluation is convincing, the adopted proxy quality metric for explanations is size, where smaller = better. A human assessment of the perceived quality of the resulting explanations would have made the work stronger - although I acknowledge this is a significant addition to the work (i.e. yes, consensus in the literature is the smaller the better, but what if the resulting explanations are *too* small, and therefore too course grained to capture a use case subtleties?)
- The paper could use a running example across the paper section, to further clarify some of the key concepts (e.g. globally and locally cardinal-minimal feature subset). The HELOC example in Figure 1 is popular enough to be a good candidate.

**Questions:**

- Does the method support discrete input features (I suppose so, could you confirm)?
- What is the impact of perturbation radius $\epsilon_p$?
- Has the local subset minimality been introduced in the narrative exclusively to ensure a feasible experimental campaign while comparing against naive baselines?

---

> ### Author Response · Authors · 2025-11-25
>
> We thank the reviewer for their helpful and constructive feedback and for acknowledging the significance of our work.
>
> **Model specific explanation method due to focus on NAMs**
>
> We agree with the reviewer that, by design, our approach is a model-specific explanation method for NAMs. We would first like to emphasize that NAMs have attracted substantial attention in recent years within the interpretability literature due to their inherently interpretable structure, a highly desirable property in safety-critical domains [e.g., 1-7]. To the best of our knowledge, our work is the first to propose an algorithm that provides *certifiable* explanations for NAMs.
>
> A second reason we view our results as significant is that they demonstrate how a problem that is infeasible to solve for general neural networks can be made efficiently computable for NAMs. This suggests a promising pathway for identifying additional neural-network families for which explanations can be certified efficiently while preserving high expressivity. NAMs, as we show, constitute a strong first step in this direction. We believe that future research could extend this line of work to more expressive variants as well as to other families of “interpretable’’ neural network architectures. We think that our work provides an initial and significant step toward this broader research agenda.
>
>
> **Incorporating a running example**
>
> We thank the reviewer for this valuable suggestion. In response, we have added a new figure (Figure 1, Section 2) to clearly illustrate the distinction between subset-minimal explanations (local optima) and cardinally-minimal explanations (global optima). We have also expanded the running example in Fig. 2 to further clarify this point.
>
>
> **Are smaller explanations really “better” and the relation to the human-perspective**
>
> We agree with the reviewer that the motivation for minimality should be emphasized more clearly. Explanations with *minimality* guarantees are widely regarded as desirable across many prior works (e.g., [8-11], among many more), and it is indeed a standard assumption in the literature that *smaller explanations are better*.
>
> The core reason behind this is that *non-minimal explanations can be misleading*. For example, consider a medical setting where a clinician examines whether a subset of attributes is provably sufficient to diagnose or rule out a disease. If the explanation includes an unnecessary attribute - say, “blood pressure = X” even though it can be removed without affecting sufficiency - this may misinform the clinician about the true grounds for the model’s decision.
>
> We also recognize the reviewer’s point that humans often prefer simpler, coarser explanations. Importantly, this can be addressed without sacrificing minimality entirely: one can first define a simplified feature space (e.g., superpixels in images or feature aggregates in tabular data) and then compute a *minimal* sufficient explanation in that coarser space. This preserves the guarantee of minimality (under that feature space) while yielding explanations that are easier for humans to interpret. Our algorithm can naturally incorporate such domain simplifications and we will mention this in the final version.
>
> We agree that this distinction, and the broader point raised by the reviewer about minimality, merits further discussion and will be elaborated in the final version.

---

> ### Author Response · Authors · 2025-11-25
>
> **Extensions for discrete input spaces**
>
> We thank the reviewer for raising this interesting point. Because our sufficiency guarantees hold over entire continuous domains, any discrete domain contained within such a region naturally inherits these sufficiency guarantees (i.e., our guarantee is strictly stronger). We also note that, in principle, one could impose discrete constraints on the input encoding or explore richer hybrid input spaces that combine discrete and continuous features. We will include a discussion of this aspect in the final version.
>
> **The impact of the perturbation radius $\epsilon_p$**
>
> We agree with the reviewer that this is indeed an important parameter to look at and note that we have included an ablation study on the impact of the perturbation radius in Appendix D.2.
>
>
> **The narrative on (local) subset minimality**
>
> We discuss both notions because they are the two most commonly studied forms of minimality in the literature [e.g., 8-12]. Crucially, the *cardinally minimal* sufficient explanations we obtain - i.e., the global optimum - are substantially stronger and therefore considerably more desirable.
> Prior works have focused on “subset-minimal’’ explanations [e.g., 8-12] mainly because computing cardinally minimal explanations for standard neural networks is computationally prohibitive, typically requiring a worst-case exponential number of NP-hard neural network verification queries [8, 10]. Our algorithm, by contrast, solves this strictly harder task.
>
> In fact, attempting to benchmark our method against a true analogue that computes cardinally minimal explanations for general neural networks would be meaningless: such a “naive’’ method would simply time out in all cases, providing no informative comparison. For this reason, our empirical comparison is conducted against the more tractable objective of finding *locally minimal* (subset-minimal) explanations.
>
> Interestingly, we show that our algorithm not only finds strictly smaller explanations (as guaranteed), but also achieves *faster* runtimes - meaning that we compute a cardinally minimal explanation more quickly than prior methods compute only a subset-minimal one.
>
> We thank the reviewer for this comment and we will enhance its discussion in the final version.
>
>
>
> [1] Neural Additive Models: Interpretable Machine Learning with Neural Nets (Agarwal et al., NeurIPS 2021)
>
> [2] Neural Basis Models for Interpretability (Radenovic et al., NeurIPS 2022)
>
> [3] The Intelligible and Effective Graph Neural Additive Network (Bechler-Speicher et al., NeurIPS 2024)
>
> [4] Generalizing Neural Additive Models via Statistical Multimodal Analysis (Kim et al., TMLR 2024)
>
> [5] Node-GAM: Neural Generalized Additive Model for Interpretable Deep Learning (Chang et al., ICLR 2021)
>
> [6] NAISR: A 3D Neural Additive Model for Interpretable Shape Representation (Jiao et al., ICLR 2024)
>
> [7] Neural Additive Models for Location Scale and Shape: A Framework for Interpretable Neural Regression Beyond the Mean (Thielmann et al., AISTATS 2024)
>
> [8] Model interpretability through the lens of computational complexity (Barcelo et al., NeurIPS 2020)
>
> [9] Explaining, Fast and Slow: Abstraction and Refinement of Provable Explanations (Bassan et al., ICML 2025)
>
> [10] Abduction-based explanations for machine learning models (Ignatiev et al., AAAI 2019)
>
> [11] Distance-Restricted Explanations: Theoretical Underpinnings & Efficient Implementation (Izza et al., KR 2024)
>
> [12] VeriX: Towards Verified Explainability of Deep Neural Networks (Wu et al., NeurIPS 2023)

---

### Meta-Review · Area_Chair_qZ61 · 2025-12-30

**Summary:**

The paper introduces a novel algorithm for computing provably minimal (instance-level) explanations for NAMs. Importantly, the proposed method is computationally efficient, and the empirical results demonstrate its effectiveness on real-world datasets.

Reviewers raised relevant issues and requested clarifications, most of which were well addressed by the authors through revisions incorporated into the paper. All reviewers agreed on the soundness, relevance, and overall quality of the work, and I am therefore happy to recommend acceptance of the paper at the conference.

**Reviewer Concerns:**

Overall, I believe the authors’ rebuttal was very effective in addressing the raised concerns. Most reviewers acknowledged the responses, with no indication of any important issues remaining unresolved. Below, I provide further details.

> Reviewer ``Bqyn`` raised concerns regarding the perceived limited impact of the work and the use of explanation size as a quality measure. The reviewer also suggested adding more examples and providing further clarifications.

Overall, I believe the authors have sufficiently addressed these points by emphasizing the growing interest in NAMs, adding illustrative figures and an ablation study, and clarifying that minimality guarantees are a broadly adopted objective in the explanation literature.

> Reviewer ``f9gy`` raised concerns related to i) insufficient theoretical results; ii) limited novelty (as an extension of Barcelo et al., 2020); iii) notation; and iv) correctness (possibly identifying some corner cases).

In response, the authors added additional theoretical results and clarifications, and more clearly articulated the significance of their contributions relative to Barcelo et al. (2020). The reviewer acknowledged that the paper is now in much better shape and increased their rating.

> Reviewer ``mvRn`` drew attention to issues such as (i) the dependence of the theoretical results on the precision factor and parallelism, (ii) missing related work, and requested ablation studies showing performance as a function of the separation parameter and near-identical features.

The authors added a discussion in Appendix C addressing the corner case of very small precision factors, and included additional ablation study and relevant related works. The reviewer acknowledged the rebuttal and indicated no further questions.

> Finally, Reviewer ``uiyA`` raised concerns regarding the clarity of the steps in Proposition 3 and Figure 1, the choice of $\epsilon$, and issues in Definition 1.

During the discussion phase, the reviewer explicitly acknowledged that these concerns were addressed in the authors’ response.

**Reviewer Scores:**

Three out of four reviewers engaged in the discussion and acknowledged the rebuttal. Two reviewers (``mvRn`` and ``uiyA``) maintained their supportive ratings (6 and 8), while Reviewer ``f9gy`` increased their score (from 2 to 6) following the authors’ clarifications. Based on the discussion phase, I believe that all reviewers would have maintained supportive ratings had the discussion continued.

---

### Decision · Program_Chairs · 2026-01-26

Accept (Poster)